# Wasserstein Distributionally Robust Optimization Through the Lens of Structural Causal Models and Individual Fairness

**Ahmad-Reza Ehyaei**
Max Planck Institute for Intelligent Systems, Tübingen AI Center, Germany
ahmad.ehyaei@tuebingen.mpg.de

**Golnoosh Farnadi**[*]
Mila Québec AI Institute ; McGill University, Montréal, Canada
farnadig@mila.quebec

**Samira Samadi**
Max Planck Institute for Intelligent Systems, Tübingen AI Center, Germany
ssamadi@tuebingen.mpg.de

## Abstract

In recent years, Wasserstein Distributionally Robust Optimization (DRO) has garnered substantial interest for its efficacy in data-driven decision-making under distributional uncertainty. However, limited research has explored the application of DRO to address individual fairness concerns, particularly when considering causal structures and sensitive attributes in learning problems. To address this gap, we first formulate the DRO problem from causality and individual fairness perspectives. We then present the DRO dual formulation as an efficient tool to convert the DRO problem into a more tractable and computationally efficient form. Next, we characterize the closed form of the approximate worst-case loss quantity as a regularizer, eliminating the max-step in the min-max DRO problem. We further estimate the regularizer in more general cases and explore the relationship between DRO and classical robust optimization. Finally, by removing the assumption of a known structural causal model, we provide finite sample error bounds when designing DRO with empirical distributions and estimated causal structures to ensure efficiency and robust learning.

## 1 Introduction

Machine learning models must address discrimination because they often reflect and amplify biases present in their training datasets [31]. These biases can significantly influence decisions in domains such as healthcare [30], education [3], recruitment [18], and lending services [6]. Consequently, these decisions disproportionately affect individuals based on sensitive attributes like race or gender, perpetuating systemic discrimination.

To address and quantify unfairness, researchers have developed concepts like **group fairness** and **individual fairness** [45, 4]. Group fairness aims to achieve equitable outcomes across demographic groups, while individual fairness ensures that similar individuals receive similar treatment. Formally, with $\mathcal{V}$ as the feature space and $\mathcal{Y}$ as the label space, a model $h : \mathcal{V} \to \mathcal{Y}$ ensures individual fairness

---

[*]Lead scientific advisor on the project

38th Conference on Neural Information Processing Systems (NeurIPS 2024).

if it satisfies the condition in [20]:

$$d_{\mathcal{Y}}(h(v), h(v')) \leq L d_{\mathcal{V}}(v, v') \quad \text{for all } v, v' \in \mathcal{V}, \tag{1}$$

where $d_{\mathcal{V}}$ and $d_{\mathcal{Y}}$ are dissimilarity functions, often referred to as **fair metrics** on the input and output spaces. These functions capture the proximity of individuals and $L \in \mathbb{R}^+$ is a Lipschitz constant. The metric $d_{\mathcal{V}}$ reflects the intuition about which instances should be considered similar by the model.

Due to challenges in defining such metrics, group fairness is often prioritized in fairness literature because it more straightforwardly addresses observable disparities among distinct groups, making measurement and implementation easier in practice [8]. Therefore, it is crucial to study and formulate individual fairness under different assumptions in machine learning.

Individual fairness can be achieved through robust optimization methods such as **Wasserstein DRO**, which has gained significant attention for its applications in learning and decision-making [55, 43]. DRO incorporates a regularization term to mitigate overfitting [17, 26, 59]. By using a fair metric as the transportation cost function in computing the Wasserstein distance, models are designed to deliver consistent performance across varied data distributions, ensuring similar individuals receive comparable outcomes, thus satisfying individual fairness.

Incorporating causal structures and sensitive attributes into data models complicates using an individual fair metric as a cost function within the DRO framework. The fair metric must account for perturbations in sensitive attributes based on counterfactuals to ensure counterfactual fairness [22]. This can violate the positive-definite property, where $d(v, v') = 0$ implies $v = v'$, a key assumption in many DRO theorems [55, 43].

Although previous works [42, 67, 70, 69, 57] have attempted to apply DRO to address individual fairness, they often do not explore the implications when causal structures and sensitive attributes are present in the learning problem. These studies are typically limited to linear **Structural Causal Model (SCM)** with specific metrics and do not discuss the form of the regularizer for other classical DRO theorems when using a fair metric. To accurately compare our work with related studies, we will postpone this discussion until after presenting our results in Section 4.1.

## 1.1 Our Contributions

In this work, we adopt the definition of a *fair metric* from [22] to define a **Causally Fair Dissimilarity Function (CFDF)**, which delineates how to establish a fair metric through causality and sensitive attributes. Using CFDF, we introduce **Causally Fair DRO** and present a strong duality theorem for our approach. Under mild assumptions about CFDF and causal structure, we demonstrate that the DRO regularizer can be estimated, or in some cases can be explicitly solved. This estimation often leads to being more practical and computationally efficient than solving the min-max problem in (4), as supported by advancements in algorithms from previous research such as [14, 15]. Finally, Our numerical analysis of both real and synthetic data demonstrates the practicality of our theoretical framework in real-world applications. (§ 5). In summary, the main contributions of this work are:

- Define a causally fair dissimilarity function, an individual fair metric incorporating causal structures and sensitive attributes (Def. 1), along with its representation form (Prop. 1).
- Define a causally fair DRO problem with a causally fair dissimilarity function cost (§ 4).
- Present the strong duality theorem for causally fair DRO (Thm. 1).
- Provide the exact regularizer for linear SCM under mild conditions for the loss function in regression and classification problems (Thm. 2 and Thm. 3).
- Estimate the first-order causally fair DRO regularizer for non-linear SCM (Thm. 4).
- Provide the relation between classical robust optimization and causally fair DRO (Prop. 2).
- Demonstrate that under unknown SCM assumptions, by estimating the SCM or cost function, we have finite sample guarantees for convergence of empirical DRO problems (Thm. 5).

## 2 Preliminaries & Notations

**Data Model.** Let $\mathbf{V} \in \mathcal{V}$ denote a vector of feature space (predictor variables) and let $\mathbf{Y} \in \mathcal{Y}$ represent the response variable, such that $\mathbf{Z} = (\mathbf{V}, \mathbf{Y})$ comprises the observation variables with an

underlying probability $\mathbb{P}_*$. Furthermore, assume that the feature vector $\mathbf{V} = (\mathbf{A}, \mathbf{X})$ comprises both sensitive attributes $\mathbf{A} \in \mathcal{A}$ and non-sensitive attributes $\mathbf{X} \in \mathcal{X}$. Let $\{z^i = (v^i, y^i)\}_{i=1}^N$ represent the observations used to construct the empirical distribution $\mathbb{P}_N$, defined as $\mathbb{P}_N := \frac{1}{N} \sum_{i=1}^N \delta_{z^i}$, where $\delta_z$ is the Dirac delta function. Given a loss function $\ell : \mathcal{Z} \times \Theta \to \mathbb{R}$, the risk function for a parameter $\theta \in \Theta$ and a probability measure $\mathbb{P}$ is $\mathcal{R}(\mathbb{P}, \theta) = \mathbb{E}_{\mathbb{P}}[\ell(Z, \theta)]$. This leads to the common **empirical risk minimization** approach. This method seeks to find the minimizer $\theta_N^{\text{erm}}$ within the set $\theta_N^{\text{erm}} \in \arg\min_{\theta \in \Theta} \mathcal{R}(\mathbb{P}_N, \theta)$, as an empirical way to obtaining the optimal solution $\theta_*$, which is given by $\theta_* = \inf_{\theta \in \Theta} \mathcal{R}(\mathbb{P}_*, \theta)$.

Assume the feature space is represented by a **structural causal model (SCM)** $\mathcal{M} = \langle \mathcal{G}, \mathbf{V}, \mathbf{U}, \mathbb{P}_{\mathbf{U}} \rangle$ [51]. This model includes **structural equations** $\{\mathbf{V}_i := f_i(\mathbf{V}_{\text{Pa}(i)}, \mathbf{U}_i)\}_{i=1}^n$, which delineate the causal relations among an endogenous variable $\mathbf{V}_i$, its causal predecessors $\mathbf{V}_{\text{Pa}(i)}$, and an exogenous variable $\mathbf{U}_i$ representing unobservable factors. The model's structure is encapsulated in a directed acyclic graph $\mathcal{G}$. Exogenous variables are posited as mutually independent, enabling $\mathbb{P}_{\mathbf{U}}$ to be expressed as $\prod_{i=1}^n \mathbb{P}_{\mathbf{U}_i}$, assuming causal sufficiency and excluding hidden confounders [53].

**Counterfactuals.** In causal structures, data perturbation is achieved through **counterfactuals**, which are derived from **interventions** in SCMs. These interventions, conducted using $do$-calculus, include both **hard** and **soft** types [51]. Hard interventions fix a subset $\mathcal{I} \subseteq \{1, \ldots, n\}$ of features $\mathbf{V}_{\mathcal{I}}$ to a constant $\tau$, modifying their causal connections within the causal graph while maintaining the structural equations of other features [51]. This type of intervention is denoted as $\mathcal{M}^{do(\mathbf{V}_{\mathcal{I}} := \tau)}$ and its structural equations are obtained by:

$$\{\mathbf{V}_i := \tau_i, \quad \forall i \in \mathcal{I}; \quad \mathbf{V}_i := f_i(\mathbf{V}_{\mathbf{Pa}(i)}, \mathbf{U}_i), \quad \forall i \notin \mathcal{I}\}.$$

Soft interventions, on the other hand, adjust the functions in the structural equations, such as through additive interventions, without disrupting existing causal links [53]. In an **additive (or shift)** intervention, a value $\Delta \in \mathbb{R}^n$ is added to each feature within the SCM to enact manipulation:

$$\{\mathbf{V}_i := f_i(\mathbf{V}_{\mathbf{Pa}(i)}, \mathbf{U}_i) + \Delta_i\}_{i=1}^n.$$

In SCMs, counterfactuals are computed by modifying structural equations to reflect hard interventions on specific variables, thus exploring what would occur if the intervention was applied. Under the assumption of acyclicity, a unique function $F : \mathcal{U} \to \mathcal{V}$ exists such that $F(u) = v$. Acyclicity remains unchanged by either hard or shift interventions, allowing for the existence of modified functions $F^{do(\mathbf{V}_{\mathcal{I}} := \tau)}$ and $F^{do(\mathbf{V}_{\mathcal{I}} += \Delta)}$ corresponding to these interventions, respectively. The counterfactual outcome for a hard intervention can thus be calculated using $\mathbf{CF}(v, \tau) = F^{do(\mathbf{V}_{\mathcal{I}} := \tau)}(F^{-1}(v))$, and similarly, for a shift intervention, it is defined as $\mathbf{CF}(v, \Delta)$. These interventions are frequently applied in this analysis.

Counterfactuals involving the modification of sensitive attributes (termed **twins**) are essential for addressing individual-level fairness [40, 64]. Twins are generated by altering the sensitive attribute from $a$ to $a'$ across its domain $\mathcal{A}$. For any instance, $v \in \mathcal{V}$, a set of counterfactual twins is produced as $\{\ddot{v}_a = \text{CF}(v, a) : a \in \mathcal{A}\}$, facilitating the analysis of fairness by comparing outcomes under different sensitive attribute values.

**Counterfactual Identifiability.** To estimate the effects of interventions from observational data, counterfactuals must be **identifiable** within a causal framework. A notable example of such identifiable SCMs is the **additive noise models (ANMs)**, which suggest that structural equations can be represented as:

$$\{\mathbf{V}_i := f_i(\mathbf{V}_{\mathbf{Pa}(i)}) + \mathbf{U}_i\}_{i=1}^n \implies \mathbf{U} = (I - f)(\mathbf{V}) \implies \mathbf{V} = (I - f)^{-1}(\mathbf{U}) \qquad (2)$$

leading to a bijective mapping between $U_i$ and $V_i$, ensuring no loss of information from exogenous to endogenous variables[50]. This relationship implies that $\mathbf{V}$ can be derived from $\mathbf{U}$ through a bijective reduced-form mapping $F = (I - f)^{-1}$, where $I(x) = x$ is the identity function. Besides ANM, there are other counterfactually identifiable models such as LSNM [34] and PNL [71]. However, for the sake of simplicity, our focus remains on ANM. **Linear SCMs** is a specific instance of ANMs, characterized by linear functions $f_i$.

**Individual Fairness Through Robustness.** In machine learning, individual fairness [20] is achieved through robustness by ensuring that similar individuals receive similar outcomes, regardless of

variations in their inputs. This concept aligns with the notion of Lipschitz continuity in decision functions (Eq. 1), where small changes in input should not lead to excessively large changes in output.

Depending on how the uncertainty set is defined, various types of robust optimization can be employed. In **adversarially robust optimization** [44, 7], the uncertainty set is defined by introducing a slight perturbation $\delta$ based on the metric $d$ to the input data. The goal is to find the optimal $\theta$ that minimizes risk even under the worst-case perturbation quantity:

$$\mathcal{R}_\delta^{adv}(\mathbb{P}, \theta) = \mathop{\mathbb{E}}_{v \sim \mathbb{P}} \left[ \sup_{d^p(v, v+\Delta) \leq \delta} \ell(v + \Delta, y, \theta) \right], \tag{3}$$

where $p \in [0, \infty]$. This formulation ensures that the optimization considers the maximum potential loss within the defined perturbation bounds.

In **counterfactually robust optimization** [40, 37, 64, 23, 24], the uncertainty set is generated by twins, which are obtained by creating counterfactuals concerning all levels of the sensitive attribute. In this scenario, the worst-case loss quantity is obtained by calculating the maximum loss over the twins of the input data:

$$\mathcal{R}_\delta^{cf}(\mathbb{P}, \theta) = \mathop{\mathbb{E}}_{v \sim \mathbb{P}} \left[ \sup_{a \in \mathcal{A}} \ell(\ddot{v}_a, y, \theta) \right].$$

**Distributionally Robust Optimization** [43, 55] is a data-driven approach designed to minimize the discrepancies between in-sample and out-of-sample expected losses, using ambiguity sets based on Wasserstein distances. Consider a lower semi-continuous cost function $c(\cdot, \cdot) : \mathcal{Z} \times \mathcal{Z} \to [0, \infty]$ that satisfies $c(z, z) = 0$ for all $z \in \mathcal{Z}$, serving as a fair metric. The **optimal transport cost** between two distributions $\mathbb{P}, \mathbb{Q} \in \mathcal{P}(\mathcal{Z})$, is represented by:

$$W_{c,p}(\mathbb{P}, \mathbb{Q}) \triangleq \min_{\pi \in \mathcal{P}(\mathcal{Z} \times \mathcal{Z})} \left\{ \left( \mathop{\mathbb{E}}_{(z, z') \sim \pi} [c^p(z, z')] \right)^{\frac{1}{p}} : \pi_1 = \mathbb{P}, \pi_2 = \mathbb{Q} \right\},$$

Here, $\pi \in \mathcal{P}(\mathcal{Z} \times \mathcal{Z})$ denotes the set of all joint probability distributions, and $\pi_1$ and $\pi_2$ are the marginals of $\pi$ under first and second coordinates [54, 63]. When $c(z, z')$ acts as a metric (in mathematics term) on $\mathcal{Z}$, $W_{c,p}$ is called the **Wasserstein distance** [63].

An important ingredient in the DRO formulation is the description of the distributional uncertainty region $\mathbb{B}_\delta(\mathbb{P})$ that is defined by optimal transport cost:

$$\mathbb{B}_\delta(\mathbb{P}) := \{\mathbb{Q} \in \mathcal{P}(\mathcal{V}) : W_{c,p}(\mathbb{Q}, \mathbb{P}) \leq \delta\}.$$

DRO problem minimizes worst-case loss quantity:

$$\mathcal{R}_\delta(\mathbb{P}, \theta) \triangleq \sup_{\mathbb{Q} \in \mathbb{B}_\delta(\mathbb{P})} \{\mathbf{E}_\mathbb{Q}[\ell(\mathbf{Z}, \theta)]\}, \tag{4}$$

and obtained the $\theta_N^{\mathrm{dro}} \in \arg\min_{\theta \in \Theta} \mathcal{R}_\delta(\mathbb{P}_N, \theta)$. The main tool in DRO is the **strong duality theorem** [26, 46], which converts an infinite-dimensional problem into a finite optimization problem. The theorem states that:

$$\sup_{\mathbb{Q} \in \mathbb{B}_\delta(\mathbb{P})} \left\{ \mathop{\mathbb{E}}_{v \sim \mathbb{Q}} [\psi(v)] \right\} = \inf_{\lambda \geq 0} \left\{ \lambda \delta^p + \mathop{\mathbb{E}}_{v \sim \mathbb{P}} [\psi_\lambda(v)] \right\}, \tag{5}$$

where $\psi_\lambda(v)$ is defined as $\psi_\lambda(v) := \sup_{v' \in \mathcal{V}} \{\psi(v') - \lambda d^p(v, v')\}$.

## 3 Causally Fair Dissimilarity Function

The key to robust optimization and individual fairness is the metric that measures individual similarity. This section outlines the properties of such a metric in a causal framework to protect sensitive attributes. We begin with an illustrative example.

**Example 1** *Let $\mathcal{M}_1$ and $\mathcal{M}_2$ represent two SCMs describing the relationships among the variables gender (**G**), education (**E**), and income (**I**). $\mathcal{M}_1$ models these variables as independent, whereas $\mathcal{M}_2$ specifies a linear causal relationship:*

$$\mathcal{M}_1 = \begin{cases} \mathbf{G} := \mathbf{U}_G, & \mathbf{U}_G \sim \mathcal{B}(0.5) \\ \mathbf{E} := \mathbf{U}_E, & \mathbf{U}_E \sim \mathcal{N}(0, 1) \\ \mathbf{I} := \mathbf{U}_I, & \mathbf{U}_I \sim \mathcal{N}(0, 1) \end{cases}, \quad \mathcal{M}_2 = \begin{cases} \mathbf{G} := \mathbf{U}_G, & \mathbf{U}_G \sim \mathcal{B}(0.5) \\ \mathbf{E} := \mathbf{G} + \mathbf{U}_E, & \mathbf{U}_E \sim \mathcal{N}(0, 1) \\ \mathbf{I} := \mathbf{G} + 2\mathbf{E} + \mathbf{U}_I, & \mathbf{U}_I \sim \mathcal{N}(0, 1) \end{cases}$$

*Where $\mathbf{U}_G$ represents the population distribution of gender, modeled by a Bernoulli distribution, while $\mathbf{U}_E$ and $\mathbf{U}_I$ are intrinsic talents for academic and income achievements, respectively, modeled by normal distributions. To compare individuals, let's consider the $L_1$ norm on non-sensitive attributes ($d(v,v') = |e - e'| + |i - i'|$). If two individuals have less than a 0.1 unit difference, they are deemed similar. Now, consider an individual with data $v = (M, 1, 1)$. Based on experience, we expect that a perturbation in educational talent by .05 units will not significantly alter this individual's status. We model this perturbation with a shift intervention $\Delta = (0, .05, 0)$. In Model 1, the result $\textbf{CF}(v, \Delta) = (M, 1.05, 1)$ is considered similar to $v$. However, in Model 2, $\textbf{CF}(v, \Delta) = (M, 1.05, 1.1)$ results in a distance of $d(v, \textbf{CF}(v, \Delta)) = 0.15$, indicating dissimilarity. In the presence of causality, one attribute can be amplified multiple times in the final feature space. Therefore, we need to control our intuition of dissimilarity between the exogenous variables and the feature space.*

*To protect against gender bias, we need to ensure that people with the same intrinsic characteristics but different genders behave similarly. This is modeled by a counterfactual change in gender. In Model 1, $\textbf{CF}(v, F) = (F, 1, 1)$ shows no difference ($d(v, \ddot{v}_F) = 0$). However, in Model 2, $\textbf{CF}(v, F) = (F, 0, -2)$ results $d(v, \ddot{v}_F) = 4$, which means that they are not similar.*

The example 1 demonstrates that in the presence of causality and protected variables, the standard $l_p$-norm or any metric fails to accurately capture the intuition of similarity. In these scenarios, a dissimilarity function should incorporate counterfactuals and uniformly control for non-sensitive perturbations to effectively capture proximity. This approach is further elaborated in the following definition. Before proceeding, we introduce some notation. For a vector $v$ or $u$, we define $P_{\mathcal{A}}(\cdot)$ and $P_{\mathcal{X}}(\cdot)$ as the projections onto the sensitive and non-sensitive parts, respectively.

**Definition 1 (Causally Fair Dissimilarity Function)** *Let $d : \mathcal{V} \times \mathcal{V} \to [0, \infty]$ be a dissimilarity function defined on the feature space $\mathcal{V}$, generated by a SCM $\mathcal{M}$. Let $\mathbf{A}$ denote a set of sensitive attributes, and $\mathcal{I}$ represent their corresponding index within $\{1, \ldots, n\}$. The metric is called a causally fair dissimilarity function or **CFDF** if it adheres to the following properties:*

- ***Zero Dissimilarity for Twin Pairs:** For any $v \in \mathcal{V}$ and $a \in \mathbf{A}$, the dissimilarity $d(v, \ddot{v}_a)$ between an instance and its twins is zero.*

- ***Guaranteed Similarity for Minor Perturbations:** For every $v \in \mathcal{V}$ and any $\delta > 0$, there exists an $\epsilon$ such that for any sufficiently small intervention ($\|\Delta\| \le \epsilon$) on the non-sensitive attributes ($P_{\mathcal{A}}(\Delta) = 0$), the distance $d(v, \textbf{CF}(v, \Delta))$ remains less than $\delta$.*

To understand the shape of $d$ under the assumptions of Def. 1, we must first recognize that the CFDF needs to be defined on a larger space than $\text{Range}(\mathcal{M})$ [22]. This is because, generally, when $\mathcal{M}$ is intervened upon by some sensitive attribute level $a$, we have $\text{Range}(\mathcal{M}) \subseteq \bigcup_{a \in \mathcal{A}} \text{Range}(\mathcal{M}^{do(\mathbf{A}:=a)})$. The complete space encompassing all counterfactual values can be defined as follows.

**Definition 2 (Parent-Free Sensitive Attribute SCM)** *Consider $\mathcal{M}$ with sensitive attributes indexed by $\mathcal{I}$. The parent-free sensitive attribute SCM denoted as $\mathcal{M}_0$, is derived from $\mathcal{M}$ by removing the causal effects of parents of sensitive attributes and replacing their exogenous variables with indigenous ones. The structural equations for $\mathcal{M}_0$ are as follows:*

$$\mathbf{V}_i^0 := \begin{cases} \mathbf{U}_i & \mathbf{U}_i := \mathbf{V}_i \sim \mathbb{P}_{\mathbf{V}_i}, & i \in \mathcal{I} \\ f_i(\mathbf{V}_{\boldsymbol{pa}(i)}^0) + \mathbf{U}_i & \mathbf{U}_i \sim \mathbb{P}_{\mathbf{U}_i}, & i \notin \mathcal{I} \end{cases}$$

*The exogenous space corresponding to $\mathcal{M}_0$, denoted by $\mathcal{U}_0$, includes the sensitive attributes and the non-sensitive parts of the exogenous variables of $\mathcal{M}$. This space called the semi-latent space, is constructed as $\mathcal{U}_0 = \mathcal{A} \times \mathcal{U}_{\mathcal{X}}$, where $\mathcal{U}_{\mathcal{X}}$ is the non-sensitive part of the exogenous space in $\mathcal{M}$.*

If we know the structural equations of $\mathcal{M}$, we can first map the CFDF to the exogenous space. In this space, the exogenous variables are assumed to be independent. Therefore, we can design a dissimilarity function for each variable separately and then combine them using product topology (§.2 [48]). Following this intuition, we introduce the bijective map $g : \mathcal{V} \to \mathcal{U}_0$ from the feature space to the semi-latent space, along with its inverse, defined as follows:

$$g_i(v) := \begin{cases} v_i & i \in \mathcal{I} \\ F_i(v) & i \notin \mathcal{I} \end{cases}, \quad g_i^{-1}(u) := \begin{cases} u_i & i \in \mathcal{I} \\ f_i(g_{\boldsymbol{pa}(i)}^{-1}(u)) + u_i & i \notin \mathcal{I} \end{cases} \tag{6}$$

If all sensitive attributes have no parents, the semi-latent space is equivalent to the exogenous space, and $g = F^{-1}$. The counterfactual with respect to $\mathcal{M}_0$ is denoted by $\textbf{CF}_0(v, \Delta)$. We can now present the following proposition to determine the shape of $d$.

**Proposition 1** *Let $\mathcal{M}$ be an ANM, with $g$ as its corresponding map to the semi-latent space 6 , and $P_\mathcal{X}(u)$ the projection of vector $u$ to the non-sensitive part $\mathcal{U}_\mathcal{X}$. Then:*

*(i) If $d_\mathcal{X}$ is a continuous dissimilarity function on diagonal $\mathcal{U}_\mathcal{X} \times \mathcal{U}_\mathcal{X}$, then the function $d$ defined as:*

$$d(v, v') = d_\mathcal{X}(P_\mathcal{X}(g(v)), P_\mathcal{X}(g(v'))) \tag{7}$$

*satisfies the definitions of a CFDF.*

*(ii) If $d : \mathcal{V} \times \mathcal{V} \to [0, \infty]$ satisfies the CFDF definition and the triangle inequality property, then $d$ can be represented as a dissimilarity function $d_\mathcal{X}$ dependent solely on the non-sensitive components $\mathcal{U}_\mathcal{X}$ i.e., $d(v, v') = d_\mathcal{X}(P_\mathcal{X}(g(v)), P_\mathcal{X}(g(v')))$.*

Since $d_\mathcal{X}$ is defined on independent coordinates, its relation to the components is less complex than the CFDF $d$. We assume the dissimilarity function $d_\mathcal{X}(x, x')$ is translation-invariant. Therefore, for simplicity, we assume $d_\mathcal{X}(x', x) = \|x' - x\|$. The dual of $\| \cdot \|$ is defined as $\|x\|_* = \sup_{x'}\{x^T x' \mid \|x'\| \leq 1\}$. Now we establish our assumptions about the SCM and its CFDF.

**Assumption 1**      *(i) $\mathcal{M}$ is an ANM with known structural equations and a semi-latent map $g$.*

     *(ii) The CFDF is defined as $d(v, v') = \|P_\mathcal{X}(g(v)) - P_\mathcal{X}(g(v'))\|$, where $\|.\|$ is a some norm.*

     *(iii) Cost function over $\mathcal{Z}$ has form $c((v, y), (v', y')) = d(v, v') + \infty \cdot |y - y'|$.*

     *(iv) The ambiguity set is defined as: $\mathbb{B}_\delta(\mathbb{P}) = \{\mathbb{Q} \in \mathcal{P}(\mathcal{V}) : W_{c,p}(\mathbb{P}, \mathbb{Q}) \leq \delta\}$, for $p \in [1, \infty)$.*

**Remark 1** *All results of this work apply to the **homogeneous dissimilarity function** (Def. 6), which includes a broad family of dissimilarity functions, such as norms.*

## 4 Causally Fair Distributionally Robust Optimization

To find out the impact of the CFDF in DRO problems, we first consider the dual form of the worst-case loss quantity, which simplifies the infinite-dimensional primal problem into a more tractable and computationally manageable form.

**Theorem 1 (Causally Fair Strong Duality)** *If Assumption 1 is satisfied, then for any reference probability distribution $\mathbb{P}$ and any function $\psi : \mathcal{V} \to \mathbb{R}$ that is both upper semi-continuous and $L_1$-integrable, the following duality holds:*

$$\sup_{\mathbb{Q} \in \mathbb{B}_\delta(\mathbb{P})} \left\{ \mathbb{E}_{v \sim \mathbb{Q}} [\psi(v)] \right\} = \inf_{\lambda \geq 0} \left\{ \lambda \delta^p + \mathbb{E}_{v \sim \mathbb{P}} \left[ \sup_{a \in \mathcal{A}} \psi_\lambda(\ddot{v}_a) \right] \right\}, \tag{8}$$

*where $\psi_\lambda(v)$ is defined as*

$$\psi_\lambda(v) := \sup_{\Delta \in \mathcal{X}} \left\{ \psi(\textbf{CF}_0(v, \Delta)) - \lambda^p d(v, \textbf{CF}_0(v, \Delta)) \right\}, \tag{9}$$

*and $\textbf{CF}_0$ is counterfactual regarding parent-free SCM $\mathcal{M}_0$.*

**Remark 2** *The intuition behind the above formula is as follows: In the case where all features are independent, let $v = (a, x)$. The CFDF should exhibit no difference between $(a, x)$ and $(a', x)$ for each $a, a' \in \mathcal{A}$. Consequently, the distance metric satisfies $d((a, x), (a', x')) = d_\mathcal{X}(x, x')$. Under this condition, the classical strong duality theorem (Eq. 5) provides the following relationship:*

$$\psi_\lambda(v) = \sup_{(a', x') \in \mathcal{V}} \left\{ \psi((a', x')) - \lambda d_\mathcal{X}^p(x, x') \right\} = \sup_{a \in \mathcal{A}} \left\{ \sup_{\Delta \in \mathcal{X}} \psi((a, x + \Delta)) - \lambda d_\mathcal{X}^p(x, x + \Delta) \right\}$$

*When we incorporate causal structure instead of coordinating $a$ and $x$, the two dimensions $\ddot{v}_a$ and $\textbf{CF}_0(v, \Delta)$ are replaced accordingly.*

In the DRO formulation, the worst-case loss is expressed in a dual form and can act as a regularizer for parameter learning. Explicitly solving the dual problem eliminates the need to compute the worst-case distribution, resulting in faster, more efficient learning algorithms [14, 16, 62, 73]. Before presenting the general theorem, the next two theorems show that, under mild conditions, the dual formula for specific loss functions in classification and regression problems can be explicitly solved.

**Theorem 2 (Higher Order Linear Loss)** *Given Assumptions 1, let $\mathcal{M}$ be a linear SCM and the loss function $\ell(z,\theta)^p$, where $\ell(z,\theta)$ is of the form $h(y - \langle\theta,v\rangle)$ or $h(y \cdot \langle\theta,v\rangle)$ for functions $h(t)$ such as $|t|$, $\max(0,t)$, $|t-\tau|$, or $\max(0,t-\tau)$ for some $\tau \geq 0$, and $p \in [1,\infty)$. Then the DRO problem 4 can be reduced to:*

$$\mathcal{R}_\delta(\mathbb{P}_N,\theta) = \begin{cases} \left(\mathcal{R}_\delta^{cf}(\mathbb{P}_N,\theta)^{\frac{1}{p}} + \delta\left\|P_\mathcal{X}(M^T\theta)\right\|_*\right)^p, & \operatorname{diam}(\mathcal{A}) < \infty \\[3mm] \left(\mathcal{R}(\mathbb{P}_N,\theta)^{\frac{1}{p}} + \delta\left\|P_\mathcal{X}(M^T\theta)\right\|_*\right)^p, & s.t. \quad P_\mathcal{A}(M^T\theta) = 0; \quad \operatorname{diam}(\mathcal{A}) = \infty \end{cases}$$

*where $M$ is the corresponding matrix for the linear map $g^{-1}$ (see Eq. 6).*

**Remark 3** *In real-world datasets, the sensitive part always satisfies $\operatorname{diam}(\mathcal{A}) < \infty$. According to the above theorem, $\mathcal{R}_\delta(\mathbb{P}_N,\theta) \geq \mathcal{R}_\delta^{cf}(\mathbb{P}_N,\theta)$. For practical applications, if the worst-case loss must not exceed a certain value, we can replace $\infty$ with some constant in the above theorem.*

**Example 2** *Here are specific examples of the above theorem. We offer a framework to study the equivalence between the worst-case loss in the DRO problem, with the cost function derived from the CFDF, and the regularization scheme for classification and regression problems.*

| Regression | Lower Partial Moments |
|---|---|
| $E_\mathbb{P}[\|\mathbf{Y} - \langle\theta,\mathbf{V}\rangle\|^p]$, $p \geq 1$ | $E_\mathbb{P}[(\mathbf{Y} - \langle\theta,\mathbf{V}\rangle - \tau)_+^p]$, $p \geq 1$, $\tau \in \mathbb{R}$ |
| Ridge Linear Regression | $\tau$-Insensitive Regression |
| $E_\mathbb{P}[(\mathbf{Y} + \langle\theta,\mathbf{V}\rangle)^2]$ | $E_\mathbb{P}[(\|\mathbf{Y} - \langle\theta,\mathbf{V}\rangle\| - \tau)_+^p]$, $p \geq 1$, $\tau \in \mathbb{R}$ |
| Hinge Loss Binary Classification | Support Vector Machine Classification |
| $E_\mathbb{P}[(1 - \mathbf{Y} \cdot \langle\theta,\mathbf{V}\rangle)_+^p]$, $p \geq 1$ | $E_\mathbb{P}[\|1 - \mathbf{Y} \cdot \langle\theta,\mathbf{V}\rangle\|^p]$, $p \geq 1$ |

The Thm. 2 can be extended to the non-linear regression loss function.

**Theorem 3 (Nonlinear Loss)** *Let assumptions 1 be satisfied, with $p = 1$, $\mathcal{M}$ linear with matrix $M$ corresponding to map $g^{-1}$, and a loss function $\ell(z,\theta)$ of the form $h(y - \langle\theta,v\rangle)$ for regression and $h(y \cdot \langle\theta,v\rangle)$ for classification, where $h$ has the following two properties:*

   *(i) $h$ is Lipschitz on $\mathbb{R}$ with $L_h$ constant, i.e., $|h(t_2) - h(t_1)| \leq L_h|t_2 - t_1|$, $\forall t_1, t_2 \in \mathbb{R}$.*

   *(ii) There exists sequence of $\{t_k\}_{k=1}^\infty$ goes to $\infty$ such that for each $t_0 \in \mathbb{R}$ we have $lim_{k\to\infty} \frac{|h(t_0 + t_k) - h(t_0)|}{|t_k|} = L_h$.*

*By the above assumption, DRO problem 4 can be reduced as:*

$$\mathcal{R}_\delta(\mathbb{P}_N,\theta) = \begin{cases} \mathcal{R}_\delta^{cf}(\mathbb{P}_N,\theta) + \delta L_h\left\|P_\mathcal{X}(M^T\theta)\right\|_*, & \operatorname{diam}(\mathcal{A}) < \infty \\[3mm] \mathcal{R}(\mathbb{P}_N,\theta) + \delta L_h\left\|P_\mathcal{X}(M^T\theta)\right\|_*, & s.t. \quad P_\mathcal{A}(M^T\theta) = 0; \quad \operatorname{diam}(\mathcal{A}) = \infty \end{cases}$$

**Example 3** *The following forms of the loss function satisfy the conditions of $h$ in Thm. 3:*

Now, the first-order estimation of the regularizer for non-linear SCM and loss function is ready to be stated.

**Theorem 4 (First-Order Estimation of DRO Regularizer)** *Assume $\mathcal{M}$ has structural equation $f$, which $f$ and loss function $\ell$ are both twice continuously differentiable respect to non-sensitive*

| Log-cosh Loss | Huber Loss |
|---|---|
| $h\colon t \mapsto \log(\cosh(t))$ | $h\colon t \mapsto \begin{cases} \frac{1}{2}t^2 & \text{if } |t| \leq 1, \\ |t| - \frac{1}{2} & \text{otherwise;} \end{cases}$ |
| Quantile Loss | Log-exponential Loss |
| $h\colon t \mapsto \begin{cases} \gamma t & \text{if } t \geq 0, \\ -t & \text{otherwise,} \end{cases}$ with $\gamma \in (0,1)$; | $h\colon t \mapsto \log(1 + \exp(-t))$ |
| Smooth Hinge Loss | Truncated Pinball Loss |
| $h\colon t \mapsto \begin{cases} 0 & \text{if } t \geq 1, \\ \frac{1}{2}(1-t)^2 & \text{if } 0 < t < 1, \\ \frac{1}{2} - t & \text{otherwise;} \end{cases}$ | $h\colon t \mapsto \begin{cases} 1-t & \text{if } t \leq 1, \\ \tau_1(t-1) & \text{if } 1 < t < \tau_2 + 1, \\ \tau_1\tau_2 & \text{otherwise,} \end{cases}$ where $\tau_1 \in [0,1], \tau_2 \geq 0$ are two given constants. |

*attributes,* $\operatorname{diam}(\mathcal{U}) < \infty$ *and c satisfies the assumption 1 with* $p \in [2, \infty]$. *The necessary condition for the existence of a finite DRO solution is that for each* $v \in \mathcal{V}$:

$$\sup_{a \in \mathcal{A}} \{\ell(\ddot{v}_a, y, \theta)\} < \infty.$$

*By these conditions, the worst-case loss quantity is equal to:*

$$\mathcal{R}_\delta(\mathbb{P}_N, \theta) = \underset{v \sim \mathbb{P}_N}{\mathbb{E}} \left[ \sup_{a \in \mathcal{A}} \ell(\ddot{v}_a, y, \theta) \right] + \delta \cdot \left( \underset{v \sim \mathbb{P}_N}{\mathbb{E}} \left[ \sup_{a \in \mathcal{A}} \|\nabla^{cr}\ell(\ddot{v}_a, y, \theta)\|_*^q \right] \right)^{1/q} + O(\delta^2), \quad (10)$$

*where the* $O(\delta^2)$ *term is uniform over all* $\theta \in \Theta$, *q is p's conjugate, and the gradient* $\nabla^{cr}\ell$ *equals to:*

$$\nabla^{cr}\ell(v, y, \theta) = \lim_{\Delta \to 0} \frac{\ell(\boldsymbol{CF}_0(v, \Delta), y, \theta) - \ell(v, y, \theta)}{\|\Delta\|}$$

*where* $\boldsymbol{CF}_0$ *is counterfactual regarding parent-free SCM* $\mathcal{M}_0$.

By applying Prop. 2 from Gao's work [28], the next proposition presents the relationship between classical adversarial optimization 3 and DRO for CFDF.

**Proposition 2 (Approximation by Robust Optimization)** *Suppose* $\mathcal{A}$ *is a finite set and let* $\{(v^i, y^i)\}_{i=1}^N$ *be observational data. Under Assumption 1, assume that for the loss function* $\ell$ *there exist constants* $L, M \geq 0$ *such that*

$$|\ell(v, y, \theta) - \ell(v', y, \theta)| < Ld^p(v, v') + M \quad \text{for all} \quad v, v' \in \mathcal{V} \text{ and } p \in [1, \infty).$$

*For an arbitrary* $K \in \mathbb{N}$, *consider the adversarial loss within the setting:*

$$\tilde{\mathcal{R}}_\delta^{adv}(\mathbb{P}_N) := \sup_{(w^{ik})_{i,k} \in \tilde{B}_\delta} \left\{ \frac{1}{NK} \sum_{i=1}^N \sum_{k=1}^K \sup_{a \in \mathcal{A}} \ell(\ddot{w}_a^{ik}, y_i, \theta) \right\},$$

*where the uncertainty set* $\tilde{B}_\delta$ *is defined as:*

$$\tilde{B}_\delta := \left\{ (w^{ik})_{i,k} : \frac{1}{N} \sum_{i=1}^N \sum_{k=1}^K d^p(v^i, w^{ik}) \leq \delta, \ w^{ik} \in \mathcal{V} \right\}.$$

*Then, the DRO can be approximated by adversarial optimization as follows:*

$$\tilde{\mathcal{R}}_\delta^{adv}(\mathbb{P}_N) \leq \mathcal{R}_\delta(\mathbb{P}_N) \leq \tilde{\mathcal{R}}_\delta^{adv}(\mathbb{P}_N) + \frac{LD + M}{NK},$$

*where D is independent of K.*

One of the main challenges in designing DRO for SCMs is that the CFDF depends on the causal structure. When the functional structure is unknown, it must be estimated from data. This empirical estimation impacts the DRO learning process. Therefore, it is crucial to control the uniform convergence error of the DRO problem between the true metric and distribution and the DRO estimated from the data. The following theorem guarantees learning from sample data, but certain assumptions need to be established first.

**Assumption 2**  *(i) $\mathcal{M}$ is an unknown ANM, $\operatorname{diam}(\mathcal{V}) < \infty$, and $\Theta$ is a compact subset of $\mathbb{R}^d$.*

*(ii) The loss function $\ell$ is uniformly bounded: there exists a positive constant $M$ such that $0 \leq \ell(z, \theta) \leq M$ for all $\theta \in \Theta$. Moreover, $\ell$ is Lipschitz with respect to the counterfactual in $\mathcal{M}_0$; that is, there exists a constant $L$ such that:*

$$|\ell(v, y, \theta) - \ell(\boldsymbol{CF}_0(v, \Delta), y, \theta)| \leq \|\ell\|_{\text{Lip}}).$$

*(iii) $\hat{d}$ is an estimation of the CFDF such that, with probability $1 - \epsilon$, there exists $M_d$ such that, at a rate of $N^{-\eta}$, the discrepancy is uniformly bounded by:*

$$\forall v, v' \in \mathcal{V}: \quad |d(v, v') - \hat{d}(v, v')| \leq M_d N^{-\eta}, \quad \text{for some} \quad \eta > 0.$$

The following theorem states that the efforts to estimate the metric or causal structures and the parameter $\hat{\theta}_N^{\text{dro}}$,

$$\hat{\theta}_N^{\text{dro}} := \inf_{\theta \in \Theta} \left\{ \sup_{\mathbb{Q}: W_{\hat{c}, p}(\mathbb{Q}, \mathbb{P}_N) \leq \delta} \; \mathbb{E}_{z \sim \mathbb{Q}}[\ell(z, \theta)] \right\}$$

Where $\hat{c}$ is the $\hat{d}$ corresponding cost on $\mathcal{Z}$, leading to the estimation of the true parameters of the DRO problem. To state our result, we need the Dudley entropy integral [61], which measures the complexity of the loss function class.

**Theorem 5 (Learning Finite Sample Guarantee)**  *With assumption 1 and 2, then for $\hat{\theta}_N^{\text{dro}}$ we have:*

$$\mathcal{R}_\delta(\mathbb{P}_*, \hat{\theta}_N^{\text{dro}}) - \inf_{\theta \in \Theta} \mathcal{R}_\delta(\mathbb{P}_*, \theta) \leq N^{-1/2} \left[ c_0 + c_1 \delta^{1-p} + c_2 \delta^{1-p} N^{-\eta+1/2} + c_3 \sqrt{\log(2/\epsilon)} \right],$$

*With probability at least $1 - 2\epsilon$. With $\mathfrak{C}(\mathcal{L})$ denoting the Dudley entropy integral for the function class $\{\ell(\cdot, \theta) : \theta \in \Theta\}$, the constants $c_0$, $c_1$ and $c_2$ are identified as follows:*

$$c_0 := 96\mathfrak{C}(\mathcal{L}), \; c_1 := 96L \cdot \operatorname{diam}(\mathcal{V})^p, \; c_2 := 2pL \cdot \operatorname{diam}(\mathcal{V})^{p-1} \cdot M_d, \; \text{and} \; c_3 := 2\sqrt{2} \times M.$$

The final theorem completes our framework, enabling us to perform DRO on real-world datasets without knowing the SCM structures while providing performance bounds.

## 4.1 Related Works

**Causally Fair Dissimilarity Function.** Various studies have addressed the specification and learning of individual fair metrics, such as [33, 68, 70, 47], but their construction based on causal structure and sensitive attributes remains unclear. Our work adopts and extends the concept of a causal fair metric, as discussed in the works [23, 24].

**DRO and Individual Fairness.** Previous works, such as [68, 70, 47], address the DRO problem with an individual fairness metric but are limited to linear SCMs and $p = 2$. These studies do not discuss the duality theorem or regularizers. Additionally, [42] studied DRO, but its connection to causality remains unclear.

**Strong Duality Theorem.** Various versions of the strong duality theorem have been explored in prior works. For instance, in [59, 46, 9, 14, 27, 28, 66], the cost function must be a metric or [74] has convex property. Additionally, in [72, 11, 58], the distance function $d$ must be positive-definite, meaning $d(v, v') = 0$ if and only if $v = v'$. However, these conditions are not met for CFDF, necessitating a new formulation of the duality theorem 1.

**DRO as Regularizer.** Previous works on using DRO as a regularizer, explicitly solved [60, 14, 16, 29] or through k-order estimation [5, 9, 6, 66, 27], only consider cases where the cost function is derived from a metric or a positive-definite dissimilarity function. Therefore, their theorems do not apply directly to our CFDF. We present new results in Theorems 2, 3, and 4 tailored for our cases.

**Finite Sample Guarantee.** Various works provide bounds on the performance of DRO solutions with finite samples [41, 25, 10, 12], but these do not apply to our CFDF due to previously mentioned reasons. The studies [68, 70, 47] offer performance bounds only for the case of linear SCMs with $p = 2$. Therefore, we present a general case in Theorem 5.

**Optimal Transport and Causality.** Recent works [39, 35, 13, 32, 21, 1, 2] on causal optimal transport focus on the causal structure of the transport map or plan, which differs from our problem. In our case, causality pertains to the transportation cost derived from SCMs.

# 5 Numerical Studies

In our numerical studies, we evaluate the impact of using causally fair DRO to mitigate individual unfairness, henceforth referred to as **CDRO**. We compare CDRO's performance against Empirical Risk Minimization (ERM), non-causal Adversarial Learning (AL) [44], and the Ross method [56]. Our experiments employ real-world datasets, namely the Adult [38] and COMPAS [65] datasets, pre-processed according to [19]. Additionally, we use a synthetic dataset for linear SCM (LIN) with formulations detailed in Appendix C.1. We first fit a linear structural equation model for both the

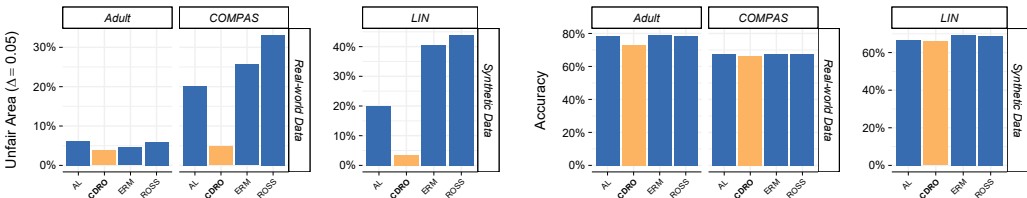

Figure 1: Displays the findings from our numerical experiment, assessing the performance of DRO across different models and datasets. (left) Bar plot showing the comparison of models based on the unfair area percentage (lower values are better) for $\Delta = .05$. (right) Bar plot comparing methods by prediction accuracy performance (higher values are better).

Adult and COMPAS datasets. Logistic regression is employed for classification, and performance is evaluated based on accuracy. Fairness is assessed using the Unfair Area Index (UAI), which is defined by the following equation:

$$\mathrm{U}_\Delta := \mathbb{P}\big(\{v \in \mathcal{V} : \quad \exists v' \in \mathcal{V} \quad \text{s.t.} \quad d(v, v') \leq \Delta \quad \wedge \quad h(v) \neq h(v')\}\big).$$

We evaluate UAI across different $\Delta$ values, specifically 0.05 and 0.01. Additionally, we calculate the UAI for scenarios where no sensitive attributes are considered, representing the percentage of non-robust data. Detailed computational experiment procedures are provided in Section C.1.

Our experiments, conducted using 100 different seeds, are summarized in Table 1. Figures 1, 2 and 3 illustrate that the CDRO method achieves a lower unfair area ($U_\Delta$) for $\Delta = .05$, and $\Delta = 0.01$ in all scenarios. Although CDRO shows slightly lower accuracy than ERM, this trade-off is a common observation in several studies [52]. Additional results can be found in § C.6.

# 6 Discussion and Limitations

Our study introduces a novel framework for causally fair DRO, integrating causal structures and sensitive attributes into the DRO paradigm. This framework is supported by several theoretical advancements, including a strong duality theorem, explicit regularizer formulation, first-order regularizer estimation, and finite sample guarantees with unknown SCMs, enhancing its efficiency and practicality for real-world applications. Our experimental results demonstrate its effectiveness in various settings, highlighting its potential for mitigating biases in machine learning models.

Despite the promising results, our study has several limitations that warrant further investigation. Firstly, the assumption of an additive noise model may not capture the complexity of all real-world causal relationships, posing challenges in computing additive interventions in general SCMs. Secondly, while Theorem 2 and Theorem 3 could be extended to more general cases, we omitted these extensions to avoid complexity. Lastly, further work is needed to explore the relationship between our method and causal optimal transport [13, 32].

## Acknowledgments

The authors thank the Max Planck Institute for Intelligent Systems, Tübingen AI Center, for supporting this project. Partial funding support was also provided by the Canada CIFAR AI Chair program.

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

# A   Supplementary Theoretical Details

**Notations.**   In this work random variables are denoted by bold letters (e.g., $\mathbf{V}$), their corresponding probability spaces by calligraphic letters (e.g., $\mathcal{V}$), and instances by normal letters (e.g., $v$). The space of probability measures on $\mathcal{V}$ is represented by $\mathcal{P}(\mathcal{V})$ and probability measures by blackboard bold letters (e.g., $\mathbb{P}$).

**Non-Sensitive Part.**   Let $F : \mathcal{U} \to \mathcal{V}$ be the reduced-form map of $\mathcal{M}$. The vector $v$ decomposes into sensitive and non-sensitive parts, $v = (a, x)$, and we have a corresponding decomposition in the exogenous space denoted by $u = (u_a, u_x)$ and $\mathcal{U}_\mathcal{A}, \mathcal{U}_\mathcal{X}$ are corresponding spaces. Using the ANM model, we can assume that both $\mathcal{V}$ and $\mathcal{U}$ are equivalent, and therefore, the non-sensitive feature space is the same as the non-sensitive part of the exogenous space. If $\mathbb{P}$ is a probability measure in $\mathcal{P}(\mathcal{V})$, then $(\mathbb{P})_\mathcal{X}$ refers to the marginal probability over the non-sensitive part. We also refer to $\mathbb{Q})_\mathcal{X}$ for marginal probability over the non-sensitive part of the exogenous space.

**Definition 3 (Push-forward Measure)** *Let $\mathbb{P} \in \mathcal{P}(\mathcal{V})$, $\mathbb{Q} \in \mathcal{P}(\mathcal{U})$ be two probability measures and $T : \mathcal{V} \to \mathcal{U}$ is map, the measure $\mathbb{Q}$ is called the push-forward of $\mathbb{P}$ through $T$ is denoted by $T_\#\mathbb{P}$ if:*

$$\mathbb{Q}(B) = \mathbb{P}(T^{-1}(B)), \quad \forall B \subset \mathcal{U}$$

**Definition 4 (Set of Couplings)** *The set $\Gamma(\mathbb{P}, \mathbb{Q})$ represents the couplings of probability distributions $\mathbb{P} \in \mathcal{P}(\mathcal{V})$, $\mathbb{Q} \in \mathcal{P}(\mathcal{U})$, comprising distributions over $\mathcal{V} \times \mathcal{U}$ with margins $\mathbb{P}$ and $\mathbb{Q}$. A measure $\pi$ belongs to $\Gamma(\mathbb{P}, \mathbb{Q})$ if and only if*

$$\pi(A \times \mathcal{U}) = \mathbb{P}(A) \quad and \quad \pi(\mathcal{V} \times B) = \mathbb{Q}(B) \quad \forall A \subset \mathcal{V}, B \subset \mathcal{U}$$

*By extension, a random pair $(X, Y) \sim \pi$, where $\pi \in \Gamma(\mathbb{P}, \mathbb{Q})$, will also be called a coupling of $\mathbb{P}$ and $\mathbb{Q}$.*

**Definition 5 (Diameter of a Set)** *Let $A$ be a set in a metric space with a distance function $d$. The diameter of $A$, denoted $diam(A)$, is defined as:*

$$diam(A) = \sup\{d(x, y) : x, y \in A\}$$

*where $\sup$ represents the supremum of the set of distances $d(x, y)$ for all pairs $(x, y)$ in $A$.*

**Definition 6 (Homogeneous dissimilarity function)** *Let $\Lambda$ be an extended-valued function $\Lambda \colon \mathcal{X} \to [0, \infty]$ on a real vector space $\mathcal{X}$ with absolutely homogeneous assumption i.e. $\Lambda(tx) = |t|\,\Lambda(x)$ for any $t \in \mathbb{R}$ and $z \in \mathcal{X}$. In addition, $\Lambda$ is proper it means there exists $x_0 \in \mathcal{X}$ such that $\Lambda(x_0) = 1$. The cost function $d : \mathcal{X} \times \mathcal{X} \to [0, \infty]$ is called Homogeneous dissimilarity function if is defined as $d(x', x) := \Lambda(x' - x)$ for any $x', x \in \mathcal{X}$.*

**Lemma 1** *If $\mathcal{M}$ is an additive noise model with mutually independent exogenous variables, then the parent-free sensitive $\mathcal{M}_0$ attribute model retains both of these properties. Moreover, the map-reduced form mapping of $\mathcal{M}_0$ is equivalent to $g^{-1}$, where $g$ represents the mapping to the semi-latent space.*

**Proof.**   First, $M_0$ is an additive noise model because its structure is derived from the initial equations of $\mathcal{M}$ by removing those equations related to the sensitive attributes and replacing the exogenous variable $\mathbf{U}_i$ by $\mathbf{V}_i$.

Regarding the mutual independence of the exogenous variables, in the original model $\mathcal{M}$, the variables $V_i$ for $i \in \mathcal{I}$ are not independent of $\mathbf{U}_j$ for $j \notin \mathcal{I}$ if they have parents. However, assuming a hard intervention for each instance of $V_i$ — where a do-action is executed for this intervention — it implies that the intervened variable $V_i$ can be considered independent from the other variables. Therefore, since we apply hard interventions to all sensitive variables, we can assume that $V_i$ for $i \in \mathcal{I}$ are mutually independent, and also that $V_i$ are independent of $\mathbf{U}_j$ for all $j \notin \mathcal{I}$.

Finally, by referencing equations 6, it is observable that the map-reduced form mapping of $\mathcal{M}_0$ is equivalent to the inverse of the map to the semi-latent space.

**Lemma 2** *Let $\mathcal{M}$ be an additive noise model with a mapping $g$ to semi-latent space $\mathcal{U}_0$. Assume $\mathcal{M}$ includes the sensitive attributes $\mathbf{A}$ and other non-sensitive attributes $\mathbf{X}$ that belong to the vector*

space $\mathcal{X}$. Consider $v = (a, x)$ as an instance in $\mathcal{M}$, and let $\Delta \in \mathcal{X}$ represent a shift intervention value. Then, the counterfactual corresponding to additive shit is obtained by:

$$P_{\mathcal{X}}(\boldsymbol{CF}(v, \Delta)) = P_{\mathcal{X}}(g^{-1}(g(v) + (0, \Delta))).$$

Moreover, if $a' \in \mathcal{A}$ represents another level of sensitive attributes, then the hard intervention concerning $\mathbf{A} := a'$ is achieved by:

$$\ddot{v}_{a'} = \boldsymbol{CF}(v, do(\mathbf{A}{:=}a')) = g^{-1}((a', P_{\mathcal{X}}(g(v)))).$$

**Proof.** In additive noise models, an additive intervention can be conceptualized as adding a value $\delta$ to the exogenous variables, while all structural equations remain unchanged. Consequently, during such an intervention, the reduced-form mapping $F_{\mathcal{M}^\triangle}$ of the intervened SCM remains unchanged. Therefore by definition of intervention, it follows that:

$$\mathbf{CF}(v, do(\mathbf{X}{+}{=}\Delta)) = F_{\mathcal{M}^\triangle}(F^{-1}(v) + (0, \Delta)) = F(F^{-1}(v) + (0, \Delta)).$$

Since $F$ and $g^{-1}$ are coincide in non-sensitive coordinates then $P_{\mathcal{X}}(\psi(F^{-1}(v) + (0, \Delta))) = P_{\mathcal{X}}(g^{-1}(g(v) + (0, \Delta)))$ and it completes the first part. In this case, we denote $F$ as $M$, which is an invertible matrix. Consequently, the counterfactual $\mathbf{CF}(v, do(\mathbf{X}{+}{=}\Delta))$ can be expressed as $v + M^{-1}(0, \Delta)$.

To prove the second part, when intervention is performed on sensitive attributes, $\mathcal{M}$ transforms into a parent-free sensitive attribute model where the sensitive attribute $a$ is replaced by $a'$. Given that the map-reduced form of the parent-free sensitive attribute model aligns with $g^{-1}$, and since $g$ and $F^{-1}$ coincide on the non-sensitive parts, the counterfactual can be expressed as follows:

$$\ddot{v}_{a'} = \mathbf{CF}(v, do(\mathbf{A}{:=}a')) = g^{-1}((a', P_{\mathcal{X}}(g(v)))).$$

# B    Proof Section

**Proof of Proposition 1.**

(i) If $d$ adheres to Eq. 7, it means that for each $v \in \mathcal{V}$, the mapping $g(v) = (a, x)$. For its counterfactual $\ddot{v}_a$, we have $g(\ddot{v}_{a'}) = (a', x)$. Using Eq. 7, we can express:

$$d(v, \ddot{v}_a) = d_{\mathcal{X}}(P_{\mathcal{X}}(g(v)), P_{\mathcal{X}}(g(\ddot{v}_a))) = d_{\mathcal{X}}(x, x) = 0$$

This demonstrates that $d$ retains the first property of Def. 1.

Additionally, since $d_{\mathcal{X}}$ is continuous, for each $x$ and any $\epsilon > 0$, there exists a $\delta > 0$ such that if $\|\Delta\| < \delta$, then $d_{\mathcal{X}}(x, x + \Delta) < \epsilon$. Referencing Lemma 2 and the formulation of $d$, it follows that for $\|\Delta\| < \delta$ and for each $a \in \mathcal{A}$:

$$d(v, \mathbf{CF}(v, \Delta)) = d(P_{\mathcal{X}}(g(v)), P_{\mathcal{X}}(g(\mathbf{CF}(v, \Delta)))) = d_{\mathcal{X}}(x, x + \Delta) < \epsilon$$

Thus, it satisfies property (ii) of the CFDF.

(ii) Let's consider a CFDF denoted as $d : \mathcal{V} \times \mathcal{V} \to \mathbb{R}$, with an embedding $\mathbf{g} : \mathcal{V} \to \mathcal{Q}$ that maps from the feature space to a semi-latent space. We define $d^*$ as the pull-back of $d$ onto $\mathcal{Q}$, $d^*(q_1, q_2) = d(g^{-1}(q_1), g^{-1}(q_2))$ where $d^*$ is a dissimilarity function, and we aim to clarify which properties it inherits from Def. 1. We utilize a decomposition of $\mathcal{Q}$ into $\mathcal{A} \times \mathcal{X}$, where $q = g^{-1}(v)$ and $v \in \mathcal{V}$, denoting $q$ as $(a, x)$. Property (i) of the CFDF ensures:

$$d(v, \ddot{v}_{a'}) = d^*((a, x), (a', x)) = 0 \quad \forall a' \in \mathcal{A}$$

This property confirms that $d^*$ is insensitive to changes in the sensitive part $\mathcal{A}$. To demonstrate, consider any two points $q_1 = (a_1, x_1)$ and $q_2 = (a_2, x_2)$, with an arbitrary $a_0 \in \mathcal{A}$. By triangle property of dissimilarity function $d$ it can be seen:

$$d^*((a_1, x_1), (a_2, x_2)) \leq d^*((a_1, x_1), (a_0, x_1)) + d^*((a_0, x_1), (a_2, x_2)) \implies$$
$$d^*((a_1, x_1), (a_2, x_2)) \leq d^*((a_0, x_1), (a_2, x_2))$$

Here, $d^*((s_1, x_1), (s_0, x_1))$ is zero due to property (i). Similarly, we can argue:

$$d^*((a_0, x_1), (a_2, x_2)) \leq d^*((a_0, x_1), (a_1, x_1)) + d^*((a_1, x_1), (a_2, x_2)) \implies$$
$$d^*((a_0, x_1), (a_2, x_2)) \leq d^*((a_1, x_1), (a_2, x_2))$$

This results in $d^*((a_1, x_1), (a_2, x_2)) = d^*((a_0, x_1), (a_2, x_2))$. With similar reasoning, we have:

$$d^*((a_1, x_1), (a_2, x_2)) = d^*((a_1, x_1), (a_0, x_2)) \implies d^*((a_1, x_1), (a_2, x_2)) = d^*((a_0, x_1), (a_0, x_2))$$

Hence, $d^*$ is invariant to the sensitive subspace. If $d_{\mathcal{X}}$ is the dissimilarity function induced by $d^*$ on $\mathcal{X}$, then $d^*((a_1, x_1), (a_2, x_2)) = d_{\mathcal{X}}(x_1, x_2)$. In accordance with Lemma 2, the second property of Def. 1 states that for each $\epsilon > 0$, there exists a $\delta$ such that if $|\Delta| < \delta$, then $d_{\mathcal{X}}(x, x + \Delta) < \epsilon$. This property demonstrates the continuity of $d_{\mathcal{X}}$ along the diagonal.

Finally, $d$ can be embedded in semi-latent space and described by another dissimilarity function on it that only depends on the non-sensitive part of exogenous space:

$$d(v, w) = d_{\mathcal{X}}(P_{\mathcal{X}}(g(v)), P_{\mathcal{X}}(g(w)))$$

This concludes the proof.

**Lemma 3 (Transformation by a Bijective Map)** *Let $g : \mathcal{V} \to \mathcal{U}$ be an invertible function and let the transportation cost function $c$ be constructed by $c(v, v') = d(g(v), g(v'))$ where $d$ is a metric on the space $\mathcal{U}$. For every $\mathbb{P}, \mathbb{Q} \in \mathcal{P}(\mathcal{V})$, the following equation holds:*

$$W_{c,p}(\mathbb{P}, \mathbb{Q}) = W_{d,p}(g_\# \mathbb{P}, g_\# \mathbb{Q})$$

*where $W_c$ and $W_d$ represent the Wasserstein distances with respect to the metrics $c$ and $d$, respectively.*

**Proof.** By the definition of the Wasserstein distance,

$$W_{c,p}(\mathbb{P}, \mathbb{Q}) = \inf_{\pi \in \Gamma(\mathbb{P}, \mathbb{Q})} \int_{\mathcal{V} \times \mathcal{V}} c^p(v, v') \, d\pi(v, v').$$

Substituting $c(v, v') = d(g(v), g(v'))$ and $u = g(v)$ gives:

$$W_{c,p}(\mathbb{P}, \mathbb{Q}) = \inf_{\pi \in \Gamma(\mathbb{P}, \mathbb{Q})} \int_{\mathcal{V} \times \mathcal{V}} d^p(g(v), g(v')) \, d\pi(v, v').$$

Consider a coupling $\pi$ of $\mathbb{P}$ and $\mathbb{Q}$. Define a measure $\tilde{\pi}$ on $\mathcal{U} \times \mathcal{U}$ by $\tilde{\pi}(A \times B) = \pi(g^{-1}(A) \times g^{-1}(B))$. $\tilde{\pi}$ is a coupling of $g_\# \mathbb{P}$ and $g_\# \mathbb{Q}$ because:

$$\tilde{\pi}(A \times \mathcal{U}) = \pi(g^{-1}(A) \times \mathcal{V}) = g_\# \mathbb{P}(A); \quad \tilde{\pi}(\mathcal{U} \times B) = \pi(\mathcal{V} \times g^{-1}(B)) = g_\# \mathbb{Q}(B).$$

Therefore, the $p$-Wasserstein distance for the push-forward measures is

$$\inf_{\pi \in \Gamma(\mathbb{P}, \mathbb{Q})} \int_{\mathcal{V} \times \mathcal{V}} d^p(g(v), g(v')) \, d\pi(v, v') = \inf_{\tilde{\pi} \in \Gamma(g_\# \mathbb{P}, g_\# \mathbb{Q})} \int_{\mathcal{U} \times \mathcal{U}} d^p(u, u') \, d\tilde{\pi}(u, u')$$
$$= W_{d,p}(g_\# \mathbb{P}, g_\# \mathbb{Q}).$$

Since $\tilde{\pi}$ arises from $\pi$ via $g$, and $g$ is invertible and measure-preserving in this context, the values in the integrals of the definitions of $W_{c,p}(\mathbb{P}, \mathbb{Q})$ and $W_{d,p}(g_\# \mathbb{P}, g_\# \mathbb{Q})$ match. Thus, we have shown that $W_{c,p}(\mathbb{P}, \mathbb{Q}) = W_{d,p}(g_\# \mathbb{P}, g_\# \mathbb{Q})$.

**Lemma 4 (Optimal Transportation Cost on Subspace)** *Let $\mathcal{U} \subseteq \mathbb{R}^n$ and suppose $\mathcal{U}$ is decomposed into two subspaces, $\mathcal{U} = (\mathcal{A}, \mathcal{X})$, where $\mathcal{A}$ corresponds to the subset of some coordinates and $\mathcal{X}$ to its complements. Let $P_{\mathcal{X}}$ denote the projection function onto the $\mathcal{X}$ space, i.e., $P_{\mathcal{X}}(u)$ projects $u \in \mathcal{U}$ onto $\mathcal{X}$ components. Define a cost function $c(u, u') = d(P_{\mathcal{X}}(u), P_{\mathcal{X}}(u'))$, where $d$ is a cost function on the space $\mathcal{X}$. Consider probability measures $\mathbb{P}, \mathbb{Q} \in \mathcal{P}(\mathcal{U})$, and define $\mathbb{P}_{\mathcal{X}} = P_{\mathcal{X} \#} \mathbb{P}$ and $\mathbb{Q}_{\mathcal{X}} = P_{\mathcal{X} \#} \mathbb{Q}$ as the pushforward measures of $\mathbb{P}$ and $\mathbb{Q}$ under the projection $P_{\mathcal{X}}$, respectively, placing them in $\mathcal{P}(\mathcal{X})$. Let $\pi_{\mathcal{X}}^*$ be the optimal transport plan concerning the Wasserstein distance $W_d(\mathbb{P}_{\mathcal{X}}, \mathbb{Q}_{\mathcal{X}})$. Then, any transport plan $\pi \in \mathcal{P}(\mathcal{U} \times \mathcal{U})$, whose marginal distribution over $\mathcal{X} \times \mathcal{X}$ equals $\pi_{\mathcal{X}}^*$, should also be an optimal solution for the Wasserstein distance $W_c(\mathbb{P}, \mathbb{Q})$ concerning the cost function $c$.*

**Proof.** Given any coupling $\pi \in \Gamma(\mathbb{P}, \mathbb{Q})$, we consider elements $u = (a, x)$ and $u' = (a', x')$ in $\mathcal{U} = \mathcal{A} \times \mathcal{X}$. The cost function $c$ is defined by $c((a, x), (a', x')) = d(x, x')$, where $d$ is a metric on

the space $\mathcal{X}$. By definition of optimal transport cost $W_c(\mathbb{P}, \mathbb{Q})$ we have:

$$\sup_{\pi \in \Gamma(\mathbb{P}, \mathbb{Q})} \left\{ \int_{\mathcal{U} \times \mathcal{U}} c((a, x), (a', x')) \, d\pi \right\} = \sup_{\pi \in \Gamma(\mathbb{P}, \mathbb{Q})} \left\{ \int_{\mathcal{U} \times \mathcal{U}} d(x, x') \, d\pi \right\} =$$

$$\sup_{\pi \in \Gamma(\mathbb{P}, \mathbb{Q})} \left\{ \int_{\mathcal{X} \times \mathcal{X}} \left( \int_{\mathcal{A} \times \mathcal{A}} d(x, x') \, d\pi((a, a') | \mathbf{X} = x, \mathbf{X}' = x') \right) d(\pi)_{\mathcal{X} \times \mathcal{X}} \right\} =$$

$$\sup_{\pi \in \Gamma(\mathbb{P}, \mathbb{Q})} \left\{ \int_{\mathcal{X} \times \mathcal{X}} d(x, x') d(\pi)_{\mathcal{X} \times \mathcal{X}} \right\} = \sup_{\pi \in \Gamma((\mathbb{P})_{\mathcal{X}}, (\mathbb{Q})_{\mathcal{X}})} \left\{ \int_{\mathcal{X} \times \mathcal{X}} d(x, x') \pi \right\}$$

where $(\mathbb{P})_{\mathcal{X}}$ and $(\pi)_{\mathcal{X} \times \mathcal{X}}$ is marginal distribution over $\mathcal{X}$ and $\mathcal{X} \times \mathcal{X}$ respectively. $\pi(. | \mathbf{X} = x, \mathbf{X}' = x')$ is conditional distribution of $\pi$ condition to the first and second $\mathcal{X}$ components equal to $x$ and $x'$.

We observe that this integral effectively only depends on the $\mathcal{X}$ component since the cost function $c$ does not involve $\mathcal{A}$. Hence, we reduce the expression to:

$$\sup_{\pi \in \Gamma((\mathbb{P})_{\mathcal{X}}, (\mathbb{Q})_{\mathcal{X}})} \left\{ \int_{\mathcal{X} \times \mathcal{X}} d(x, x') \pi \right\} \tag{11}$$

The Eq. 11 shows that the optima cost function of $W_c(\mathbb{P}, \mathbb{Q})$ equals to $W_c((\mathbb{P})_{\mathcal{X}}, (\mathbb{Q})_{\mathcal{X}})$. Therefore if $\pi_{\mathcal{X}}^*$ be the optimal transport plan for $\mathbb{P}_{\mathcal{X}}$ to $\mathbb{Q}_{\mathcal{X}}$ with respect to $d$ on $\mathcal{X}$, then any coupling $\pi$ in $\mathcal{U} \times \mathcal{U}$ that its marginal distribution $(\pi)_{\mathcal{X} \times \mathcal{X}}$ equals $\pi_{\mathcal{X}}^*$ is the solution of optimal transport. It results that the conditional distribution $\pi((.,.) | \mathbf{X} = x, \mathbf{X}' = x')$ could be any distribution. This completes the proof.

**Lemma 5** *Let $X$ and $A$ be sets, and let $f : X \times A \to \mathbb{R}$ be a function. Then*

$$\sup_{x \in X} \sup_{a \in A} f(x, a) = \sup_{a \in A} \sup_{x \in X} f(x, a).$$

**Proof.**    Define:
$$L = \sup_{x \in X} \sup_{a \in A} f(x, a) \quad \text{and} \quad R = \sup_{a \in A} \sup_{x \in X} f(x, a).$$

To show that $L = R$, we need to prove that $L \leq R$ and $R \leq L$. Consider any $x \in X$ and $a \in A$. By definition, $f(x, a) \leq \sup_{a \in A} f(x, a)$ for each fixed $x$. Therefore,

$$f(x, a) \leq \sup_{a \in A} f(x, a) \leq \sup_{x \in X} \sup_{a \in A} f(x, a) = R.$$

Since $f(x, a)$ was arbitrary, we have:

$$\sup_{a \in A} f(x, a) \leq R \quad \text{for all } x \in X,$$

and thus,

$$L = \sup_{x \in X} \sup_{a \in A} f(x, a) \leq R.$$

Similarly, for any fixed $a \in A$, $f(x, a) \leq \sup_{x \in X} f(x, a)$. Hence,

$$f(x, a) \leq \sup_{x \in X} f(x, a) \leq \sup_{a \in A} \sup_{x \in X} f(x, a) = L.$$

As before, since $f(x, a)$ was arbitrary, we conclude:

$$\sup_{x \in X} f(x, a) \leq L \quad \text{for all } a \in A,$$

and thus,

$$R = \sup_{a \in A} \sup_{x \in X} f(x, a) \leq L.$$

Since $L \leq R$ and $R \leq L$, it follows that $L = R$. Therefore, we have proven that:

$$\sup_{x \in X} \sup_{a \in A} f(x, a) = \sup_{a \in A} \sup_{x \in X} f(x, a).$$

This demonstrates the Principle of the Iterated Suprema.

## B.1 Proof of Theorem 1.

We prove the assertion in two steps: first, we assume that none of the sensitive attributes have parents, and second, we address and prove the general case. When all sensitive attributes do not have parents, in this case by definition2 semi-latent space equivalent with exogenous space and therefore $g = F^{-1}$. First, we show that the worst-case loss quantity can be decomposed into sensitive and non-sensitive components like as below equation:

$$\sup_{\mathbb{Q} \in \mathbb{B}_\delta(\mathbb{P})} \left\{ \mathbb{E}_{v \sim \mathbb{Q}} [\psi(v)] \right\} = \sup_{\mathbb{Q} \in \mathbb{B}_\delta((F_\#^{-1}\mathbb{P})_\mathcal{X})} \left\{ \mathbb{E}_{u_x \sim \mathbb{Q}} \left[ \sup_{u_a \in \mathcal{U}_\mathcal{A}} \{\psi(F((u_a, u_x)))\} \right] \right\}. \tag{12}$$

By the assumption, the CFDF has a form $d(v, v') = d_\mathcal{X}(P_\mathcal{X}(F^{-1}(v)), P_\mathcal{X}(F^{-1}(v')))$. By Def. 2 in a case that sensitive attributes have no parents then the semi-latent space coincides with exogenous space and the map between feature space and semi-latent space equals $g = F^{-1}$. Therefore in the following equations, we use $g$ instead of $F^{-1}$. Moreover since $g$ is invertible by Lemma 3, we can write:

$$\sup_{\mathbb{Q} \in \mathbb{B}_\delta(\mathbb{P})} \left\{ \mathbb{E}_{v' \sim \mathbb{Q}} [\psi(v')] \right\} = \sup_{\mathbb{Q} \in \mathbb{B}_\delta(g_\#\mathbb{P})} \left\{ \mathbb{E}_{u' \sim \mathbb{Q}} [\psi(F(u'))] \right\} =$$

$$\sup_{\pi \in \mathcal{P}(\mathcal{U} \times \mathcal{U})} \left\{ \mathbb{E}_{u' \sim \pi_2} [\psi(F(u'))] \, \Big| \, u \sim g_\#\mathbb{P}, \, \mathbb{E}_{(u,u') \sim \pi} [\tilde{d}_\mathcal{X}(u, u')] \leq \delta \right\} =$$

$$\sup_{\pi \in \mathcal{P}(\mathcal{U} \times \mathcal{U})} \left\{ \mathbb{E}_{u' \sim \pi_2} [\psi(F((u'_a, u'_x)))] \, \Big| \, u \sim g_\#\mathbb{P}, \, \mathbb{E}_{(u,u') \sim \pi} [d_\mathcal{X}(u_x, u'_x)] \leq \delta \right\} =$$

$$\sup_{\pi \in \mathcal{P}(\mathcal{U} \times \mathcal{U})} \left\{ \int_\mathcal{U} \psi(F((u'_a, u'_x))) \, d\pi_2(u') \, \Big| \, u \sim g_\#\mathbb{P}, \, \mathbb{E}_{(u_x, u'_x) \sim \pi_{\mathcal{X} \times \mathcal{X}}} [d_\mathcal{X}(u_x, u'_x)] \leq \delta \right\} = *,$$

where $\tilde{d}$ be a cost function on $\mathcal{U}$ defined as $\tilde{d}(u, u') = d_\mathcal{X}(P_\mathcal{X}(u), P_\mathcal{X}(u'))$, $\pi_2$ denotes the marginal distribution on second part and $\mathbb{B}_\delta(g_\#\mathbb{P}) = \{\mathbb{Q} \in \mathcal{P}(\mathcal{U}) : W_{\tilde{d}}(\mathbb{Q}, g_\#\mathbb{P}) \leq \delta\}$. Using the disintegration theorem ( [36] Chapter 3), the joint distribution $\pi_2$ can be decomposed into the product of the conditional distribution of $\mathcal{U}_\mathcal{A}$ given $\mathcal{U}_\mathcal{X}$ and the marginal distribution on $\mathcal{U}_\mathcal{X}$. Therefore we have

$$\int_\mathcal{U} \psi(F((u'_a, u'_x))) \, d\pi_2(u') = \int_{\mathcal{U}_\mathcal{X}} \left( \int_{\mathcal{U}_\mathcal{A}} \psi(F((u_a, u_x))) \, d\pi_2(u'_a | \mathbf{U}_\mathcal{X} = u'_x) \right) d_\mathcal{X}(\pi_2)_\mathcal{X}(u'_x),$$

where $(\pi_2)_\mathcal{X}$ is the marginal distribution of $\pi_2$ over the non-sensitive part and $\pi_2(u'_a | \mathbf{U}_\mathcal{X} = u'_x)$ is a conditional distribution of the sensitive part of exogenous space condition by $\mathbf{U}_\mathcal{X} = u'_x$. By disintegration formula, (*) can be rewritten as:

$$\sup_{\pi_2 \in \mathcal{P}(\mathcal{U})} \left\{ \int_\mathcal{U} \psi(F((u'_a, u'_x))) d\pi_2(u') \Big| \pi \in \mathcal{P}(\mathcal{U} \times \mathcal{U}), \pi_1 = g_\#\mathbb{P}, \, \mathbb{E}_{(u_x, u'_x) \sim (\pi)_{\mathcal{X} \times \mathcal{X}}} [d(u_x, u'_x)] \leq \delta \right\}$$

$$= \sup_{\pi_2 \in \mathcal{P}(\mathcal{U})} \left\{ \int_{\mathcal{U}_\mathcal{X}} \left( \int_{\mathcal{U}_\mathcal{A}} \psi(F(u_a, u_x)) \, d\pi_2(u'_a | \mathbf{U}_\mathcal{X} = u'_x) \right) d_\mathcal{X}(\pi_2)_\mathcal{X}(u'_x) \, \Big| \right.$$

$$\left. \pi \in \mathcal{P}(\mathcal{U} \times \mathcal{U}), \pi_1 = g_\#\mathbb{P}, \, \mathbb{E}_{(u_x, u'_x) \sim (\pi)_{\mathcal{X} \times \mathcal{X}}} [d(u_x, u'_x)] \leq \delta \right\}$$

$$= \sup_{(\pi_2)_\mathcal{X} \in \mathcal{P}(\mathcal{U}_\mathcal{X})} \left\{ \int_{\mathcal{U}_\mathcal{X}} \sup_{\pi_2(.|\mathbf{U}_\mathcal{X} = u'_x)} \left\{ \int_{\mathcal{U}_\mathcal{A}} \psi(F((u'_a, u'_x))) d\pi_2(u'_a | \mathbf{U}_\mathcal{X} = u'_x) \right\} d(\pi_2)_\mathcal{X}(u'_x) \, \Big| \right.$$

$$\left. \pi \in \mathcal{P}(\mathcal{U} \times \mathcal{U}), \pi_{2\mathcal{X}} = (g_\#\mathbb{P})_\mathcal{X}, \, \mathbb{E}_{(u_x, u'_x) \sim (\pi)_{\mathcal{X} \times \mathcal{X}}} [d_\mathcal{X}(u_x, u'_x)] \leq \delta \right\} \tag{13}$$

Since $d_\mathcal{X}$ depends only on the non-sensitive components, it follows from Lemma 4 that $\pi_2(.|\mathbf{U}_\mathcal{X} = u'_x)$ can achieve any distribution. Moreover, since it does not depend on the Wasserstein distance in each coupling, the marginal distribution of the sensitive attribute can be considered independent of the marginal distribution of the non-sensitive attributes. Therefore, the supremum over $\pi_2(.|\mathbf{U}_\mathcal{X} = u'_x)$ of integral equals the supremum of $\psi(F((u'_a, u'_x)))$ over all values of $u'_a$. Furthermore, the distribution

$\pi_2(u'_a|u'_x)$ does not influence the value of the Wasserstein distance. Based on these points, the last equation can be rewritten as:

$$\sup_{\pi_2 \in \mathcal{P}(\mathcal{U}_\mathcal{X})} \left\{ \int_{\mathcal{U}_\mathcal{X}} \sup_{u'_a \in \mathcal{U}_\mathcal{A}} \psi(F((u'_a, u'_x))) d\pi_2 \;\middle|\; \pi \in \mathcal{P}(\mathcal{U}_\mathcal{X} \times \mathcal{U}_\mathcal{X}), \right.$$

$$\left. \pi_1 = (g_\# \mathbb{P})_\mathcal{X}, \mathop{\mathbb{E}}_{(u_x, u'_x) \sim \pi}[d(u_x, u'_x)] \le \delta \right\} =$$

$$\sup_{\pi_2 \in \mathcal{P}(\mathcal{U}_\mathcal{X})} \left\{ \mathop{\mathbb{E}}_{u'_x \sim \pi_2} \left[ \sup_{u'_a \in \mathcal{U}_\mathcal{A}} \psi(F((u'_a, u'_x))) \right] \;\middle|\; \pi \in \mathcal{P}(\mathcal{U}_\mathcal{X} \times \mathcal{U}_\mathcal{X}), \right.$$

$$\left. \pi_1 = (g_\# \mathbb{P})_\mathcal{X}, \mathop{\mathbb{E}}_{(u_x, u'_x) \sim \pi}[d(u_x, u'_x)] \le \delta \right\} =$$

$$\sup_{\mathbb{Q} \in \mathbb{B}_\delta((g_\# \mathbb{P})_\mathcal{X})} \left\{ \mathop{\mathbb{E}}_{u'_x \sim \mathbb{Q}} \left[ \sup_{u'_a \in \mathcal{U}_\mathcal{A}} \left\{ \psi\left( F\left( (u'_a, u'_x) \right) \right) \right\} \right] \right\}.$$

The last equation concludes the proof of Eq. 12. Similarly, by altering the order of integration in Eq. 13, we arrive at the following equation:

$$\sup_{\mathbb{Q} \in \mathbb{B}_\delta(\mathbb{P})} \left\{ \mathop{\mathbb{E}}_{v \sim \mathbb{Q}}[\psi(v)] \right\} = \sup_{u_a \in \mathcal{U}_\mathcal{A}} \left\{ \sup_{\mathbb{Q} \in \mathbb{B}_\delta((g_\# \mathbb{P})_\mathcal{X})} \left\{ \mathop{\mathbb{E}}_{u_x \sim \mathbb{Q}} \left[ \psi\left( F\left( (u_a, u_x) \right) \right) \right] \right\} \right\}. \tag{14}$$

To proceed with the proof, we utilize the strong duality theorem. There are various kinds of duality theorems for DRO, but we apply the one proposed by Blanchet et al. [11].

**Strong duality [11].** Suppose the transportation cost $c : \mathcal{Z} \times \mathcal{Z} \to [0, \infty]$ satisfies $c(z, z) = 0$ for all $z \in \mathcal{Z}$ and lower semi-continuous. Then for any reference probability distribution $\mathbb{P}$ and upper semi-continuous $\psi : \mathcal{Z} \to \mathbb{R}$ satisfying $\mathop{\mathbb{E}}_\mathbb{P}[f(\mathbf{Z})] < \infty$, we have

$$\sup_{\mathbb{Q} \in \mathbb{B}_\delta(\mathbb{P})} \mathop{\mathbb{E}}_\mathbb{Q}[\psi(\mathbf{Z})] = \inf_{\lambda \ge 0} \lambda \delta + \mathop{\mathbb{E}}_\mathbb{P}[\psi_\lambda(\mathbf{Z})], \tag{15}$$

where $\psi_\lambda(z) := \sup_{z' \in \mathcal{Z}} \{ \psi(z') - \lambda c(z, z') \}$.

Based on the assumption about the CFDF, where only $d(v, \ddot{v}_{a'}) = 0$, it follows that $d(x, x') = 0$ only if $x = x'$. Therefore, we can apply the duality theorem to Eq. 12. According to the duality theorem, it can be expressed as follows:

$$\sup_{u_a \in \mathcal{U}_\mathcal{A}} \left\{ \sup_{\mathbb{Q} \in \mathbb{B}_\delta((g_\# \mathbb{P})_\mathcal{X})} \left\{ \mathop{\mathbb{E}}_{u_x \sim \mathbb{Q}} \left[ \psi\left( F\left( (u_a, u_x) \right) \right) \right] \right\} \right\} = \inf_{\lambda \ge 0} \left\{ \lambda \delta + \mathop{\mathbb{E}}_{u_x \sim (g_\# \mathbb{P})_\mathcal{X}} [\eta_\lambda(u_x)] \right\} \tag{16}$$

where $\eta_\lambda(u_x) = \sup_{u'_x \in \mathcal{U}_\mathcal{X}} \{ \sup_{u_a \in \mathcal{U}_\mathcal{A}} \{ \psi(F((u_a, u'_x))) \} - \lambda d_\mathcal{X}(u_x, u'_x) \}$. By using lemma 5 $\sup_{u'_x \in \mathcal{U}_\mathcal{X}} \{ \sup_{u_a \in \mathcal{U}_\mathcal{A}} \{ \psi(F((u_a, u'_x))) \} \} = \sup_{u_a \in \mathcal{U}_\mathcal{A}} \{ \sup_{u'_x \in \mathcal{U}_\mathcal{X}} \{ \psi(F((u_a, u'_x))) \} \}$.

Now, since sensitive attributes don't have parents then two spaces $\mathcal{U}_\mathcal{A}$ and $\mathcal{A}$ are equal. By applying Lemma 2, we can replace the above equation with hard and soft interventions as follows:

$$\eta_\lambda(u_x) = \sup_{a \in \mathcal{A}} \left\{ \sup_{u'_x \in \mathcal{U}_\mathcal{X}} \{ \psi(F((a, u'_x))) - \lambda d_\mathcal{X}(u_x, u'_x) \} \right\} =$$

$$\sup_{a \in \mathcal{A}} \left\{ \sup_{\Delta \in \mathcal{U}_\mathcal{X}} \{ \psi(F((a, u_x + \Delta))) - \lambda d_\mathcal{X}(u_x, u_x + \Delta) \} \right\} =$$

$$\sup_{a \in \mathcal{A}} \left\{ \sup_{\Delta \in \mathcal{U}_\mathcal{X}} \psi(F((a, u_x + \Delta))) - \lambda d_\mathcal{X}(P_\mathcal{X}((a, u_x)), P_\mathcal{X}((a, u_x + \Delta))) \right\} =$$

$$\sup_{a \in \mathcal{A}} \left\{ \sup_{\Delta \in \mathcal{U}_\mathcal{X}} \psi(F((a, u_x + \Delta))) - \lambda d_\mathcal{X}(P_\mathcal{X}(g^{-1}(g((a, u_x)))), P_\mathcal{X}(g^{-1}(g((a, u_x + \Delta))))) \right\} =$$

$$\sup_{a \in \mathcal{A}} \left\{ \sup_{\Delta \in \mathcal{U}_\mathcal{X}} \psi(\mathbf{CF}(\ddot{v}^a, \Delta)) - \lambda c(\ddot{v}^a, \mathbf{CF}(\ddot{v}^a, \Delta)) \right\} = \sup_{a \in \mathcal{A}} \left\{ \tilde{\psi}_\lambda(\ddot{v}^a) \right\} \tag{17}$$

Where $\tilde{\psi}_\lambda(\ddot{v}^a) := \sup_{\Delta \in \mathcal{U}_\mathcal{X}} \psi(\mathbf{CF}(\ddot{v}^a, \Delta)) - \lambda d(\ddot{v}^a, \mathbf{CF}(\ddot{v}^a, \Delta))$. The equations $F((a, u_x + \Delta)) = \mathbf{CF}(v, \Delta)$ and $F((a', u_x + \Delta)) = \mathbf{CF}(\ddot{v}^{a'}, \Delta)$ hold true according to Lemma 2. Finally, by substituting $\ddot{v}^a = \mathbf{CF}(v, a)$ into the equation, we prove the equation:

$$\sup_{\mathbb{Q} \in \mathbb{B}_\delta(\mathbb{P})} \left\{ \mathbb{E}_{v \sim \mathbb{Q}} [\psi(v)] \right\} = \inf_{\lambda \geq 0} \left\{ \lambda \delta^p + \mathbb{E}_{v \sim \mathbb{P}} \left[ \sup_{a \in \mathcal{A}} \psi_\lambda(\ddot{v}_a) \right] \right\},$$

where $\psi_\lambda(v)$ is defined as

$$\psi_\lambda(v) := \sup_{\Delta \in \mathcal{X}} \left\{ \psi(\mathbf{CF}_0(v, \Delta)) - \lambda^p d(v, \mathbf{CF}_0(v, \Delta)) \right\},$$

Since, in this case, $CF$ is equivalent to $\mathbf{CF}_0$, this completes the proof for case one.

Now consider the scenario where sensitive attributes have parents. Eq. 17 shows in strong duality computation it needs to compute function in intervened $\mathcal{M}$ concerning the sensitive attributes levels. In this case, instead of using the structural causal model $\mathcal{M}$, it is sufficient to employ the parent-free sensitive attribute SCM (Def. 2), $\mathcal{M}_0$. $\mathcal{M}_0$ aligns with the semi-latent space and is compatible with the representation form outlined in Proposition 1. By adopting this strategy, we transform $\mathcal{M}$ into a model where sensitive attributes do not have parents. The proof procedure for $\mathcal{M}_0$ remains the same as for $\mathcal{M}$ and completes the proof.

**Lemma 6** *Let $(\mathcal{Z}, c)$ be a space with cost function $c$, $\mathbb{P}_N$ an empirical probability measure based on observations $\{z_i\}_{i=1}^N$, and define $\mathbb{Q}$ as:*

$$\mathbb{Q} = \mathbb{P}_N - \frac{1}{N} \delta_{z_1} + \frac{1}{N} \delta_{z_1'}$$

*where $\delta_{z_1}$ and $\delta_{z_1'}$ are Dirac measures at $z_1$ and $z_1'$, respectively. Then, the p-Wasserstein distance between $\mathbb{P}_N$ and $\mathbb{Q}$ is given by:*

$$W_{c,p}(\mathbb{P}_N, \mathbb{Q}) = \left(\frac{1}{N}\right)^{\frac{1}{p}} c(z_1, z_1').$$

**Proof.**  The definition of the $p$-Wasserstein distance between two probability measures $\mathbb{P}_N$ and $\mathbb{Q}$ is:

$$W_{c,p}(\mathbb{P}_N, \mathbb{Q}) = \left( \inf_{\pi \in \Gamma(\mathbb{P}_N, \mathbb{Q})} \int_{\mathcal{Z} \times \mathcal{Z}} c(z, z')^p \, d\pi(z, z') \right)^{\frac{1}{p}},$$

where $\Gamma(\mathbb{P}_N, \mathbb{Q})$ represents the set of all couplings of $\mathbb{P}_N$ and $\mathbb{Q}$. Since $\mathbb{Q}$ is obtained by transferring a mass of $\frac{1}{N}$ from $z_1$ to $z_1'$, the optimal transport plan under the constraint that $\mathbb{P}_N$ and $\mathbb{Q}$ differ only at two points involves only moving the mass $\frac{1}{N}$ from $z_1$ to $z_1'$. The cost of this transportation is $c(z_1, z_1')^p$, and because the entire mass $\frac{1}{N}$ is being moved:

$$\int_{\mathcal{Z} \times \mathcal{Z}} c(z, z')^p \, d\pi(z, z') = c(z_1, z_1')^p \cdot \frac{1}{N}.$$

Therefore, substituting this into the formula for $W_p$, we obtain:

$$W_{c,p}(\mathbb{P}_N, \mathbb{Q}) = \left( c(z_1, z_1')^p \cdot \frac{1}{N} \right)^{\frac{1}{p}} = \left( \frac{1}{N} \right)^{\frac{1}{p}} c(z_1, z_1'),$$

thus proving the lemma.

**Lemma 7** *Assume that $f(a, x)$ is convex in $x$ for each fixed $a$ and continuous in both $a$ and $x$. Also, assume $f$ is uniformly continuous in $x$ across $a$. If $A$ is compact, then the function defined by*

$$g(x) = \sup_{a \in A} f(a, x)$$

*is convex and continuous in $x$.*

**Proof.** To show that $g(x)$ is convex, consider any $x_1, x_2$ in the domain and $\lambda \in [0, 1]$. By the definition of supremum and the convexity of $f(a, x)$ in $x$,

$$f(a, \lambda x_1 + (1 - \lambda)x_2) \leq \lambda f(a, x_1) + (1 - \lambda)f(a, x_2).$$

Taking the supremum over $a$ in $A$ on both sides, we get:

$$\sup_{a \in A} f(a, \lambda x_1 + (1 - \lambda)x_2) \leq \sup_{a \in A}(\lambda f(a, x_1) + (1 - \lambda)f(a, x_2)).$$

Using the properties of supremum,

$$\sup_{a \in A} f(a, \lambda x_1 + (1 - \lambda)x_2) \leq \lambda \sup_{a \in A} f(a, x_1) + (1 - \lambda) \sup_{a \in A} f(a, x_2).$$

Thus,

$$g(\lambda x_1 + (1 - \lambda)x_2) \leq \lambda g(x_1) + (1 - \lambda)g(x_2),$$

proving that $g(x)$ is convex.

To show continuity of $g(x)$ at a point $x_0$, consider any sequence $\{x_n\}$ converging to $x_0$. Since $f$ is uniformly continuous in $x$, given $\epsilon > 0$, there exists $\delta > 0$ such that for all $x, y$ with $|x - y| < \delta$,

$$|f(a, x) - f(a, y)| < \epsilon \quad \text{for all } a \in A.$$

Thus,

$$f(a, x_n) < f(a, x_0) + \epsilon \quad \text{and} \quad f(a, x_0) < f(a, x_n) + \epsilon \quad \text{for all } a \in A \text{ and } |x_n - x_0| < \delta.$$

Taking the supremum over $a$ in $A$,

$$g(x_n) \leq g(x_0) + \epsilon \quad \text{and} \quad g(x_0) \leq g(x_n) + \epsilon.$$

This implies

$$|g(x_n) - g(x_0)| \leq \epsilon,$$

establishing the continuity of $g(x)$ at $x_0$.

Hence, we conclude that $\sup_{a \in A} f(a, x)$ is convex and continuous in $x$.

### B.2 Proof of Theorem 2.

Let's consider the $\ell(v, y, \theta) = h(\mathbf{Y} - \langle \theta, \mathbf{V} \rangle)$ (or in abbreviation $\ell(y)$) or $h(\mathbf{Y} \cdot \langle \theta, \mathbf{V} \rangle)$ where $h : \mathbb{R} \to \mathbb{R}$ has one of the forms $|t|$, $\max(0, t)$, $|t - \tau|$, or $\max(0, t - \tau)$ for some $\tau \geq 0$

First consider the case diam $(\mathcal{A}) = \infty$. By property of CFDF for each Since the distance of $v$ by its twins $\ddot{v}_a$ is zero.Let $z \in \{z_i\}$ If $\mathbb{P}_N$ the empirical distribution by lemma 6 it can be seen for observation $Z = (v, y)$ the distribution

$$\mathbb{Q}_a = \mathbb{P}_N - \frac{1}{N}\delta_z + \frac{1}{N}\delta_{(\ddot{v}_a, y_1)} \implies W_{c,p}(\mathbb{Q}_a, \mathbb{P}_N) = 0 \implies \mathbb{Q}_a \in \mathbb{B}_\delta(\mathbb{P}_N).$$

This equation results that

$$\mathcal{R}_\delta(\mathbb{P}_N, \theta) \geq \sup_{a \in \mathcal{A}} \mathcal{R}_\delta(\mathbb{Q}_a, \theta) \geq \mathcal{R}(\mathbb{P}_N, \theta) - \frac{1}{N}\ell(v_1) + \frac{1}{N}\sup_{a \in \mathcal{A}}\ell(\ddot{v}_a)| \geq \frac{1}{N}\sup_{a \in \mathcal{A}}\ell(\ddot{v}_a)|$$

Let $u = (u_\mathcal{A}, u_\mathcal{X})$ such that $v = Mu$. By definition of hard intervention $\ddot{v}_a$ is obtained by the formula

$$\ddot{v}_a = (M - M_{\mathbf{pa}}) \times (u - (0, \ldots, \overbrace{a - u_\mathcal{A}}^{:=\alpha}, \ldots, 0)^T)$$

$$= M \times u - M \times (0, \ldots, \alpha, \ldots, 0) - M_{\mathbf{pa}} \times u + M_{\mathbf{pa}} \times (0, \ldots, \alpha, \ldots, 0)$$

$$= v - M_\mathcal{A} \times \alpha - C$$

where $M_{\mathbf{pa}}$ refers to the effect of parents of sensitive variables and $M_{\mathcal{A}}$ is the columns of matrix $M$ related to sensitive attributes that show the effects of sensitive attributes on non-sensitive variables. By substituting that last equation in the loss function we have:

$$\mathcal{R}_\delta(\mathbb{P}_N, \theta) \geq \frac{1}{N} \sup_{a \in \mathcal{A}} \ell(v - M_{\mathcal{A}} \times \alpha - C) \geq \frac{1}{N} \sup_{\alpha \to \infty} O(\theta^T M_{\mathcal{A}} \times \alpha)$$

where $B$ is some constant value. with the assumptions about loss function, all of them by choosing proper $z_1$, its behavior when $\alpha$ is large enough is linear so can be approximated by its input value. Now Since the diam $(\mathcal{A}) = \infty$ therefore the value of $\alpha$ goes to the infinity. Therefore to prevent the value of $\mathcal{R}_\delta(\mathbb{P}_N, \theta)$ it needs that the expression $\theta^T M_{\mathcal{A}} \times \alpha = 0$ in the other word the $P_{\mathcal{A}}(M^T \theta)$ needs to be zero. This condition implies that for all $a \in \mathcal{A}$ we have $\ell(v) = \ell(\ddot{v}_a)$. By using strong duality 8 we have:

$$\sup_{\mathbb{Q} \in \mathbb{B}_\delta(\mathbb{P})} \left\{ \mathop{\mathbb{E}}_{v \sim \mathbb{Q}} [\ell(v)] \right\} = \inf_{\lambda \geq 0} \left\{ \lambda \delta + \mathop{\mathbb{E}}_{v \sim \mathbb{P}} \left[ \sup_{a \in \mathcal{A}} \ell_\lambda(\ddot{v}_a) \right] \right\} = \inf_{\lambda \geq 0} \left\{ \lambda \delta + \mathop{\mathbb{E}}_{v \sim \mathbb{P}} [\ell_\lambda(v)] \right\}, \quad (18)$$

where

$$\begin{aligned}
\ell_\lambda(v) &= \sup_{\Delta \in \mathcal{X}} \left\{ \ell(\mathbf{CF}(v, \Delta)) - \lambda d(v, \mathbf{CF}(v, \Delta)) \right\} = \\
&\quad \sup_{\Delta \in \mathcal{X}} h(\theta^T M_0((u_a, u_x + \Delta))) - \lambda d_{\mathcal{X}}(u_x, u_x + \Delta) = \\
&\quad \sup_{\Delta \in \mathcal{X}} h(P_{\mathcal{X}}(M^T \theta)^T (u_x + \Delta)) - \lambda d_{\mathcal{X}}(u_x, u_x + \Delta) = \\
&\quad \sup_{\Delta \in \mathcal{X}} h(\langle \theta_0, u_x + \Delta \rangle) - \lambda \|\Delta\| = h_\lambda(u_x) \quad (19)
\end{aligned}$$

In the equation 19, $M_0$ is reduced-form mapping of the parent-free sensitive attribute $\mathcal{M}_0$. By the definition it is easy in both SCM, the effect of the sensitive attributes is equal therefore $P_{\mathcal{A}}(M_0^T \theta) = P_{\mathcal{A}}(M^T \theta) = 0$. Moreover since in $\mathcal{M}_0$ the structure of non-sensitive attribute has not changed then $P_{\mathcal{X}}(M_0^T \theta) = P_{\mathcal{X}}(M^T \theta)$. Finally, by substituting $\theta_0 = P_{\mathcal{X}}(M^T \theta)$, it can be seen that the problem of finding worst-case loss quantity converts to the regular problem in space $\mathcal{X}$. This problem was solved previously in works of [14, 58, 25, 66]. By using Theorem 3.2 and 3.3 and Proposition 4.1 and 4.2, of work by Chu et al. [14], we can write

$$\begin{aligned}
\inf_{\lambda \geq 0} \left\{ \lambda \delta + \mathop{\mathbb{E}}_{v \sim \mathbb{P}_N} [\ell_\lambda(v)] \right\} &= \inf_{\lambda \geq 0} \left\{ \lambda \delta + \mathop{\mathbb{E}}_{u_x \sim (g_\# \mathbb{P}_N)_{\mathcal{X}}} [h_\lambda(u_x)] \right\} = \\
\left( \mathcal{R}((g_\# \mathbb{P}_N)_{\mathcal{X}}, \theta_0)^{\frac{1}{p}} + \delta \|\theta_0\|_* \right)^p &= \left( \mathcal{R}(\mathbb{P}_N, \theta)^{\frac{1}{p}} + \delta \|P_{\mathcal{X}}(M^T \theta)\|_* \right)^p
\end{aligned}$$

The equality $\mathcal{R}((g_\# \mathbb{P}_N)_{\mathcal{X}}, \theta_0) = \mathcal{R}(\mathbb{P}_N, \theta)$ holds by definition and property $P_{\mathcal{A}}(M^T \theta) = 0$. The last equation completes the proof of the first case.

Now let's consider the case that diam $(\mathcal{A}) < \infty$.

**Case:** $p \in (1, \infty)$. Lets consider Eq. 16 it implies that:

$$\mathcal{R}_\delta(\mathbb{P}) = \inf_{\lambda \geq 0} \left\{ \lambda \delta + \mathop{\mathbb{E}}_{u_x \sim (g_\# \mathbb{P})_{\mathcal{X}}} \left[ \tilde{\ell}_\lambda(u_x) \right] \right\} \quad (20)$$

where, $\tilde{\ell}(u_x) = \sup_{u_a \in \mathcal{U}_{\mathcal{A}}} \{\ell(M(u_a, u'_x)\}$ and $\eta_\lambda(u_x) = \sup_{\Delta \in \mathcal{U}_{\mathcal{X}}} \tilde{\ell}(u_x + \Delta) - \lambda d_{\mathcal{X}}(u_x, u_x + \Delta)\}$. By assumption, the whole type of loss functions are form $h(\langle \theta, v \rangle)$ and convex and continuous. $h$ can be written by $M$ in the form $h(\langle \theta, M(u_a, u_x) \rangle)$. Then $\tilde{\ell}(u_x) = \sup_{u_a \in \mathcal{U}_{\mathcal{A}}} h(P_{\mathcal{A}}(M^T \theta) u_a + P_{\mathcal{X}}(M^T \theta) u_x)$. Since all forms of function are uniformly continuous concerning the $u_x$ then lemma 7 implies that the $\tilde{\ell}(u_x)$ is still continuous and convex. Theorem 6 and 7 of work [66] states that:

**Theorem. [66]** Let $\ell : \mathbb{R} \to \mathbb{R}$ be a non-negative, Lipschitz continuous and convex function. For an integer $p \in (1, \infty)$, suppose that for any $\mathbb{P} \in \mathcal{P}(\mathcal{X})$, and $\epsilon \geq 0$, we have:

$$\sup_{\mathbb{Q} \in \mathbb{B}_\delta(\mathbb{P})} \mathbb{E}_{x \sim \mathbb{Q}}[\ell^p(\theta^T x)] = \left( \left( \mathbb{E}_{x \sim \mathbb{P}}[\ell^p(\theta^T x)] \right)^{1/p} + \delta \|\theta\|_* \right)^p.$$

By applying above theorem in Eq. 20 it can be seen:

$$\mathcal{R}_\delta(\mathbb{P}) = \left( \left( \mathbb{E}_{u_x \sim (g_\# \mathbb{P})_\mathcal{X}} [\tilde{\ell}^p(u_x)] \right)^{1/p} + \delta \left\| P_\mathcal{X}(M^T \theta) \right\|_* \right)^p =$$

$$\left( \left( \mathbb{E}_{v \sim \mathbb{P}} [\sup_{a \in \mathcal{A}} \ell^p(\ddot{v}_a)] \right)^{1/p} + \delta \left\| P_\mathcal{X}(M^T \theta) \right\|_* \right)^p$$

**Case:** $p = 1$. To complete the proof we use Theorem 2 and Corollary 2 of Gao et al. work [27].

**Theorem [27]** If $\ell$ is Lipschitz $|\ell(x_1) - \ell(x_2)| \leq L \|x_k - x_0\|$ and satisfies tightness at infinity, i.e. for every $v_0$ there exists sequence $\{v_k\}_{k=1}^\infty \in \mathcal{V}$ such that $\|x_k - x_0\| \to \infty$ we have:

$$\lim_{\|x_k - x_0\| \to \infty} \frac{|\ell(x_k) - \ell(x_0)|}{\|x_k - x_0\|} = L \tag{21}$$

then we have $\mathcal{R}_\delta(\mathbb{P}) = \mathcal{R}(\mathbb{P}) + \delta . L$.

Now back to the Eq. 20. It is necessary to show that $\tilde{\ell}$ satisfies the Gao's theorem. Let $h(t)$ be one of the functions $|t|, (t - \tau)_+, (|t| - \tau)_+$. All of loss function can be written as form $h(y - \langle \theta, v \rangle)$ or $h(y . \langle \theta, v \rangle)$. It is easy to check that $h$ is Lipschitz with constant 1 and there exist $t_k$ such that for each $t_0$ we have $lim_{k \to \infty} \frac{|h(t_0 + t_k) - h(t_0)|}{|t_k|} = 1$.

To use this theorem for $\tilde{\ell}$, we need to prove that $\tilde{\ell}$ is Lipschitz and has a tightness condition at infinity. Since the $h$ is Lipschitz we know that:

$$\forall u_a \in \mathcal{U}_\mathcal{A}: \quad |h(P_\mathcal{A}(M^T \theta)u_a + P_\mathcal{X}(M^T \theta)u_x) - h(P_\mathcal{A}(M^T \theta)u_a + P_\mathcal{X}(M^T \theta)u_x')| \leq$$

$$|P_\mathcal{X}(M^T \theta)(u_x - u_x')| \leq \left\| P_\mathcal{X}(M^T \theta) \right\|_* \|u_x - u_x'\| \Rightarrow$$

$$\sup_{u_a \in \mathcal{U}_\mathcal{A}} |h(P_\mathcal{A}(M^T \theta)u_a + P_\mathcal{X}(M^T \theta)u_x) - h(P_\mathcal{A}(M^T \theta)u_a + P_\mathcal{X}(M^T \theta)u_x')|$$

$$\sup_{u_a \in \mathcal{U}_\mathcal{A}} |h(P_\mathcal{A}(M^T \theta)u_a + P_\mathcal{X}(M^T \theta)u_x) - h(P_\mathcal{A}(M^T \theta)u_a + P_\mathcal{X}(M^T \theta)u_x')| =$$

$$|\sup_{u_a \in \mathcal{U}_\mathcal{A}} h(P_\mathcal{A}(M^T \theta)u_a + P_\mathcal{X}(M^T \theta)u_x) - \sup_{u_a \in \mathcal{U}_\mathcal{A}} h(P_\mathcal{A}(M^T \theta)u_a + P_\mathcal{X}(M^T \theta)u_x')| =$$

$$|\tilde{\ell}(u_x) - \tilde{\ell}(u_x')| \leq \left\| P_\mathcal{X}(M^T \theta) \right\|_* \|u_x - u_x'\| \Rightarrow \quad \tilde{\ell} \text{ is Lipbschitz.}$$

To satisfy the condition of Gao's theorem, it remains to show for $u_x^0$ there exists sequence $\{u_x^k\}_{k=1}^\infty$ such that $\lim_{\|u_x^k - u_x^0\| \to \infty} \frac{|\tilde{\ell}(u_x^k) - \tilde{\ell}(u_x^0)|}{\|u_x^k - u_x^0\|} = L$ To prove it we consider that for function $h$ there exists sequence such that $lim_{k \to \infty} \frac{|h(t_0 + t_k) - h(t_0)|}{|t_k|} = 1$. consider the specific point $u_x^0$ and $u_a \in \mathcal{U}_\mathcal{A}$. Lets define $t_0 = P_\mathcal{A}(M^T \theta)u_a + P_\mathcal{X}(M^T \theta)u_x^0$. Then there exists $t_k$ that satisfies infinity tightness. Therefore there exist $\Delta_k \in \mathcal{U}_\mathcal{X}$ such that $P_\mathcal{X}(M^T \theta)u_x^k - P_\mathcal{X}(M^T \theta)u_x^0 = t_k$ therefore it can be written:

$$1 = lim_{k \to \infty} \frac{|h(t_0 + t_k) - h(t_0)|}{|t_k|} =$$

$$lim_{k \to \infty} \frac{|h(P_\mathcal{A}(M^T \theta)u_a + P_\mathcal{X}(M^T \theta)u_x^k) - h(P_\mathcal{A}(M^T \theta)u_a + P_\mathcal{X}(M^T \theta)u_x^0)|}{|P_\mathcal{X}(M^T \theta)u_x^k|} \Rightarrow$$

$$lim_{k \to \infty} \frac{|h(M^T \theta(u_a, u_x^k)) - h(M^T \theta(u_a, u_x^0))|}{\|u_x^k\|} = \left\| P_\mathcal{X}(M^T \theta) \right\|_* \Rightarrow$$

$$\sup_{u_a \in \mathcal{U}_\mathcal{A}} \left\{ lim_{k \to \infty} \frac{|h(M^T \theta(u_a, u_x^k)) - h(M^T \theta(u_a, u_x^0))|}{\|u_x^k\|} \right\} = \left\| P_\mathcal{X}(M^T \theta) \right\|_* \Rightarrow$$

$$lim_{k \to \infty} \sup_{u_a \in \mathcal{U}_\mathcal{A}} \left\{ \frac{|h(M^T \theta(u_a, u_x^k)) - h(M^T \theta(u_a, u_x^0))|}{\|u_x^k\|} \right\} = \left\| P_\mathcal{X}(M^T \theta) \right\|_* \Rightarrow$$

$$lim_{k \to \infty} \frac{|\tilde{\ell}(u_x^k) - \tilde{\ell}(u_x^0)|}{\|u_x^k\|} = \left\| P_\mathcal{X}(M^T \theta) \right\|_*$$

In the above equation, since we have uniform convergence, we can change the limit and supremum. The last equation shows that there exits sequence of $\{u_x^k\}_{k=1}^\infty$ satisfies tightness condition in $\infty$. By applying Gao's theorem we have:

$$\mathcal{R}_\delta(\mathbb{P}) = \mathbb{E}_{u_x \sim (g_\# \mathbb{P})_\mathcal{X}}[\tilde{\ell}(u_x)] + \delta \left\| P_\mathcal{X}(M^T \theta) \right\|_* = \mathbb{E}_{v \sim \mathbb{P}}[\sup_{a \in \mathcal{A}} \ell^p(\ddot{v}_a)] + \delta \left\| P_\mathcal{X}(M^T \theta) \right\|_*$$

The last equation completes the proof.

### B.3   Proof of Theorem 3.

The case diam $(\mathcal{A}) = \infty$ corresponds exactly to the first part of the proof of Theorem 2, with the only difference being that in that theorem we have $\|h\|_{\text{Lip}} = 1$, while in this case, we have $\|h\|_{\text{Lip}} = L_h$. Therefore we have the below equation:

$$\mathcal{R}_\delta(\mathbb{P}) = \mathcal{R}(\mathbb{P}) + L_h \left\| P_\mathcal{X}(M^T \theta) \right\|_*$$

Now consider the case diam $(\mathcal{A}) < \infty$. To prove our assertion, we use Eq. 16 and it implies that:

$$\mathcal{R}_\delta(\mathbb{P}) = \inf_{\lambda \geq 0} \left\{ \lambda \delta + \mathbb{E}_{u_x \sim (g_\# \mathbb{P})_\mathcal{X}} \left[ \tilde{\ell}_\lambda(u_x) \right] \right\}$$

where, $\tilde{\ell}(u_x) = \sup_{u_a \in \mathcal{U}_\mathcal{A}} \{\ell(M(u_a, u_x'))\}$ and $\eta_\lambda(u_x) = \sup_{\Delta \in \mathcal{U}_\mathcal{X}} \tilde{\ell}(u_x + \Delta) - \lambda d_\mathcal{X}(u_x, u_x + \Delta)\}$.

To compute the right side of the above equation, we use the theorem 3.2 of work [14] that states.

**Theorem [14].**   Let $\mathcal{Z}_N := \{z_1, \ldots, z_n\} \subset \mathcal{Z}$ be a given dataset and $\mathbb{P}_N$ be the corresponding empirical distribution. In addition, let $c(\cdot, \cdot)$ be a cost function on $\mathcal{Z} \times \mathcal{Z}$ and $\delta \in (0, \infty)$ be a scalar. Suppose the loss function $\ell : \mathcal{Z} \times \Theta \to \mathbb{R}$, where satisfies the following assumptions:

(A1)  $\ell$ is Lipschitz respect o the cost function $d$ at set $\mathcal{Z}_N$ with $L_\theta^{\mathcal{Z}_N} \in (0, \infty)$;

(A2)  for any $\epsilon \in (0, L_\theta^{\mathcal{Z}_N})$ and each $z_i \in \mathcal{Z}_N$, there exists $\tilde{z}_i \in \mathcal{Z}$ such that $\delta \leq d(\tilde{z}_i, z_i) < \infty$ and
$$\ell(\tilde{z}_i) - \ell(z_i) \geq (L_\theta^{\mathcal{Z}_N} - \epsilon) c(\tilde{z}_i, z_i).$$

Then we have:

$$\sup_{\mathbb{P}: W_{d,1}(\mathbb{Q}, \mathbb{P}_N) \leq \delta} \mathbb{E}_\mathbb{Q}[\ell(\mathbf{Z}, \theta)] = \mathbb{E}_{\mathbb{P}_N}[\ell(\mathbf{Z}, \theta)] + L_\theta^{\mathcal{Z}_N} \delta.$$

To use the above theorem we need that $\tilde{\ell}$ satisfies conditions (A1) and (A2).

**A1.**   To prove the Lipschitz condition it can be seen:

$$\forall u_a \in \mathcal{U}_\mathcal{X} : \quad |h(y - \theta^T M(u_a, u_x)) - h(y - \theta^T M(u_a, u_x'))|$$
$$\leq L_h |P_\mathcal{X}(M^T \theta)(u_a - u_a')| \leq L_h \left\| P_\mathcal{X}(M^T \theta) \right\|_* \|u_a - u_a'\| \Rightarrow$$
$$\sup_{u_a \in \mathcal{U}_\mathcal{X}} |h(y - \theta^T M(u_a, u_x)) - h(y - \theta^T M(u_a, u_x'))| =$$
$$|\tilde{\ell}(u_a) - \tilde{\ell}(u_x')| \leq L_h \left\| P_\mathcal{X}(M^T \theta) \right\|_* \|u_x - u_x'\|$$

The case $h(y. \langle \theta, y \rangle)$ is similar so we omit it.

**A2.**   To check that $\tilde{\ell}$ satisfies (A2) condition we show that there exists sequence $\{\Delta^k\}$ such that the $\|\Delta_k\| \to \infty$ for every $v \in \mathcal{V}$ and sequence $v_k = \mathbf{CF}(v, \Delta_k)$ and we have:

$$lim_{k \to \infty} \frac{|h(y - \langle \theta, v_k \rangle) - h(y - \langle \theta, v \rangle)|}{d(v_k, v)} = L_h \cdot \left\| P_\mathcal{X}(M^T \theta) \right\|_* \tag{22}$$

By assumption about $h$ we have: For each $t_0 \in \mathbb{R}$ there exists sequence of $\{t_k\}_{k=1}^\infty$ goes to $\infty$ then $lim_{k \to \infty} \frac{|h(t_0 + t_k) - h(t_0)|}{|t_k|} = L_h$. By changing variable $v = M(u_a, u_x)$. Let $t_0 =$

$y - \theta^T M(u_a, u_x)$ and $\Delta_k \in \mathcal{X}$ such that $P_{\mathcal{X}}(M^T\theta)\Delta_k = t_k$ it is clear $\Delta_k$ exist. No if we define $v_k = \mathbf{CF}(v, \Delta_k)$ we have:

$$L_h = lim_{k\to\infty} \frac{|h(t_0 + t_k) - h(t_0)|}{|t_k|} =$$

$$lim_{k\to\infty} \frac{|h(y - \theta^T M(u_a, u_x) + P_{\mathcal{X}}(M^T\theta)\Delta_k) - h(y - \theta^T M(u_a, u_x))|}{|P_{\mathcal{X}}(M^T\theta)\Delta_k|} =$$

$$lim_{k\to\infty} \frac{|h(y - \theta^T M(u_a, u_x + \Delta_k)) - h(y - \theta^T M(u_a, u_x))|}{|P_{\mathcal{X}}(M^T\theta)\Delta_k|} =$$

$$lim_{k\to\infty} \frac{|h(y - \theta^T v_k) - h(y - \theta^T v)|}{\|P_{\mathcal{X}}(M^T\theta)\|_* \|\Delta_k\|} = lim_{k\to\infty} \frac{|h(y - \langle\theta, v_k\rangle) - h(y - \langle\theta, v\rangle)|}{\|P_{\mathcal{X}}(M^T\theta)\|_* d(v_k, v)}$$

$$\implies lim_{k\to\infty} \frac{|h(y - \langle\theta, v_k\rangle) - h(y - \langle\theta, v\rangle)|}{d(v_k, v)} = \|P_{\mathcal{X}}(M^T\theta)\|_* .L_h$$

The Last equation is valid because, by Holder inequality, there exists $\Delta$ such that for all $\lambda\Delta$ the Holder inequality converts to equality. Now it is sufficient that find proper $\lambda$ such that $\lambda_k = t_k / \|P_{\mathcal{X}}(M^T\theta)\Delta\|_*$ so by define $\Delta_k = \lambda_k\Delta$ we find sequence that holds the assertion.

The case of $h(y. \langle\theta, v\rangle)$ is similar. By discussion of the first part, we can find $\Delta_k$. Now since we have binary classification, so $y \in \{-1, 1\}$ we define $\tilde{\Delta}_k = \text{sign}(y)\Delta_k$ therefore for such $\Delta_k$, we have $y.P_{\mathcal{X}}(M^T\theta)\tilde{\Delta}_k) = t_k$. By assumption, it can be written as:

$$L_h = lim_{k\to\infty} \frac{|h(t_0 + t_k) - h(t_0)|}{|t_k|} =$$

$$lim_{k\to\infty} \frac{|h(y.\theta^T M(u_a, u_x) + y.P_{\mathcal{X}}(M^T\theta)\tilde{\Delta}_k) - h(y.\theta^T M(u_a, u_x))|}{|P_{\mathcal{X}}(M^T\theta)\tilde{\Delta}_k|} =$$

$$lim_{k\to\infty} \frac{|h(y.\theta^T M(u_a, u_x) + y.P_{\mathcal{X}}(M^T\theta)\Delta_k) - h(y.\theta^T M(u_a, u_x))|}{|P_{\mathcal{X}}(M^T\theta)\Delta_k|} =$$

$$lim_{k\to\infty} \frac{|h(y.\theta^T M(u_a, u_x + \Delta_k)) - h(y.\theta^T M(u_a, u_x))|}{|P_{\mathcal{X}}(M^T\theta)\Delta_k|} =$$

$$lim_{k\to\infty} \frac{|h(y.\theta^T v_k) - h(y.\theta^T v)|}{\|P_{\mathcal{X}}(M^T\theta)\|_* \|\Delta_k\|} = lim_{k\to\infty} \frac{|h(y. \langle\theta, v_k\rangle) - h(y - \langle\theta, v\rangle)|}{\|P_{\mathcal{X}}(M^T\theta)\|_* d(v_k, v)}$$

$$\implies lim_{k\to\infty} \frac{|h(y - \langle\theta, v_k\rangle) - h(y - \langle\theta, v\rangle)|}{d(v_k, v)} = \|P_{\mathcal{X}}(M^T\theta)\|_* .L_h$$

Let $\ell(v) = h(y - P_{\mathcal{X}}(M^T\theta)u_x - P_{\mathcal{A}}(M^T\theta)u_a)$. By assumption $h$ is Lipschitz so it $\ell$ is Lipschitz concerning each $u_x$ and $u_a$ and $\mathcal{U}_{\mathcal{A}}$ is bounded so it is compact. Then these properties imply uniformly continuous so we have:

$$\sup_{u_a \in \mathcal{U}_{\mathcal{A}}} lim_{k\to\infty} \frac{|h(y - \langle\theta, v_k\rangle) - h(y - \langle\theta, v\rangle)|}{d(v_k, v)} =$$

$$lim_{k\to\infty} \sup_{u_a \in \mathcal{U}_{\mathcal{A}}} \frac{|h(y - \langle\theta, v_k\rangle) - h(y - \langle\theta, v\rangle)|}{d(v_k, v)} = lim_{k\to\infty} \frac{|\tilde{\ell}(u_x) - \tilde{\ell}(\Delta_k)|}{\|u_x - \Delta_k\|} = \|P_{\mathcal{X}}(M^T\theta)\|_* .L_h$$

The last equation satisfies the $(A_2)$ condition because since $lim_{k\to\infty} \frac{|\tilde{\ell}(u_x) - \tilde{\ell}(\Delta_k)|}{\|u_x - \Delta_k\|} = \|P_{\mathcal{X}}(M^T\theta)\|_* .L_h$ in other hand we have $|\tilde{\ell}(u_a) - \tilde{\ell}(u'_x)| \le L_h \|P_{\mathcal{X}}(M^T\theta)\|_* \|u_x - u'_x\|$, then for each $\epsilon$ there exist $\Delta_k$ such that $|\tilde{\ell}(u_x) - \tilde{\ell}(\Delta_k)| > (L_h \|P_{\mathcal{X}}(M^T\theta)\|_* \|u_x - \Delta_k\| - \epsilon) \|u_x - \Delta_k\|$

Now we can use Chu's Theorem and it implies that: $\mathcal{R}_\delta(\mathbb{P}) = \mathcal{R}^{cf}(\mathbb{P}) + L_h \|P_{\mathcal{X}}(M^T\theta)\|_*$. The classification case is the same and it completes the proof.

## B.4 Proof of Theorem 4.

Let's prove the necessary condition. By first part proof of theorem 2, we have

$$\mathcal{R}_\delta(\mathbb{P}_N, \theta) \geq \frac{1}{N} \sup_{a \in \mathcal{A}} \ell(\ddot{v}_a, y, \theta).$$

Therefore, for a finite solution to exist for the DRO problem, it is necessary that:

$$\sup_{a \in \mathcal{A}} \ell(\ddot{v}_a, y, \theta) < \infty.$$

To prove Eq. 10, we use again some parts of the proof of strong duality theorem1 and idea of proof of theorem 9.1 of Garcia's work [29]. It states:

$$\mathcal{R}_\delta(\mathbb{P}_N) = \sup_{\mathbb{Q} \in \mathbb{B}_\delta(\mathbb{P}_N)} \left\{ \mathbb{E}_{v \sim \mathbb{Q}} [\ell(v', y, \theta)] \right\} = \sup_{\substack{\mathbb{Q}_x \in \mathbb{B}_\delta((g_\# \mathbb{P}_N)_\mathcal{X}) \\ \mathbb{Q}_a \in \mathcal{P}(\mathcal{A})}} \left\{ \mathbb{E}_{\substack{u'_x \sim \mathbb{Q}_x \\ u'_a \sim \mathbb{Q}_a}} \left[ \ell\left(g^{-1}\left((u'_a, u'_x)\right), y, \theta,\right) \right] \right\}$$

where by discussion in lemma 4, we can suppose that $\mathbb{Q}_x$ and $\mathbb{Q}_a$ are independent of each other. For simplicity, we define $J(u_x, u_a, \theta) = \ell(g^{-1}((u_a, u_x)), y, \theta)$. By assumption, since the $f$ is twice differentiable, then $(I - f)^{-1}$ is also twice differentiable. Because the function $g$ is obtained by $I - f$ by removing the functional structure of sensitive attributes and is a set identity function instead of them, there are two functions $g$ and $g^{-1}$. These results show that combination $\ell(g^{-1})$ is also twice differentiable, so the gradient of $J$ exists concerning the $u_x$. By using Taylor's expansion theorem if $f : \mathbb{R} \to \mathbb{R}$ is the function that has gradient then the first order estimation of $f$ equals:

$$f(x + h) = f(x) + \nabla f(x)^\top h + \int_0^1 \left(\nabla f(x + th) - \nabla f(x)\right)^\top h \, dt.$$

Let $u_x \sim (g_\# \mathbb{P})_\mathcal{X}$ by writing Taylor's expansion around $u_x$ we have:

$$\mathbb{E}\left[J(u'_x, u'_a, \theta)\right] = \mathbb{E}\left[J(u_x, u'_a, \theta) + \nabla_x J(u_x, u'_a, \theta) \cdot (u'_x - u_x)\right.$$
$$\left. + \int_0^1 \{\nabla_x J(u_x + \lambda(u'_x - u_x), u'_a, \theta) - \nabla_x J(u_x, u'_a, \theta)\} \cdot (u'_x - u_x) d\lambda\right].$$

Since the diam$(\mathcal{U}) < \infty$ we can suppose that the space $\mathcal{U}$ is compact. By assumption $J$ is twice differentiable, so $\nabla_x J(., u_a, \theta)$ is Lipschitz with constant $\|\nabla_x J(., u_a, \theta)\|_{\text{Lip}}$ that there exist $L < \infty$ such that $\|\nabla_x J(., u_a, \theta)\|_{\text{Lip}} \leq L$. By these assumptions, it can be written:

$$\mathbb{E}\left[\int_0^1 \{\nabla_x J(u_x + \lambda(u'_x - u_x), u'_a, \theta) - \nabla_x J(u_x, u'_a, \theta)\} \cdot (u'_x - u_x) d\lambda\right] \leq$$

$$\mathbb{E}\left[\int_0^1 \|\nabla_x J(u_x + \lambda(u'_x - u_x), u'_a, \theta) - \nabla_x J(u_x, u'_a, \theta)\| \|u'_x - u_x\|_e^2 d\lambda\right] =$$

$$\mathbb{E}\left[\int_0^1 \|\nabla_x J(., u'_a, \theta)\|_{\text{Lip}} \|u'_x - u_x\|_e^2 d\lambda\right] = \mathbb{E}\left[\frac{1}{2} \|\nabla_x J(., u'_a, \theta)\|_{\text{Lip}} \|u'_x - u_x\|_e^2\right] \leq$$

$$\frac{C}{2} \|\nabla_x J(., u'_a, \theta)\|_{\text{Lip}} \mathbb{E}\left[\|u'_x - u_x\|^2\right] \leq \frac{CL}{2} \mathbb{E}\left[\|u'_x - u_x\|^2\right] \leq O(\delta^2),$$

where $C$ is a constant that arises from the equivalence of norms in $\mathbb{R}^d$, it means that there exists $C \|.\|_e \leq C \|.\|$. The inequality $\mathbb{E}\left[\|u'_x - u_x\|^2\right] \leq \delta^2$ is valid because by definition $\mathbb{Q}_x \in \mathbb{B}_\delta((g_\# \mathbb{P}_N)_\mathcal{X})$ and the cost function in the space $\mathcal{U}_\mathcal{X}$ is expressed by the $\|u'_x - u_x\|$ so by definition of $\mathbb{B}_\delta((g_\# \mathbb{P}_N)_\mathcal{X})$, for $p \geq 2$ by applying Jensen's inequality we have

$$\mathbb{Q}_x \in \mathbb{B}_\delta((g_\# \mathbb{P}_N)_\mathcal{X}) \Rightarrow \mathbb{E}_{\mathbb{Q}_x}\left[\|u'_x - u_x\|^p\right]^{\frac{1}{p}} \leq \delta \Rightarrow$$

$$\mathbb{E}_{\mathbb{Q}_x}\left[\|u'_x - u_x\|^2\right] \leq \mathbb{E}_{\mathbb{Q}_x}\left[\|u'_x - u_x\|^p\right]^{\frac{2}{p}} \leq \delta^2.$$

Since by assumption $\nabla J_x$ is uniformly Lipschitz for different value of $\theta$ and $u_a$ therefore we have:

$$\mathcal{R}_\delta(\mathbb{P}_N) = \sup_{\substack{\mathbb{Q}_x \in \mathbb{B}_\delta((g_\# \mathbb{P}_N)_\mathcal{X}) \\ \mathbb{Q}_a \in \mathcal{P}(\mathcal{A})}} \left\{ \mathbb{E}\left[J(u_x, u'_a, \theta) + \nabla_x J(u_x, u'_a, \theta) \cdot (u'_x - u_x)\right] \right\} + O(\delta^2),$$

for $O(\delta^2)$ independent of $\theta$ and $u_a$. The first expression of the above equation has the simple form:

$$\sup_{\mathbb{Q}_a \in \mathcal{P}(\mathcal{U}_\mathcal{A})} \left\{ \mathbb{E}_{\substack{u_x \sim (g_\# \mathbb{P}_N)_\mathcal{X} \\ u'_a \sim \mathbb{Q}_a}} [J(u_x, u'_a, \theta)] \right\} = \mathbb{E}_{u_x \sim (g_\# \mathbb{P}_N)_\mathcal{X}} \left[ \sup_{\mathbb{Q}_a \in \mathcal{P}(\mathcal{U}_\mathcal{A})} \left\{ \mathbb{E}_{u'_a \sim \mathbb{Q}_a} [J(u_x, u'_a, \theta)] \right\} \right] =$$

$$\mathbb{E}_{u_x \sim (g_\# \mathbb{P}_N)_\mathcal{X}} \left[ \sup_{u'_a \in \mathcal{U}_\mathcal{A}} \{ J(u_x, u'_a, \theta) \} \right] = \mathbb{E}_{u_x \sim (g_\# \mathbb{P}_N)_\mathcal{X}} \left[ \sup_{u'_a \in \mathcal{U}_\mathcal{A}} \{ \ell(g^{-1}((u_x, u'_a)), y, \theta) \} \right] =$$

$$\mathbb{E}_{v \sim \mathbb{P}_N} \left[ \sup_{a \in \mathcal{A}} \ell(\ddot{v}_a, y, \theta) \right]$$

The only term in the above equation that still depends on the $\mathbb{Q}$ is that term $\nabla_x J(u_x, u'_a, \theta) \cdot (u'_x - u_x)$. To remove this term we use extended Hölder inequality with the expectation that can be expressed using the following formula:

$$\mathbb{E}[|XY|] \leq (\mathbb{E}[|X|^p])^{\frac{1}{p}} (\mathbb{E}[|Y|^q])^{\frac{1}{q}},$$

and equality holds if and only if there exist constants $c \geq 0$ such that:

$$|Y| = c|X|^{\frac{p}{q}} \quad \text{almost surely}, \quad c \geq 0.$$

. By using Hölder inequality, with the same reasoning we have:

$$\sup_{\substack{\mathbb{Q}_x \in \mathbb{B}_\delta((g_\# \mathbb{P}_N)_\mathcal{X}) \\ \mathbb{Q}_a \in \mathcal{P}(\mathcal{A})}} \left\{ \mathbb{E}_{\substack{u'_x \sim \mathbb{Q}_x \\ u'_a \sim \mathbb{Q}_a}} [\nabla_x J(u_x, u'_a, \theta) \cdot (u'_x - u_x)] \right\} =$$

$$\sup_{\mathbb{Q}_x \in \mathbb{B}_\delta((g_\# \mathbb{P}_N)_\mathcal{X})} \left\{ \mathbb{E}_{u'_x \sim \mathbb{Q}_x} [\sup_{u'_a \in \mathcal{U}_\mathcal{A}} \{ \nabla_x J(u_x, u'_a, \theta) \cdot (u'_x - u_x) \}] \right\} =$$

$$\left( \mathbb{E}_{u_x \sim (g_\# \mathbb{P}_N)_\mathcal{X}} [\sup_{u'_a \in \mathcal{U}_\mathcal{A}} \{ \|\nabla_x J(u_x, u'_a, \theta)\|_*^q \}] \right)^{\frac{1}{q}} \sup_{\mathbb{Q}_x \in \mathbb{B}_\delta((g_\# \mathbb{P}_N)_\mathcal{X})} \left\{ \left( \mathbb{E}_{\substack{u'_x \sim \mathbb{Q}_x \\ u_x \sim (g_\# \mathbb{P}_N)_\mathcal{X}}} [\|u'_x - u_x\|^p] \right)^{\frac{1}{p}} \right\}$$

$$= \delta \left( \mathbb{E}_{u_x \sim (g_\# \mathbb{P}_N)_\mathcal{X}} [\sup_{u'_a \in \mathcal{U}_\mathcal{A}} \{ \|\nabla_x J(u_x, u'_a, \theta)\|_*^q \}] \right)^{\frac{1}{p}}$$

where equality can be attained whenever $1 \leq p \leq \infty$ for a proper choice of $u'_x - u_x$ with $(\mathbb{E}[\|u'_x - u_x\|^p])^{1/p} = \delta$. Therefore,

$$\mathcal{R}_\delta(\mathbb{P}_N) = \mathbb{E}_{v \sim \mathbb{P}_N} \left[ \sup_{a \in \mathcal{A}} \ell(\ddot{v}_a, y, \theta) \right] + \delta \left( \mathbb{E}_{v \sim \mathbb{P}_N} [\sup_{a \in \mathcal{A}} \{ \|\nabla^{\text{cf}} \ell(\ddot{v}, y, \theta)\|_*^q \}] \right)^{1/q} + O(\delta^2)$$

where

$$\nabla^{\text{cf}} \ell(v, y, \theta) = \lim_{\Delta \to 0} \frac{\ell(\mathbf{CF}_0(v, \Delta)) - f(v)}{\|\Delta\|}$$

the last equation completes the proofs.

### B.5 Proof of Proposition 2.

By Eq. 16 we have:

$$\mathcal{R}_\delta(\mathbb{P}) = \inf_{\lambda \geq 0} \left\{ \lambda \delta + \mathbb{E}_{u_x \sim (g_\# \mathbb{P})_\mathcal{X}} [\tilde{\ell}_\lambda(u_x)] \right\}$$

where, $\tilde{\ell}(u_x) = \sup_{u_a \in \mathcal{U}_\mathcal{A}} \{ \ell(M(u_a, u'_x)) \}$ and $\eta_\lambda(u_x) = \sup_{\Delta \in \mathcal{U}_\mathcal{X}} \tilde{\ell}(u_x + \Delta) - \lambda d_\mathcal{X}(u_x, u_x + \Delta)\}$. To prove we use Corollary 2 [28] for the $\tilde{\ell}$. First, we need to show that $\tilde{\ell}$ satisfies the condition of Corollary 2 [28]. By assumption for $v' \in \mathcal{V}$, $L, M \geq 0$ such that $|\ell(v, y, \theta) -$

$\ell(v', y, \theta)| < Ld^p(v, v') + M$ for all $v \in \mathcal{V}$ and $p \in [1, \infty)$. By setting $v = g^{-1}((u_a, u_a))$ and $v' = g^{-1}((u'_a, u'_a))$, and for simplicity $\ell(v) = \ell(v, y, \theta)$ we have:

$$|\ell(g^{-1}((u_a, u_x))) - \ell(g^{-1}((u'_a, u'_x)))| < L \|u_x - u'_x\|^p + M, \quad \forall u_x \in \mathcal{U}_\mathcal{X}, \ u_a \in \mathcal{U}_\mathcal{A} \Rightarrow$$

$$|\tilde{\ell}(u_x) - \tilde{\ell}(u'_x)| \leq L \|u_x - u'_x\|^p + M$$

where $\tilde{\ell}(u_x) = sup_{u_a \in \mathcal{U}_\mathcal{A}} \ell(g^{-1}((u_a, u_x)))$. This equation implies that $\tilde{\ell}$ satisfies the condition of corollary 2. So in consequence of corollary 2 and the equation 16 if we define uncertainty set:

$$\tilde{B}_\delta = \left\{ (\omega_x^{ik})_{i,k} : \frac{1}{N} \sum_{i=1}^N \sum_{k=1}^K \|u_x^i - \omega^{ik}\| \leq \delta, \ \omega^{ik} \in \mathcal{U}_\mathcal{X} \right\}$$

Since the casual fair metric $d$ and loss function $\tilde{\ell}$ do not depend on the sensitive part then $\tilde{B}_\delta$ is equivalent to the below uncertainty set.

$$B_\delta = \left\{ (w^{ik})_{i,k} : \frac{1}{N} \sum_{i=1}^N \sum_{k=1}^K d^p(v_i, w^{ik}) \leq \delta, \ w^{ik} \in \mathcal{V} \right\}.$$

By applying the uncertainty $\tilde{B}_\delta$ for $\tilde{\ell}$, the robust optimization problem has a form:

$$\tilde{\mathcal{R}}_\delta^{adv}(\mathbb{P}_N) =$$

$$\sup_{(\omega^{ik})_{i,k} \in \tilde{B}_\delta} \left\{ \frac{1}{NK} \sum_{i=1}^N \sum_{k=1}^K \tilde{\ell}(\omega^{ik}) \right\} = \sup_{(\omega^{ik})_{i,k} \in \tilde{B}_\delta} \left\{ \frac{1}{NK} \sum_{i=1}^N \sum_{k=1}^K \max_{u_a \in \mathcal{U}_\mathcal{A}} \ell(g^{-1}((u_a, \omega^{ik}))) \right\} =$$

$$\sup_{(w^{ik})_{i,k} \in B_\delta} \left\{ \frac{1}{NK} \sum_{i=1}^N \sum_{k=1}^K \max_{a \in \mathcal{A}} \ell(\ddot{w}_a^{ik})) \right\}$$

and finally for $\tilde{\mathcal{R}}_\delta^{adv}(\mathbb{P}_N)$ we have:

$$\tilde{\mathcal{R}}_\delta^{adv}(\mathbb{P}_N) \leq \mathcal{R}_\delta^n(\mathbb{P}_N) \leq \tilde{\mathcal{R}}_\delta^{adv}(\mathbb{P}_N) + \frac{LD + M}{NK},$$

and it completes the proof.

**Lemma 8** *Assume that the cost functions $c$ and $\hat{c}$ satisfy $|c(x, x') - \hat{c}(x, x')| < \alpha$ for all $x, x' \in \mathbb{R}^n$ and for some $\alpha \geq 0$. Then, for any $\lambda \geq 0$, the difference between the $\lambda$-conjugates of $f$ with respect to $c$ and $\hat{c}$ is bounded by $\lambda\alpha$:*

$$|f_\lambda(x) - \hat{f}_\lambda(x)| \leq \lambda\alpha \quad \text{for all } x \in \mathbb{R}^n.$$

**Proof.** To prove the proposition, we consider any $x \in \mathbb{R}^n$ and examine the definitions of $f_\lambda(x)$ and $\hat{f}_\lambda(x)$. Begin by expressing the bounds on $\hat{c}$:

$$\hat{c}(x, x') \leq c(x, x') + \alpha \quad \text{and} \quad \hat{c}(x, x') \geq c(x, x') - \alpha.$$

From these inequalities, for any $x' \in \mathbb{R}^n$,

$$f(x') - \lambda\hat{c}(x, x') \geq f(x') - \lambda(c(x, x') + \alpha) = f(x') - \lambda c(x, x') - \lambda\alpha,$$

$$f(x') - \lambda\hat{c}(x, x') \leq f(x') - \lambda(c(x, x') - \alpha) = f(x') - \lambda c(x, x') + \lambda\alpha.$$

Taking the supremum over all $x'$ in the above expressions, we obtain:

$$\hat{f}_\lambda(x) \geq f_\lambda(x) - \lambda\alpha \quad \text{and} \quad \hat{f}_\lambda(x) \leq f_\lambda(x) + \lambda\alpha.$$

These two bounds together imply:

$$|f_\lambda(x) - \hat{f}_\lambda(x)| \leq \lambda\alpha.$$

Thus, the proof is complete, showing that the difference between the $\lambda$-conjugates of $f$ is indeed bounded by $\lambda\alpha$.

**Lemma 9** *Let $c, \hat{c} : \mathbb{R}^n \times \mathbb{R}^n \to \mathbb{R}$ be two functions such that $|c(x, y) - \hat{c}(x, y)| < \alpha$ for all $x, y \in \mathbb{R}^n$ and some $\alpha > 0$. For any real number $p \geq 1$, the following inequality holds:*

$$|c(x, y)^p - \hat{c}(x, y)^p| \leq p \cdot M^{p-1} \cdot \alpha,$$

*where $M \geq \max\{|c(x, y)|, |\hat{c}(x, y)|\}$ for all $x, y$.*

**Proof.** Consider the functions $c$ and $\hat{c}$ and any $x, y \in \mathbb{R}^n$. By the hypothesis, we have $|c(x, y) - \hat{c}(x, y)| < \alpha$. To find a bound on the difference of their powers, apply the mean value theorem to the function $f(t) = t^p$, which is differentiable over $\mathbb{R}$ (or over $\mathbb{R}^+$ if $p$ is not an integer). The derivative of $f$ is $f'(t) = pt^{p-1}$.

Since $f$ is continuously differentiable, there exists some $\xi$ between $c(x, y)$ and $\hat{c}(x, y)$ such that

$$f(c(x, y)) - f(\hat{c}(x, y)) = f'(\xi) \cdot (c(x, y) - \hat{c}(x, y)).$$

Therefore,

$$|c(x, y)^p - \hat{c}(x, y)^p| = |p\xi^{p-1}(c(x, y) - \hat{c}(x, y))|.$$

Using the bound $|c(x, y) - \hat{c}(x, y)| < \alpha$ and noting that $\xi$ must be within the range of values between $c(x, y)$ and $\hat{c}(x, y)$, we have $\xi^{p-1} \leq M^{p-1}$. Thus,

$$|c(x, y)^p - \hat{c}(x, y)^p| \leq pM^{p-1}|c(x, y) - \hat{c}(x, y)| < pM^{p-1}\alpha.$$

**Lemma 10 ([41])** *Fix some $\mathbb{P} \in \mathcal{P}(\mathcal{Z})$, $\theta \in \Theta$ and $\lambda^* \geq 0$ via*

$$\lambda^* := \operatorname{argmin}_{\lambda \geq 0} \left\{ \lambda \delta^p + \mathbb{E}_{\mathbb{P}}[\ell_\lambda(\mathbf{Z}, \theta)] \right\}.$$

*Then under $\ell$ Lipschitz and $\operatorname{diam}(\mathcal{Z}) < \infty$ assumptions, we have $\lambda^* \leq L\delta^{-(p-1)}$.*

**Proof.** First, note that:

$$\lambda^* \delta^p \leq \lambda^* \delta^p + \mathbb{E}_{\mathbb{P}} \left[ \sup_{z' \in \mathcal{Z}} \left\{ \ell(z', \theta) - \ell(z, \theta) - \lambda^* c^p(z, z') \right\} \right] = *,$$

since the left-hand side, is greater than the case where $z' = z$ so it is positive. By the optimality of $\lambda^*$ in $\ell$, the right-hand side can be further upper-bounded as follows for any $\lambda \geq 0$:

$$* \leq \lambda \delta^p + \mathbb{E}_{\mathbb{P}} \left[ \sup_{z' \in \mathcal{Z}} \left\{ \ell(z', \theta) - \ell(z, \theta) - \lambda c^p(z, z') \right\} \right]$$

$$\leq \lambda \delta^p + \mathbb{E}_{\mathbb{P}} \left[ \sup_{z' \in \mathcal{Z}} \left\{ Lc(z, z') - \lambda c^p(z, z') \right\} \right]$$

$$\leq \lambda \delta^p + \sup_{t \geq 0} \{ Lt - \lambda t^p \},$$

using the Lipschitz property for the second line and setting $t = c(z, z')$ in the third line. If $p = 1$, by setting $\lambda = L$, we obtain:

$$\lambda^* \delta \leq L\delta + \sup_{t \geq 0} \{ Lt - Lt \} = L\delta,$$

which implies $\lambda^* \leq L$. For $p > 1$, using the optimal value $t = (L/p\lambda)^{1/(p-1)}$, we derive:

$$\lambda^* \delta^p \leq \lambda \delta^p + L^{\frac{p}{p-1}} p^{-\frac{p}{p-1}} (p - 1)\lambda^{-\frac{1}{p-1}}.$$

Minimizing the right-hand side with $\lambda = L/p\delta^{p-1}$ yields:

$$\lambda^* \delta^p \leq L\delta \Rightarrow \lambda^* \leq L\delta^{-(p-1)},$$

resulting in the stated bound on $\lambda^*$.

**Lemma 11** *If $f : A \times \mathbb{R}^n \to \mathbb{R}$ is Lipschitz with respect to $x$ uniformly in $a$, i.e., there exists a constant $L$ such that for all $a \in A$ and for all $x, y \in \mathbb{R}^n$,*

$$|f(a, x) - f(a, y)| \leq L\|x - y\|,$$

*then $\sup_{a \in A} f(a, x)$ is also Lipschitz in $x$.*

**Proof.** Let $F(x) = \sup_{a \in A} f(a, x)$. We aim to show that there exists a constant $L'$ such that for all $x, y \in \mathbb{R}^n$,

$$|F(x) - F(y)| \leq L'\|x - y\|.$$

Since $f$ is Lipschitz continuous with respect to $x$ uniformly in $a$ with Lipschitz constant $L$, it holds for each $a \in A$ and any $x, y \in \mathbb{R}^n$ that

$$|f(a, x) - f(a, y)| \leq L\|x - y\|.$$

Consider $F(x)$ and $F(y)$. By definition,

$$F(x) = \sup_{a \in A} f(a, x) \quad \text{and} \quad F(y) = \sup_{a \in A} f(a, y).$$

For any $a \in A$, since $|f(a, x) - f(a, y)| \leq L\|x - y\|$, we can infer that

$$f(a, x) \leq f(a, y) + L\|x - y\|.$$

Taking the supremum over all $a \in A$ on both sides, we obtain

$$F(x) \leq F(y) + L\|x - y\|.$$

Similarly,

$$F(y) \leq F(x) + L\|x - y\|.$$

Combining these two inequalities, we find

$$|F(x) - F(y)| \leq L\|x - y\|.$$

Therefore, $F(x) = \sup_{a \in A} f(a, x)$ is Lipschitz continuous with Lipschitz constant $L$. This completes the proof.

## B.6 Proof of Theorem 5.

Let define define $\hat{\mathcal{R}}_\delta(\mathbb{P}, \theta) := \sup_{\mathbb{Q}: W_{\hat{c}, p}(\mathbb{Q}, \mathbb{P}) \leq \delta} \mathbb{E}_\mathbb{Q}[\ell(\mathbf{Z}, \theta)]$, the worst-case loss quantity over estimation of the metric $d$ and $\theta_* := \inf_{\theta \in \Theta} \left\{ \sup_{\mathbb{Q}: W_{c, p}(\mathbb{Q}, \mathbb{P}_*) \leq \delta} \mathbb{E}_\mathbb{Q}[\ell(\mathbf{Z}, \theta)] \right\}$. Therefore by definition, we can write:

$$\mathcal{R}_\delta(\mathbb{P}_*, \hat{\theta}_N^{\mathrm{dro}}) - \mathcal{R}_\delta(\mathbb{P}_*, \theta_*) \leq \mathcal{R}_\delta(\mathbb{P}_*, \hat{\theta}_N^{\mathrm{dro}}) - \hat{\mathcal{R}}_\delta(\mathbb{P}_N, \hat{\theta}_N^{\mathrm{dro}}) - (\mathcal{R}_\delta(\mathbb{P}_*, \theta_*) - \hat{\mathcal{R}}_\delta(\mathbb{P}_N, \theta_*)) \quad (23)$$

because we have $\hat{\mathcal{R}}_\delta(\mathbb{P}_N, \hat{\theta}_N^{\mathrm{dro}}) \leq \mathcal{R}_\delta(\mathbb{P}_*, \theta_*)$.

We estimate two expression $|\mathcal{R}_\delta(\mathbb{P}_*, \hat{\theta}_N^{\mathrm{dro}}) - \hat{\mathcal{R}}_\delta(\mathbb{P}_N, \hat{\theta}_N^{\mathrm{dro}})|$ and $|\mathcal{R}_\delta(\mathbb{P}_*, \theta_*) - \hat{\mathcal{R}}_\delta(\mathbb{P}_N, \theta_*)|$. By the general strong duality theorem [26] we have:

$$\mathcal{R}_\delta(\mathbb{P}_*, \theta_*) - \hat{\mathcal{R}}_\delta(\mathbb{P}_N, \theta_*) = \sup_{\mathbb{Q}: W_{c, p}(\mathbb{Q}, \mathbb{P}_*) \leq \delta} \mathbb{E}_\mathbb{Q}[\ell(\mathbf{Z}, \theta_*)] - \sup_{\mathbb{Q}: W_{\hat{c}, p}(\mathbb{Q}, \mathbb{P}_N) \leq \delta} \mathbb{E}_\mathbb{Q}[\ell(\mathbf{Z}, \theta_*)] =$$

$$\inf_{\lambda \geq 0} \{\lambda\delta^p + \mathbb{E}_{\mathbb{P}_*}[\ell_\lambda^c(\mathbf{Z}, \theta_*)]\} - \inf_{\lambda \geq 0} \{\lambda\delta^p + \mathbb{E}_{\mathbb{P}_N}[\ell_\lambda^{\hat{c}}(\mathbf{Z}, \theta_*)]\} \leq$$

$$\lambda_N\delta^p + \mathbb{E}_{\mathbb{P}_*}[\ell_{\lambda_N}^c(\mathbf{Z}, \theta_*)]\} - (\lambda_N\delta^p + \mathbb{E}_{\mathbb{P}_N}[\ell_{\lambda_N}^{\hat{c}}(\mathbf{Z}, \theta_*)]) =$$

$$\mathbb{E}_{\mathbb{P}_*}[\ell_{\lambda_N}^c(\mathbf{Z}, \theta_*)]\} - \mathbb{E}_{\mathbb{P}_N}[\ell_{\lambda_N}^{\hat{c}}(\mathbf{Z}, \theta_*)]$$

where the $\lambda_N := \arginf_{\lambda \geq 0} \{\lambda\delta^p + \mathbb{E}_{\mathbb{P}_N}[\ell_\lambda^{\hat{c}}(\mathbf{Z}, \theta_*)]\}$

By assumption 2 and lemma 9,

$$|\ell_{\lambda_N}^c(z, \theta) - \ell_{\lambda_N}^{\hat{c}}(z, \theta)| = \left| \sup_{v' \in \mathcal{V}} \ell(z', y, \theta) - \lambda_N d^p(v', v) - \sup_{v' \in \mathcal{V}} \ell(v', y, \theta) - \lambda_N \hat{d}^p(v', v) \right|$$

$$\leq \sup_{v' \in \mathcal{V}} \lambda_N |d^p(v', v) - \hat{d}^p(v', v)| \leq \lambda_N\, p\, \mathrm{diam}\,(\mathcal{V})^{p-1}\, M_d\, N^{-\eta}.$$

This implies

$$\mathcal{R}_\delta(\mathbb{P}_*, \theta_*) - \hat{\mathcal{R}}_\delta(\mathbb{P}_N, \theta_*) \leq \mathbb{E}_{P_*}[\ell_{\lambda_N}^c(\mathbf{Z}, \theta)] - \mathbb{E}_{P_N}[\ell_{\lambda_N}^c(\mathbf{Z}, \theta)] + \lambda_N\, p\, \mathrm{diam}\,(\mathcal{V})^{p-1}\, M_d\, N^{-\eta}.$$

If define $\lambda_* := \arginf_{\lambda \geq 0} \{\lambda \delta^p + \mathbb{E}_{\mathbb{P}}[\ell^c_\lambda(\mathbf{Z}, \theta_*)]\}$, similarly,

$$\hat{\mathcal{R}}_\delta(\mathbb{P}_N, \theta_*) - \mathcal{R}_\delta(\mathbb{P}_*, \theta_*) \leq \mathbb{E}_{\mathbb{P}_N}[\ell^{\hat{c}}_{\lambda_*}(\mathbf{Z}, \theta)] - \mathbb{E}_{\mathbb{P}_*}[\ell^c_{\lambda_*}(\mathbf{Z}, \theta)] \leq$$
$$\mathbb{E}_{\mathbb{P}_N}[\ell^c_{\lambda_*}(\mathbf{Z}, \theta)] - \mathbb{E}_{\mathbb{P}_*}[\ell^c_{\lambda_*}(\mathbf{Z}, \theta)] + \lambda_* \, p \, \mathrm{diam}\,(\mathcal{V})^{p-1} \; M_d \, N^{-\eta}.$$

We need to estimate the $\lambda_N$ and $\lambda_*$. By using strong duality theorem by Eq. 16 we have:

$$\mathcal{R}_\delta(\mathbb{P}) = \inf_{\lambda \geq 0} \left\{ \lambda \delta^p + \mathop{\mathbb{E}}_{u_x \sim (g_\# \mathbb{P})_\mathcal{X}} \left[ \tilde{\ell}_\lambda(u_x) \right] \right\} \tag{24}$$

So instead the solve problem for $\ell$ is it sufficient to prove our result for $\tilde{\ell}(u_x) = \sup_{u_a \in \mathcal{U}_\mathcal{A}} \ell(g^{-1}((u_a, u_x)))$. At first, we show that $\tilde{\ell}$ is Lipschitz on the space $\mathcal{U}_\mathcal{X}$ concerning the norm $\|.\|$. By assumption, for each $a \in \mathcal{A}$ the function $\ell(g^{-1}((u_a, u_x)))$ is also Lipschitz:

$$\left\| \ell(g^{-1}((u_a, u_x)), y, \theta) - \ell(g^{-1}((u_a, u_x + \Delta)), y, \theta) \right\| =$$
$$\left\| \ell(\mathbf{CF}(v, a), y, \theta) - \ell(\mathbf{CF}(v, a, \Delta), y, \theta) \right\| \leq L d(\mathbf{CF}(v, a), \mathbf{CF}(v, a, \Delta)) = L \, \|\Delta\|$$

Now by using lemma 11 it can be concluded that the function $\tilde{\ell}(u_x) = \sup_{u_a \in \mathcal{U}_\mathcal{A}} \ell(g^{-1}((u_a, u_x)))$ also has Lipschitz property with constant $L$. By Applying lemma 10 for equation 24 and $(g_\# \mathbb{P}_N)_\mathcal{X}$, it can be seen $\lambda_N, \lambda_* \leq L \delta^{-(p-1)}$. Therefore until now, we have two inequalities:

$$\mathcal{R}_\delta(\mathbb{P}_*, \theta_*) - \hat{\mathcal{R}}_\delta(\mathbb{P}_N, \theta_*) \leq \mathbb{E}_{P_*}[\ell^c_{\lambda_N}(\mathbf{Z}, \theta)] - \mathbb{E}_{P_N}[\ell^c_{\lambda_N}(\mathbf{Z}, \theta)] + \lambda_N \, p \, \mathrm{diam}\,(\mathcal{V})^{p-1} \; M_d \, N^{-\eta},$$

$$\hat{\mathcal{R}}_\delta(\mathbb{P}_N, \theta_*) - \mathcal{R}_\delta(\mathbb{P}_*, \theta_*) \leq \mathbb{E}_{\mathbb{P}_N}[\ell^c_{\lambda_*}(\mathbf{Z}, \theta)] - \mathbb{E}_{\mathbb{P}_*}[\ell^c_{\lambda_*}(\mathbf{Z}, \theta)] + \lambda_* \, p \, \mathrm{diam}\,(\mathcal{V})^{p-1} \; M_d \, N^{-\eta} \Rightarrow$$

$$|\mathcal{R}_\delta(\mathbb{P}_*, \theta_*) - \hat{\mathcal{R}}_\delta(\mathbb{P}_N, \theta_*)| \leq \sup_{f \in \mathcal{L}^c} \left| \int_\mathcal{Z} f(z) d(\mathbb{P}_N - \mathbb{P}_*)(z) \right| + L \delta^{1-p} \, p \, \mathrm{diam}\,(\mathcal{V})^{p-1} \; M_d \, N^{-\eta},$$

where $\mathcal{L}^c = \{\ell^c_\lambda(\cdot, \theta) : \lambda \in [0, L\delta^{1-p}], \theta \in \Theta\}$ is the DR loss class.

In the remaining part of the proof, we estimate $\sup_{f \in \mathcal{L}^c} |\int_\mathcal{Z} f(z) d(\mathbb{P}_* - \mathbb{P}_N)(z)|$ using conventional methods from statistical learning theory. According to assumption 2, the functions within $\mathcal{F}$ are limited as shown:

$$0 \leq \ell^c_\lambda(v, \theta) \leq \sup_{v' \in \mathcal{V}} \ell(v', y, \theta) - \lambda d(v, v') \leq \sup_{v' \in \mathcal{V}} \ell(v', y, \theta) \leq M.$$

similar to the proof Theorem 3 [41], by utilizing the bounded-differences inequality and symmetrization, we derive that:

$$\sup_{f \in \mathcal{L}^c} \left| \int_\mathcal{Z} f(z) d(\mathbb{P}_N - \mathbb{P}_*)(z) \right| \leq 2\mathfrak{R}_N(\mathcal{L}^c) + M \sqrt{\frac{\log \frac{2}{\epsilon}}{2N}}$$

holds with a probability of at least $1 - \epsilon$, where $\mathfrak{R}_n(\mathcal{L}^c)$ represents the Rademacher complexity of $\mathcal{L}^c$:

$$\mathfrak{R}_\mathfrak{N}(\mathcal{L}^c) = \mathbb{E}\left[ \sup_{f \in \mathcal{L}^c} \frac{1}{N} \sum_{i=1}^N \sigma_i f(Z_i) \right].$$

In the proof of Theorem 2 [41], the authors has proved:

$$\mathfrak{R}_N(\mathcal{L}^c) \leq \frac{24\mathfrak{C}(\mathcal{L})}{\sqrt{N}} + \frac{24 L.\mathrm{diam}\,(\mathcal{V})^p}{\sqrt{N}\delta^{p-1}}.$$

where $\mathfrak{C}(\mathcal{L})$, the entropy integral of of loss class. By applying this result it can be written:

$$|\mathcal{R}_\delta(\mathbb{P}_*, \theta_*) - \hat{\mathcal{R}}_\delta(\mathbb{P}_N, \theta_*)| \leq M \sqrt{\frac{\log \frac{2}{\epsilon}}{2N}} + \frac{48\mathfrak{C}(\mathcal{L})}{\sqrt{n}} + \frac{48 L.\mathrm{diam}\,(\mathcal{V})^p}{\sqrt{n}\delta^{p-1}} + L\delta^{1-p} \, p \, \mathrm{diam}\,(\mathcal{V})^{p-1} \; M_d \, N^{-\eta}$$

Since the prove does depend on value of $\theta$, then $|\mathcal{R}_\delta(\mathbb{P}_*, \hat{\theta}_N^{\mathrm{dro}}) - \hat{\mathcal{R}}_\delta(\mathbb{P}_N, \hat{\theta}_N^{\mathrm{dro}})|$ also satisfies in the above inequality. By combining two terms with probability $1 - \epsilon$ we have,

$$\mathcal{R}_\delta(\mathbb{P}_*, \hat{\theta}_N^{\mathrm{dro}}) - \mathcal{R}_\delta(\mathbb{P}_*, \theta_*) \leq M \sqrt{\frac{2 \log \frac{2}{\epsilon}}{N}} + \frac{96\mathfrak{C}(\mathcal{L})}{\sqrt{n}} + \frac{96 L.\mathrm{diam}\,(\mathcal{V})^p}{\sqrt{n}\delta^{p-1}} + 2L\delta^{1-p} \, p \, \mathrm{diam}\,(\mathcal{V})^{p-1} \; M_d \, N^{-\eta}$$

Since by assumption, with probability $1 - \epsilon$ we have inequality

$$\forall z, z' \in |c(z, z') - \hat{c}(z, z')| \leq M_d N^{-\eta}, \quad \eta > 0$$

therefore by probability $1 - 2\epsilon$ the main inequality is true and it completes the proof.

## C  Numerical Analysis Supplementary

### C.1  Synthetic Data Models

The structural equations used to generate the SCMs in § 5 are listed below. For the LIN SCM, we generate the protected feature $\mathbf{A}$ and variables $\mathbf{X}_i$ according to the following structural equations:

- linear SCM (LIN):

$$\begin{cases} \mathbf{A} := \mathbf{U_A}, & \mathbf{U_A} \sim \mathcal{B}(0.5) \\ \mathbf{X}_1 := 2\mathbf{A} + U_1, & U_1 \sim \mathcal{N}(0,1) \\ \mathbf{X}_2 := \mathbf{A} - \mathbf{X}_1 + \mathbf{U}_2, & \mathbf{U}_2 \sim \mathcal{N}(0,1) \\ \mathbf{Y} \sim \mathcal{B}((1 + exp(-(\mathbf{X}_1 + \mathbf{X}_2))^{-1}) \end{cases}$$

Here, $\mathcal{B}(p)$ represents Bernoulli random variables with probability $p$, and $\mathcal{N}(\mu, \sigma^2)$ represents normal random variables with mean $\mu$ and variance $\sigma^2$. To generate the ground truth $h(\mathbf{A}, X_1, X_2)$, we use a linear model for the LIN method. In all the synthetic models considered, we treat $\mathbf{A}$ as a binary-sensitive attribute.

### C.2  Real-World Data

In our research, we have utilized the Adult dataset [38] and the COMPAS dataset [65] for our experimental analysis. To employ these datasets, we initially constructed an SCM based on the causal graph proposed by Nabi et al. [49]. For the Adult dataset, we incorporate features such as **sex**, **age**, **native-country**, **marital-status**, **education-num**, **hours-per-week**, and consider gender as a sensitive attribute. In the case of the COMPAS dataset, the utilized features comprise **age**, **race**, **sex**, and **priors count**, which function as variables. Additionally, sex is considered a sensitive attribute.

For classification purposes, we apply data standardization before the learning process.

$$\mathbf{Adult} = \begin{cases} \mathbf{A} = \mathbf{U}_A & \textbf{Sex} \\ \mathbf{X}_2 = \mathbf{U}_2 & \textbf{Age} \\ \mathbf{X}_3 = \mathbf{U}_3 & \textbf{Country} \\ \mathbf{X}_4 = \mathbf{U}_4 & \textbf{Marital Status} \\ \mathbf{X}_5 = \beta_{51}\mathbf{X}_1 + \beta_{52}\mathbf{X}_2 + \beta_{53}\mathbf{X}_3 + \beta_{54}\mathbf{X}_4 + \mathbf{U}_5 & \textbf{Education Level} \\ \mathbf{X}_6 = \beta_{61}\mathbf{X}_1 + \beta_{62}\mathbf{X}_2 + \beta_{63}\mathbf{X}_3 + \beta_{64}\mathbf{X}_4 + \beta_{65}\mathbf{X}_5 + \mathbf{U}_6 & \textbf{Hours per Week} \end{cases}$$

$$\mathbf{COMPAS} = \begin{cases} X_1 = U_1 & \textbf{Sex} \\ X_2 = U_2 & \textbf{Age} \\ X_3 = U_3 & \textbf{Race} \\ X_4 = \beta_{41}X_1 + \beta_{42}X_2 + \beta_{43}X_3 + U_4 & \textbf{Priors Count} \end{cases}$$

In these above equations, $\beta_{ij}$ are the coefficients for the linear combinations of the $X$ variables, and $U_i$ are the exogenous variables.

### C.3  Training Methods

In our study, we utilize various training objectives to train decision-making classifiers, with loss function $\ell(v)$. The training objectives are as follows:

- **Empirical Risk Minimization (ERM)**: This approach minimizes the expected risk concerning the classifier parameters $\psi$, represented by

$$\inf_{\theta \in \Theta} \mathbb{E}_{z \sim \mathbb{P}}[\ell(z, \theta)]$$

- **Adversarial Learning (AL)**: This method trains the model to withstand or defend against adversarial perturbations, represented by

$$\inf_{\theta \in \Theta} \left\{ \mathbb{E}_{z \sim \mathbb{P}} \left[ \sup_{\|\Delta\| \leq \delta} \ell(z + \Delta, \theta) \right] \right\}$$

- **ROSS**: Based on the work of Ross et al. [56], this method minimizes the expected risk along with an adversarial perturbation term, represented by

$$\inf_{\theta \in \Theta} \left\{ \mathbb{E}_{(v,y) \sim \mathbb{P}} \left[ \ell(v, y, \theta) + \inf_{\|\Delta\| \leq \delta} \ell(v + \Delta, 1, \theta) \right] \right\}$$

- **CDRO**: Our approach, as described in this paper, is formulated as follows:

$$\inf_{\theta \in \Theta} \left\{ \sup_{\mathbb{Q} \in \mathbb{B}_\delta(\mathbb{P})} \mathbf{E}_{z \sim \mathbb{Q}}[\ell(z, \theta)] \right\}$$

For our loss function $\ell$, we use the binary cross-entropy loss.

## C.4   Hyperparameter Tuning

The majority of the experimental setup is based on the work of Ehyaei et al. [22]. For each dataset and its respective label, we use a generalized linear model (GLM). Each training objective is applied to four different datasets, using 100 different random seeds. The optimization process is performed using the Adam optimizer with a learning rate of $10^{-3}$ and a batch size of 100. After optimizing the benchmark time and considering the training rate, we set the number of epochs to 10 to ensure comparability in benchmarking.

## C.5   Metrics

To assess the performance of various training methods concerning accuracy, unfair area, counterfactual fairness, and adversarial robustness, we employ seven distinct metrics as outlined below:

- **Acc**: The accuracy of the classifier, is represented as a percentage.
- $U_\delta$: The proportion of data points within the unfair area with a radius of $\delta$.

$$U_\delta := \mathbb{P}\big(\{v \in \mathcal{V} : \quad \exists v' \in \mathcal{V} \quad \text{s.t.} \quad d(v, v') \leq \delta \quad \wedge \quad h(v) \neq h(v')\}\big).$$

- $R_\delta$: The fraction of data points that are vulnerable to adversarial perturbations within a radius of $\delta$. This metric coincides with the unfair area in cases where no sensitive attribute is considered.

$$R_\delta := \mathbb{P}\big(\{v \in \mathcal{V} : \quad \exists \Delta \in \mathcal{V} \quad \text{s.t.} \quad d(v, \mathbf{CF}(v, \Delta)) \leq \delta \quad \wedge \quad h(v) \neq h(\mathbf{CF}(v, \Delta))\}\big).$$

- $CF$: The percentage of data points that exhibit counterfactual unfairness. This metric aligns with the unfair area when the perturbation radius is zero.

$$\mathrm{CF} := \mathbb{P}\big(\{v \in \mathcal{V} : \quad \exists a \in \mathcal{A} \quad \text{s.t.} \quad h(v) \neq h(\ddot{v}_a)\}\big).$$

## C.6   Additional Results

In this section, we present additional simulation results. CDRO performs well across all datasets except for the $R_\delta$ measure in the Adult dataset, likely because the linear model does not fit the SCM well. Nevertheless, CDRO demonstrates robustness and counterfactual fairness, as shown in Table 1, making it the preferred model when balancing both accuracy and fairness.

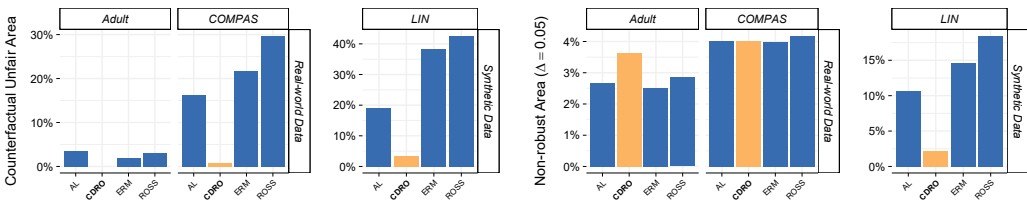

Figure 2: Displays the findings from our numerical experiment, assessing the performance of DRO across different models and datasets. (left) Counterfactual unfair area percentage (lower values are better). (right) Non-robust area performance of classifier (higher values are better) for $\Delta = .05$.

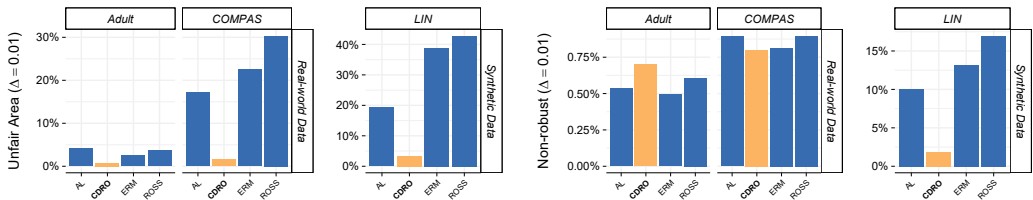

Figure 3: Displays the findings from our numerical experiment, assessing the performance of DRO across different models and datasets. (Left) Bar plot showing the comparison of models based on the unfair area percentage $U(\delta)$ (lower values are better) at $\Delta = .01$. (Right) Bar plot showing the comparison of models based on the robustness area percentage $R(\delta)$ (lower values are better) at $\Delta = .01$.

## Broader Impact Statement

Our theoretical framework bridges adversarial robustness, distributional robustness, individual fairness, and causality, aligning with the core pillars of responsible AI. By demonstrating the connection between these areas, we aim to inspire further research at their intersection and contribute to the development of safer, more equitable AI models for society. This approach holds the promise of improving decision-making under uncertainty while ensuring fairness and mitigating the impact of adversarial perturbations.

However, we acknowledge several limitations and ethical implications inherent in our approach. While our method produces fair and robust predictions under specific conditions, it is important to note that it fundamentally relies on a machine learning model, which may inherit the same vulnerabilities as the original model in areas not explicitly addressed in this work such as multiplicity,

| | Real-World Data | | | | | | | | | | | | Synthetic Data | | | | | |
| | Adult | | | | | | COMPAS | | | | | | LIN | | | | | |
| Trainer | Acc | $U_{0.5}$ | $U_{0.1}$ | CF | $R_{.05}$ | $R_{.01}$ | Acc | $U_{0.5}$ | $U_{0.1}$ | CF | $R_{.05}$ | $R_{.01}$ | Acc | $U_{0.5}$ | $U_{0.1}$ | CF | $R_{.05}$ | $R_{.01}$ |
|---|---|---|---|---|---|---|---|---|---|---|---|---|---|---|---|---|---|---|
| AL | 0.79 | 0.06 | 0.04 | 0.04 | 0.03 | 0.01 | 0.67 | 0.2 | 0.17 | 0.16 | 0.04 | 0.01 | 0.67 | 0.2 | 0.19 | 0.19 | 0.11 | 0.1 |
| CDRO | 0.73 | **0.04** | **0.01** | **0** | 0.04 | 0.01 | 0.66 | **0.05** | **0.02** | **0.01** | 0.04 | **0.01** | 0.66 | **0.04** | **0.03** | **0.03** | **0.02** | **0.02** |
| ERM | **0.79** | 0.05 | 0.02 | 0.02 | **0.03** | **0** | **0.68** | 0.26 | 0.23 | 0.22 | **0.04** | 0.01 | **0.69** | 0.4 | 0.39 | 0.38 | 0.15 | 0.13 |
| ROSS | 0.78 | 0.06 | 0.04 | 0.03 | 0.03 | 0.01 | 0.67 | 0.33 | 0.3 | 0.3 | 0.04 | 0.01 | 0.69 | 0.44 | 0.43 | 0.43 | 0.18 | 0.17 |

Table 1: The table presents the results of our numerical experiment, comparing various trainers based on their input sets in terms of accuracy (Acc, higher values are better), unfairness areas ($U_{.05}$, lower values are better), unfairness areas ($U_{.01}$, lower values are better), Counterfactual Unfair area (CF, lower values are better), the non-robust percentage concerning adversarial perturbation with radii 0.05 ($\mathcal{R}_{.05}$, lower values are better), and the non-robust percentage concerning adversarial perturbation with radii 0.01 ($\mathcal{R}_{.01}$, lower values are better). The top-performing techniques for each trainer, dataset, and metric are highlighted in bold. The findings demonstrate that CDRO excels in reducing unfair areas. The average standard deviation for CDRO is .029, while for the other methods, it is .031.

privacy breaches, lack of explainability, and safety/security concerns. Additionally, our work operates under simplifying assumptions regarding the fairness notion. In real-world applications, fairness is a complex and context-dependent concept. Therefore, it is essential to define fairness carefully and consider multiple dimensions of fairness when applying our approach. We emphasize that this work is a proof of concept, and we strongly recommend involving diverse stakeholders, including ethicists, domain experts, and affected communities, before applying our approach to high-risk application domains. It's important for users to be aware of these limitations and potential biases that might not be fully addressed by our framework.

