# OpenReview forum: "Wasserstein Distributionally Robust Optimization through the Lens of Structural Causal Models and Individual Fairness"
_NeurIPS.cc/2024/Conference — NeurIPS 2024 poster_

### Official Review · Reviewer_zg8Z · 2024-07-02

**Soundness:** 3
**Presentation:** 2
**Contribution:** 2
**Rating:** 5
**Confidence:** 3

**Summary:**

The authors propose a novel framework to enhance individual fairness guarantees under a Wasserstein distributionally robust optimization strategy. For such purposes, they employ counterfactuals based on the underlying causal structure of the model at hand. They further propose an alternative with theoretical guarantees for when the causal structure is unknown.

**Strengths:**

S1 - The issue being investigated is of significant importance.

S2 - The introduction and abstract effectively substantiate all the claims made, including the contributions put forth by the authors. These assertions find validation through a thorough description of the methodology employed and the theoretical results provided.

S3 - The paper demonstrates a strong mathematical foundation, supported by numerous theoretical results.

S4 - The paper demonstrates commendable attention to reproducibility by providing detailed information regarding the experimental setup.

**Weaknesses:**

W1 - The text lacks clarity in several areas. It becomes quite technical at times, and it would benefit from including more examples alongside the technical explanations, particularly in the introduction, to help readers develop a better intuition. Additionally, there are instances where the ideas presented lack cohesion and do not clearly relate to one another, making the overall narrative difficult to follow. Improving the flow and connection between ideas would enhance the readability and comprehension of the text.

W2 - The paper does not cite many significant works on individual fairness and algorithmic fairness in general. For instance, citing [1] would be relevant, as it shares many similar points with the presented work. It would be beneficial to outline both the similarities and differences between them. Additionally, the related works section overlooks a significant body of literature on algorithmic fairness that is based on causality. These works are numerous and, in my opinion, should be acknowledged.

W3 - It is not very clear which are the main advantages and benefits of the proposed approach with respect to existing works in the literature.

W4 - As I understand it, the authors propose modifying only the sensitive attribute while leaving all other attributes unchanged (please correct me if I am wrong). However, this approach can create unrealistic twins. For example, consider a male instance with a height of 1.85m, which is a typical height in a European country. If we change the gender to female without adjusting the height, the resulting instance would be an outlier and not representative of a typical female, making the two instances not equivalent. In the presence of such unrealistic twins, the classifier's decisions regarding these instances may not be informative. This issue arises because bias exists not only in the sensitive attribute but also in the non-sensitive attributes; for example, gender and income can be highly correlated.  Therefore, when we change the sensitive attribute of an instance, in order to create an ‘equivalent’ instance of the other gender, non-sensitive attributes should as well be modified. That is, to ensure the relevance of their analysis, the authors should verify that the twins they consider are realistic and plausible instances. For more insight on this matter, see [1].

W5 - The empirical evaluation of the proposal is poor. It does not include popular benchmark fairness-enhancing interventions, and only a few classification tasks are considered, despite the availability of numerous widely-used datasets in the algorithmic fairness literature. Besides, there is no deep discussion regarding the results.

W6 - The model lacks any discussion or insights regarding its computational complexity or cost.

W7 - The acronym SCM is used before it is defined: it is first used in line 49, but it is defined in line 89.

W8 - (minor) I suggest moving the related works section into a separate section, as it represents a distinct aspect of the work.

W9 - (typos), line 60 (Our → our), line 113 (variables[50] → variables [50]), line 204 (space 9 ??)

[1] De Lara, L., González-Sanz, A., Asher, N., Risser, L., & Loubes, J. M. (2024). Transport-based counterfactual models. Journal of Machine Learning Research, 25(136), 1-59.

**Questions:**

Q1 - Which are the main benefits/advantages that are provided by this proposal with respect to existing works in the literature?

Q2 - Does the approach use the conventional Wassterstein-ball based uncertainty set or is the uncertainty set considered the one from equation (16)? Is there any equivalence between them?

 Q3 - Recognizing that assuming knowledge of the underlying causal structure for a given classification task is generally unrealistic, the authors propose an alternative approach that operates under more realistic conditions, where only a set of samples/instances is available. They offer guarantees as long as the assumptions outlined in Assumption 2 are met. However, it is unclear how realistic these assumptions are. Are they typically satisfied in real-world applications? Can you provide real examples where these assumptions hold true?

Q4 - Could this method be employed in classification tasks beyond tabular data?

Q5 - What do you mean by ‘We further estimate the regularizer in more general cases and explore the relationship between DRO and classical robust optimization.’? Where is this claim validated in the main text?

**Limitations:**

The authors discuss the limitations of their method in Section 6.

---

> ### Author Rebuttal · Authors · 2024-08-06
>
> First of all, thank you for your detailed comments. We will address each of them in detail.
>
> **W1.** In the global response, we explained that introducing a new DRO framework requires considering key theorems such as strong duality, closed-form worst-case loss, regularizer estimation, and finite sample guarantees to ensure practical applicability. We aimed to present a minimal and complete framework, making our theoretical results the main part of our work.
>
> **W2.** The paper you mentioned shares only keywords with my work and differs significantly in scope. They aim to provide a new method for computing counterfactual instances by bypassing the conventional three steps (Abduction, Action, and Prediction). They propose using an optimal transport map, suggesting that if $(X, S=s) \sim \mathbb{P}\_s$ and $(X, S=s' \mid X=x, S=s) \sim \mathbb{P}\_{s'}$ are probability distributions, then under strict conditions, the optimal transport theory can find a map $T: \mathcal{X} \to \mathcal{X}$ such that $T\_{\\#}\mathbb{P}\_s = \mathbb{P\}\_{s'}$. This means they use $T(x)$ instead of causally computing counterfactuals. However, since they rely on Euclidean distance to find the optimal map, this approach only works in very special cases, as the authors mentioned in the paper [1]. Therefore, this work does not align with the scope of our work.
>
> As mentioned in the global response, our work is not limited to algorithmic fairness and might omit some works in that field, similar to how we excluded papers in reinforcement learning and other fields. Our focus is on introducing new distributionally ambiguity set, and we have cited relevant references as comprehensively as possible.
>
> **W3.** In the global response, we discussed the philosophy and advantages of our work compared to existing methods. In addition, Chapter 4.1 outlines the benefits of our approach through comparisons with other works.
>
> **W4.** As mentioned in the background section, when we refer to counterfactuals, we mean computing them through the three steps: Abduction, Action, and Prediction. In line 100, we provide the counterfactual formulation. To clarify further, we detailed Example 1, demonstrating that the counterfactual of the instance $(M,1,1)$ is $(F,0,-2)$.
>
>
> **W5.** As mentioned in the global response, our framework is a general approach to applying DRO with causality and protected variables, applicable in fairness, adverse learning, robust learning, reinforcement learning, transfer learning, GAN, NLP, and more. We used fair learning as an example to demonstrate one application and compared our method only with those having similar assumptions, not all algorithmic fairness methods.
>
> **W6.** In lines 236-240, we discuss the computational aspect of the proposed DRO and reference papers that guarantee fast algorithms for computing our methods. We provide Theorems 2, 3, and 4 to demonstrate the computational efficiency of our approach by solving or estimating the worst-case quantity and incorporating it into the loss function.
>
> **W7-9.** We thank you for your suggestions; we have incorporated the changes and addressed these minor comments in the new version, and the related works may form a distinct chapter.
>
> **Q1.** We provided a detailed response to this in our global response.
>
> **Q2.** The uncertainty set in our method is defined using the Wasserstein distance, incorporating the causally fair dissimilarity function, effectively creating a Wasserstein ball equipped with CFDF. Proposition 2 shows that we can estimate the worst-case loss quantity using the robust optimization method defined in Equation 16, establishing their equivalence.
>
> **Q3.** The first part of Assumption 2 is natural in real applications, as it assumes that the feature space is bounded and the parameter space is bounded and closed.
>
> The third part is reasonable, as it relates to the statistical properties of the estimator for the cost function or causal model structure. For example, in a linear SCM, we can estimate its functional structure with a convergence rate of $O(N^{-\frac{1}{2}})$, allowing us to design a metric that satisfies the third assumption.
>
> The second part of Assumption 2 might seem new but is essentially a Lipschitz condition on perturbing non-sensitive features. Since perturbations in the causal structure are derived from counterfactuals, this is quite natural. For example, consider a linear SCM with reduced-form mapping $M$, where $X = MU$ and all sensitive attributes are parents. Here, $CF_0(v, \Delta) = v + M\Delta$. Given a loss function $\ell(v, y, \theta) = h(\theta^T v - y)$ where $h$ is Lipschitz, assuming a norm $\ell_p$ in the exogenous space leads to $d(v, CF_0(v, \Delta)) = \|\Delta\|_q$. This makes the Lipschitz condition:
>
> $$
> \vert \ell(v, y, \theta) - \ell(CF_0(v, \Delta), y, \theta) \vert = \vert h(\theta^T v - y) - h(\theta^T v - y + \theta^T M \Delta) \vert \leq L \|\theta^T M\|_p \|\Delta\|_q
> $$
>
> This naturally satisfies condition 2.
>
> **Q4.** Our method can be applied in fields where the Wasserstein distance or optimal transport is used, allowing for the incorporation of causality and protection of feature variables. In neural networks, especially GAN models, it enhances sensivity of model respct to causal structure. In signal processing, reinforcement learning, and NLP, it introduces causality and fairness into the Wasserstein distance.
>
> **Q5.** This statement refers to Theorem 4, which provides a first-order estimation of the worst-case loss quantity, described as a regularizer in our DRO framework. Proposition 2 further establishes the relationship between our proposed DRO method and conventional robust optimization techniques that typically involve adversarial perturbations.
>
> In conclusion, we hope our responses are comprehensive enough to capture your interest in this work.
>
> [1] De Lara et al. (2024). Transport-based counterfactual models.

---

> > ### Author Response · Authors · 2024-08-14
> > **Request for Reviewer Engagement During Rebuttal Process**
> >
> > We respectfully wish to express our concern regarding the lack of response from one of the reviewers during the rebuttal process. We invested significant effort in thoroughly addressing the concerns raised, providing a detailed response of approximately 6,000 characters to ensure that each issue was adequately covered. We were hopeful that the reviewer would engage with our rebuttal, as this dialogue is crucial for ensuring that all points are fully understood and resolved. Given the importance of this interaction, we kindly ask whether the reviewer feels that their concerns have been satisfactorily addressed or if there are any remaining questions that we can clarify during this stage. We believe that this feedback loop is essential to the integrity of the review process and the fair evaluation of our work.

---

### Official Review · Reviewer_rhFX · 2024-07-11

**Soundness:** 3
**Presentation:** 3
**Contribution:** 3
**Rating:** 6
**Confidence:** 2

**Summary:**

This paper uses wasserstein distributionally robust optimization to address individual fairness concerns with causal structures and sensitive attributes.

**Strengths:**

The problem is well-motivated and novel to my knowledge. The formulation is clear. The solution is novel.

**Weaknesses:**

It does not seem easy to scale up this method.

minor issue: the DRO objective in line 142 should be sup_{Q in B_delta(P)} En_Q [l(Z,theta)].

**Questions:**

How are the causal relationships determined for the experiments? If one assumes no causal relationship (e.g. i.i.d. formulation), does that impair the performance significantly?

**Limitations:**

adequately addressed

---

> ### Author Rebuttal · Authors · 2024-08-06
>
> **W1.** Thank you for pointing out this issue. We would like to address your question from two different perspectives:
>
> 1. **Regularization and Scalability**: In our work, we demonstrate the strong duality theorem (Theorem 1), which shows that the DRO learning problem can be transformed into an empirical risk minimization (ERM) problem with an added regularizer. This transformation removes the need to compute the worst-case loss quantity directly, making the computation **more scalable**. Previous works, such as [1] and [2], support the use of algorithms that incorporate regularizers to improve learning efficiency.
>
> 2. **Curse of Dimensionality**: The curse of dimensionality is a significant concern in traditional DRO, where the convergence rate of the distance between the empirical measure and the underlying distribution is affected by the feature space dimension:
> $$W(\mathbf{P}\_{N}, \mathbf{P}_*) = O(N^{-\frac{1}{d}})$$
>
> However, if we assume that the distribution $\mathbf{P}_\ast$ is derived from a SCM, this rate improves to $O(N^{-\frac{1}{2}})$, effectively breaking the curse of dimensionality. We plan to explore and demonstrate this point in detail in our following work.
>
> **Q1.** In our experiment, we assume that the data is generated by an additive noise model. We first learn the functional structure, and then design our causally fair dissimilarity function. Using the theorems, we can subsequently calculate the worst-case loss quantity.
>
> If we do not make assumptions about causality (e.g. i.i.d. case) and do not have a protected variable, our results are equivalent to traditional Wasserstein DRO, ensuring that everything functions as expected.
>
>
> [1] Hong TM Chu, Kim-Chuan Toh, and Yangjing Zhang. On regularized square-root regression 391 problems: distributionally robust interpretation and fast computations. Journal of Machine 392 Learning Research, 23(308):1–39, 2022.
>
> [2] Yangjing Zhang, Ning Zhang, Defeng Sun, and Kim-Chuan Toh. An efficient hessian based 530 algorithm for solving large-scale sparse group lasso problems. Mathematical Programming, 531 179:223–263, 2020.

---

> > ### Comment · Reviewer_rhFX · 2024-08-13
> > **Reply to rebuttal**
> >
> > I thank the authors for their clarification. I would like to maintain my score.

---

### Official Review · Reviewer_rNAj · 2024-07-12

**Soundness:** 3
**Presentation:** 3
**Contribution:** 3
**Rating:** 7
**Confidence:** 3

**Summary:**

This submission studies the connection between Wasserstein Distributionally Robust Optimization (DRO) and individual fairness in certain Structural Causal Models (SCMs). Namely, it is first shown that, in the case that the SCM at hand is an Additive Noise Model (ANM) with known structural equations, one may define a Causally Fair Dissimilarity Function (CFDF) on the feature space in a canonical manner.

With this, the remainder of the paper concerns the problem of DRO of the risk function (i.e. minimization of $\mathcal R_{\delta}(\mathbb P,\theta)$ the worst-case risk over all distributions which lie in a (Wasserstein) ball of radius $\delta$ from $\mathbb P$). Notably, a dual form for $\mathcal R_{\delta}$ is provided by extending known results; this effectively converts the infinite-dimensional maximization problem defining $\mathcal R_{\delta}$ to a finite dimensional minimization problem. In the case that the center $\mathbb P$ for the Wasserstein ball in $\mathcal R_{\delta}$ is an empirical measure from $N$ samples, $\mathbb P_N$, it is shown that, under certain assumptions on the loss function, $\mathcal R_{\delta}$ can be recast exactly in terms of the standard empirical risk or the objective from the counterfactually robust optimization problem depending on the size of the diameter of the set of sensitive attributes. Under weaker assumptions on the loss and that the SCM is linear, a first order (in $\delta$) expansion of $\mathcal R_{\delta}$ is also provided. Next, it is demonstrated that standard adversarial optimization can be used to approximate DRO. Finally, the rate of convergence of $\mathcal R_{\delta}(\mathbb P_{\star},\hat \theta_N^{\mathrm{dro}})$ to $\inf_{\theta\in\Theta}\mathcal R_{\delta}(\mathbb P_{\star},\theta)$ is characterized.

The paper concludes with a numerical study of the described causally fair DRO (CDRO). Namely, a comparison between the CDRO and other common approaches is provided on real-world and synthetic datasets. It is shown empirically that the CDRO exhibits slightly lower accuracy than the other models, but yields a lower unfair area (this is especially evident in the COMPAS and LIN datasets).

**Strengths:**

The paper is well-written and its contributions are clearly identified relative to the existing body of work.

Although the connection between DRO and individual fairness has been considered before (in the linear SCM case), I believe the extension to the ANM case is of interest. Furthermore, the section on duality and corresponding representations of $\mathcal R_{\delta}$ provide a nice interpretation nice interpretation for this approach.

**Weaknesses:**

1. It is difficult to get a sense for how strong some of the assumptions made in this work are. Although most of the assumptions are coupled with some examples of cases where they apply, I believe it would be relevant to provide some rationale for why these assumptions are necessary or describe primitive classes of examples where these assumptions hold rather than just some specific examples.

2. Assumption 2 (iii) requires estimation of the CFDF; it would be useful to expand a bit more on this assumption keeping in mind the above point or at least provide some heuristics for what rates one can expect in general.

3. More generally, the derived results would benefit from some additional discussion regarding their implications.

**Questions:**

1. I believe there is a small mistake regarding assumptions (i)-(ii) and the given examples. Notably the quantile loss is 1-Lipschitz, but $|h(t_0+t_k)-h(t_0)|=|\gamma t_k|$ if $t_0\geq 0$ or $|(\gamma-1)t_0+\gamma t_k|$ otherwise. In either case the limit from assumption (ii) is $\gamma\neq 1$. Perhaps there is a sign mistake?

2. In Theorem 4 it is stated that the necessary condition for the existence of an infinite DRO solution is that [...]. Should it not be a finite DRO solution?

**Limitations:**

The authors address the limitations of their work in the conclusion and the broader impact is addressed in the appendix.

---

> ### Author Rebuttal · Authors · 2024-08-06
>
> First of all, I would like to thank you for your insightful comments. We appreciate the attention given to our work. In the following, I respond to the mentioned weaknesses and questions in detail.
>
> **W1.** In the global response, we explained the intuition behind our assumptions.
> Assumption 2 seems new, but it is common in finite sample guarantee theorems, such as Theorem 6 of [1]. The first part of Assumption 2 is natural, assuming the feature space is bounded and the parameter space is bounded and closed.
>
> The third part is also reasonable, relating to the statistical properties of the estimator for the cost function or causal model. For example, in a linear SCM, we can estimate its functional structure with a convergence rate of $O(N^{-\frac{1}{2}})$, allowing us to design a metric that satisfies the third assumption.
>
> The second part of Assumption 2 is the Lipschitz condition on perturbing non-sensitive features, derived from counterfactuals. For example, consider a linear SCM with reduced-form mapping $M$, where $X = MU$ and all sensitive attributes are parents. Here, $CF_0(v, \Delta) = v + M\Delta$. Given a Lipschitz loss function $\ell(v, y, \theta) = h(\theta^T v - y)$ and an $\ell_p$ norm in the exogenous space, we get $d(v, CF_0(v, \Delta)) = \\|\Delta\\|_q$. This makes the Lipschitz condition:
>
> $$
> \vert \ell(v, y, \theta) - \ell(CF_0(v, \Delta), y, \theta) \vert = \vert h(\theta^T v - y) - h(\theta^T v - y + \theta^T M \Delta) \vert \leq L \|\theta^T M\|_p \\|\Delta\\|_q
> $$
>
> If you need further clarification, we would be happy to engage during the discussion period.
>
> **W2.** Thank you for highlighting this point. Assumption 2(iii), related to the estimation of ANM, depends on the functional structure and smoothness of the model, resulting in different convergence rate orders. For example, for a linear ANM, the convergence rate is $O(N^{-1/2})$. We omitted a detailed discussion on this rate as it pertains to the estimation of SCMs more broadly.
> Each estimator with convergent rate $O(N^{-\alpha})$ where $\alpha>0$ works in Theorem 5.
> In the revised version, we will add explanations to clarify this topic further.
>
> **W3.** To present a minimal and complete version of our proposed method, we focused heavily on the theoretical section. To address its applications and implications, we detailed its relation to previous research and included numerous references.
>
> **Q1.** We appreciate your attention. The words **for each** and **exists** were used incorrectly, and we have fixed them in the revised version. The correct assumption is:
>
> - For each $t_0 \in \mathbb{R}$, there exists a sequence  $\\{t_k\\}$ that goes to $\infty$ such that we have
> $$
> \lim_{k \rightarrow \infty} \dfrac{|h(t_0 + t_k) - h(t_0)|}{|t_k|} = L_h
> $$
>
> To provide intuition behind this assumption, for each $t_0$ and each $\epsilon >0$, we need to find $t_\ast$ such that $|h(t_\ast)-h(t_0)|>(L_h -\epsilon) |t_\ast-t_0|$ with $|t_\ast-t_0| > \delta$. Assumption (ii) addresses this without considering the magnitude of $\delta$.
>
> As you mentioned, the quantile loss is $1$-Lipschitz. Let $t_0$ be given. If we choose $\\{t_k\\}$ that goes to $-\infty$, then
> $$
> \lim_{k \rightarrow \infty} \dfrac{|h(t_0 + t_k) - h(t_0)|}{|t_k|} = 1
> $$
> Therefore, our assumption is satisfied.
>
>
> **Q2.** We apologize for the oversight; in fact, this should be finite. We corrected it.
>
> [1] Blanchet, Jose, Karthyek Murthy, and Viet Anh Nguyen. "Statistical analysis of Wasserstein distributionally robust estimators."

---

> > ### Comment · Reviewer_rNAj · 2024-08-09
> >
> > I have read the authors' rebuttal.
> > Their response has answered the questions I raised.

---

### Official Review · Reviewer_vsYr · 2024-07-12

**Soundness:** 3
**Presentation:** 2
**Contribution:** 3
**Rating:** 5
**Confidence:** 4

**Summary:**

This paper proposes a novel framework called Causally Fair Distributionally Robust Optimization (CDRO) to address individual fairness in machine learning. It combines causal modeling with distributionally robust optimization, using a causally fair dissimilarity function (CFDF) to measure individual similarity while considering sensitive attributes. The framework provides a strong duality theorem, enabling efficient computation of worst-case losses under distributional uncertainty. It offers explicit solutions for the regularizer in linear Structural Causal Models (SCMs) and estimates it for non-linear SCMs, mitigating overfitting and ensuring fairness. Additionally, the framework provides finite sample guarantees for convergence even with unknown SCMs, enhancing its practicality. Empirical evaluations on real-world and synthetic datasets demonstrate CDRO's effectiveness in reducing unfairness while maintaining accuracy compared to other methods.

**Strengths:**

- Introduces a new framework that integrates causality, individual fairness, and adversarial robustness into DRO, providing a comprehensive approach to address fairness concerns in machine learning.

- Offers several theoretical advancements, including a strong duality theorem, explicit regularizer formulations, and finite sample guarantees, contributing to the theoretical foundation of fair and robust machine learning.

- The framework is designed to be practical for real-world applications, even when the underlying causal structure is unknown, making it a valuable tool for addressing fairness in various domains.

**Weaknesses:**

- Please define the abbreviation SCM before its first use in line 49.

- The experimental setting description could be improved. Consider providing an algorithm for the proposed approach to enhance clarity.

- The paper appears to have considerable overlap with Ehyaei et al. (https://arxiv.org/pdf/2310.19391):
  - The first contribution claimed in Section 1.1 (line 64) seems to have been previously established by Ehyaei et al.
  - Definition 1 and Proposition 1 in Section 3 appear to closely resemble Definition 2 and Proposition 1 in Ehyaei et al. Please clarify the novelty of these elements.

- The framework's reliance on an additive noise model assumption may limit its applicability in complex real-world scenarios. Could you discuss potential impacts on CFDF accuracy and fairness guarantees, and any plans to address this limitation?

**Questions:**

Given that the proposed approach demonstrates lower prediction accuracy compared to existing methods, could the authors provide insights into potential factors contributing to this outcome? Additionally, how might this trade-off between prediction accuracy and other performance metrics be justified in the context of the method's overall objectives?

---

> ### Author Rebuttal · Authors · 2024-08-06
>
> Thank you for your helpful comments. We will address each point of weakness and questions, labeled **Wi** and **Qi** respectively, in order.
>
> **W1.** We will add the Structural Causal Model (SCM) in line 49.
>
> **W2.** We appreciate your comment and agree that the numerical section could be improved. As mentioned in our global response, our method is not specific to fair learning. In the numerical section, we aim to showcase just one of the many applications of this framework. However, we faced challenges in finding datasets and methods compatible with our problem.
>
> The algorithms in our work are similar to those used in regular optimal transport, such as the Sinkhorn algorithm [1]. In these algorithms, we simply replace the matrix of the cost function with the one calculated based on the CFDF function.
>
> **W3.** As mentioned in line 54, we adopt the definition of a fair metric from Ehyaei et al. to define a causally fair dissimilarity function (CFDF). However, their notion of a fair metric is not compatible with our assumptions. We require a more general definition, namely a dissimilarity function  because our cost function does not satisfy some of the fundamental properties of a metric, such as:
>
> - **Identity of Indiscernibles**: $d(x, y) = 0$ if and only if $x = y$.
> - **Triangle Inequality**: $d(x, z) \leq d(x, y) + d(y, z)$.
>
> Therefore, to avoid misleading use of the term "metric" and to have broader applicability, we define a general notion of a dissimilarity function instead of a metric, which has weaker assumptions.
>
> To obtain a representation theorem of the CFDF, we need Proposition 1. Unfortunately, Proposition 1 in the work of Ehyaei et al. does not work in the general case and is limited to metrics.
>
> **W4.** As mentioned in the global response, since our work uses counterfactuals from structural causal models, our model must be counterfactually identifiable. Therefore, we should base our method on counterfactually identifiable SCMs, such as the bijective generative mechanism (BGM). To avoid additional complexity, we focus on the additive noise model, a specific BGM instance. However, our results are valid for general BGM.
>
> In our framework, the primary concern is the counterfactual identifiability of the SCM model. Once the SCM model or corresponding cost function is estimated, Theorem 5 guarantees that the learning problem has a convergent solution for finite samples.
>
> **Q1.** In fair learning, it is well known that there is a trade-off between individual fairness and accuracy. Achieving individual fairness necessitates adjusting the model to account for variations across individuals, which increases complexity and the potential for overfitting. This adjustment may reduce accuracy, as the model must balance fitting the overall data distribution with adhering to fairness constraints, potentially limiting its predictive precision. Therefore, increasing fairness often requires sacrificing some degree of accuracy.
>
> This trade-off is justified by the method's primary objectives: reducing disparate impact and ensuring equitable treatment across individuals. While accuracy is important, the approach prioritizes mitigating biases to promote fairness and inclusion in algorithmic decision-making. Thus, the intentional trade-off of reduced accuracy aims to achieve a more equitable and socially responsible model.
>
> [1] Cuturi, M. (2013). Sinkhorn Distances: Lightspeed Computation of Optimal Transport. Advances in Neural Information Processing Systems, 26, 2292–2300.

---

> > ### Comment · Reviewer_vsYr · 2024-08-10
> >
> > Thank you for your responses! My concerns have been partially addressed, and I am willing to raise my score to 5. However, as you mentioned, "we faced challenges in finding datasets and methods compatible with our problem," I still have reservations about the applicability of the proposed approach to a wide range of real-world data scenarios.

---

> > > ### Author Response · Authors · 2024-08-11
> > > **Enhancing Causal Consistency in Optimal Transport Applications**
> > >
> > > Thank you for your insightful comments, which have improved the clarity and impact of our work.
> > >
> > > Our primary goal was to establish a theoretical framework for optimal transport tools tailored to dissimilarity cost functions derived from causal models, especially when data originates from such models. We argue that traditional metrics like $l\_p$ norms may not preserve causal relationships in these scenarios.
> > >
> > > Due to space constraints, we focused on a fair learning example to demonstrate our method's effectiveness, though many applications remain. A particularly promising application is in generative adversarial networks, where our approach, not only compares distributions but also preserves the original data's causal structure.

---

### Author Rebuttal · Authors · 2024-08-06

We thank the reviewers for their valuable feedback and constructive comments. We are honored to have received your attention.

**Motivation:** Distributionally Robust Optimization (DRO) is a data-driven framework addressing out-of-sample challenges, such as distribution overfitting or shifts, using an adversarial approach. It defines a distributional ambiguity set (DAS) around the estimated true probability based on empirical measures, ensuring the true distribution lies within this set.

In conventional Wasserstein DRO, the DAS includes all probability distributions within a certain Wasserstein distance, constructed by a metric on space (e.g., $\ell_p$ norm) from the empirical distribution, $B^W_{\delta}(\mathbb{P}) = \\{\mathbb{Q} \in \mathcal{P}(\mathcal{X}): W(\mathbb{P}, \mathbb{Q}) \leq \delta \\}$. This works well when data lacks specific structures, but when data or distribution shifts follow structures like temporal patterns or causal relationships, the Wasserstein DAS must be pruned to exclude unrealistic scenarios; otherwise, models become overly conservative and lose accuracy.

For example, consider a model predicting income based on gender, age, and education. In the sample data, the average age is 25, and the educational level is 3 (Upper is highly educated). Using a simple $\ell_p$ cost in Wasserstein distance, a population with an average age of 20 and educational level of 3.5 is treated the same as another with an age of 30 and level of 2.5 by the conventional Wasserstein DAS. However, education depends on age, making the first scenario unrealistic.

**Contribution:** To address this, we derive the transportation cost function from the estimated  causal structure instead of conventional norms, preventing impossible scenarios. Figure 1 (uploaded PDF) shows that a causal cost function results in a DAS that includes the true underlying probability with fewer unrealistic scenarios.

If there is a protected variable, the cost function must capture the proximity between an instance $v$ and its counterfactual $v_a$ from the causal structure, not simply by changing the labels of the instances. In Section 3, especially in Example 1, we demonstrate that $\ell_p$ norms cannot capture counterfactual proximity. We propose the causally fair dissimilarity function (CFDF) to address this.

Since the CFDF needs to capture the similarity between instances and their counterfactuals, counterfactuals must be identifiable from sample data, which requires the SCM to be counterfactually identifiable. We chose the additive noise model (ANM) to avoid added mathematical complexity. While our results apply to bijective generative mechanisms [1], a broad class of SCMs that are counterfactually identifiable. ANM is often preferred over general SCMs due to its simplicity, interpretability, and effective handling of noise. This makes it ideal for fields like statistics, causal inference, signal processing, image processing, economics, and social science, where additive noise is prevalent.

The assumptions behind CFDF are intuitive. Since the variables in the exogenous space are mutually independent (by assumption), assume each has its own cost function, which can be combined through product topology. This allows the pushforward of the CFDF to have a simpler form in the exogenous space.
 Unfortunately, capturing causality in CFDF means it is not a true metric because it lacks the positivity property, which means $d(v,v_a) = 0 \nRightarrow v = v_a$. Most optimal transport facts assume a metric-based cost function. Using CFDF instead of a metric posed significant challenges and added theoretical
 complexity to our work.

After introducing CFDF, we prove the Strong Duality Theorem to demonstrate our proposed DAS's real-world applicability. This theorem shows DRO problem converts into a tractable, computational efficient form. Theorems 2 and 3 showcases the effectiveness of our approach. For linear SCMs, many popular models (Examples 2 and 3) provide closed-form solutions for worst-case loss, which enables fast learning algorithms by eliminating worst-case step computation.

To ensure compatibility with complex nonlinear SCMs and neural networks, we present a first-order regularizer estimation in Theorem 4. We address the relationship between DRO and classical robust optimization in Proposition 2.

A key challenge is that CFDF is based on SCM's functional structure, which must be estimated in real applications. Theorem 5 shows how causally fair DRO performs well with an estimated structure, that demonstrates a finite sample theorem. Our results rely on Assumption 2, which is common in such theorems. Part (ii) of Assumption 2 is new, requiring the loss function to have a Lipschitz property for non-sensitive attribute perturbations. Since in SCM, perturbation is obtained by counterfactual, so this property is expressed like this.

Regarding the numerical experiment, we agree that this section could include more examples and datasets. Our framework is a general approach for DRO with causality and protected variables, applicable in areas like Fairness, Adversarial Learning, Reinforcement Learning, and NLP. In this paper, we applied it to fair learning to demonstrate one application given the limited space and availability of appropriate dataset/use cases.

We believe that our method efficiently captures the true underlying probability without unrealistic DAS scenarios, as shown in our results. Future work will need to demonstrate additional properties, such as breaking the curse of dimensionality in traditional DRO problems.

We appreciate your comments on notation and typesetting and have incorporated them into the revised version. If you need further clarification, we would be happy to discuss this during the review period. Next, we will address each reviewer's weaknesses and questions in more detail.

[1] Nasr-Esfahany et al. [2023], Counterfactual identifiability of bijective causal models.

---

### Decision · Program_Chairs · 2024-09-25

**Decision:**

Accept (poster)

**Comment:**

The paper introduces Causally Fair Distributionally Robust Optimization (CDRO) framework to address individual fairness in machine learning using Wasserstein Distributionally Robust Optimization (DRO). The framework integrates causal modeling with DRO to handle fairness concerns involving sensitive attributes. It presents a dual formulation of DRO that simplifies the problem into a more tractable form, estimates regularizers for linear and non-linear Structural Causal Models (SCMs), and provides finite sample error bounds. Standard fairness evaluation datasets are used in empirical evaluations to demonstrate the effectiveness of the proposed method.

The reviewers found the combination of causal modeling, DRO, and rigorous optimization frameworks to address individual fairness novel. They appreciate the novelty in theory and relevance in practice. In general, the paper is found beneficial to the community with sufficient contributions. As for feedback for improvement, the reviewer mentioned that the paper could benefit from clearer explanations and more examples to improve readability. They also noted that the literature review part could expand significantly, and the discussions with existing works can further improve. There were additional comments for improving the readability and presentation of the paper (please see individual reviews).